# O-GlcNAcylation of circadian clock protein Bmal1 impairs cognitive function in diabetic mice

Ya Hui[1,2,3,7], Yuanmei Zhong[1,3,4,7], Liuyu Kuang[1,4,5,7], Jingxi Xu[1,2,3,7], Yuqi Hao[1,2,3], Jingxue Cao[1,2,4] & Tianpeng Zheng [ID] [1,2,3,6✉]

## Abstract

Neuronal damage in the hippocampus induced by high glucose has been shown to promote the onset and development of cognitive impairment in diabetes, but the underlying molecular mechanism remains unclear. Guided by single-cell RNA sequencing, we here report that high glucose increases O-GlcNAcylation of Bmal1 in hippocampal neurons. This glycosylation promotes the binding of Clock to Bmal1, resulting in the expression of transcription factor Bhlhe41 and its target Dnajb4. Upregulated Dnajb4 in turn leads to ubiquitination and degradation of the mitochondrial Na + /Ca2+ exchanger NCLX, thereby inducing mitochondrial calcium overload that causes neuronal damage and cognitive impairment in mice. Notably, Bhlhe41 downregulation or treatment with a short peptide that specifically blocks O-GlcNAcylation of Bmal1 on Ser424 mitigated these adverse effects in diabetic mouse models. These data highlight the crucial role of O-GlcNAcylation in circadian clock gene expression and may facilitate the design of targeted therapies for diabetes-associated cognitive impairment.

**Keywords** Diabetes; Cognitive Impairment; O-GlcNAcylation; Bmal1-Clock Complex; Mitochondria Calcium Overload
**Subject Categories** Metabolism; Neuroscience; Post-translational Modifications & Proteolysis

## Introduction

High glucose in diabetic patients is a key pathogenic factor that contributes to cognitive impairment (Rawlings et al, 2019). In addition, previous studies have shown that high-glucose environment increased mitochondrial reactive oxygen species (mtROS) accumulation and promoted apoptosis in hippocampal neurons (Hui et al, 2023). Although high-glucose-induced neuronal damage in the hippocampus has been proven to promote the onset and development of cognitive impairment in diabetes, the underlying mechanisms remain poorly understood.

Mitochondria are essential organelles for a multitude of cellular processes, such as ATP production, reactive oxygen species generation, and apoptosis induction (Pathak and Trebak, 2018). In addition, the maintenance of mitochondrial calcium homeostasis has been demonstrated to be a crucial factor in preserving mitochondrial function. This homeostasis is regulated by various factors that control calcium influx and efflux, and any disturbance to this balance may result in mitochondrial calcium overload and dysfunction (Garbincius and Elrod, 2022). It has been demonstrated that mitochondrial dysfunction plays a key role in the pathogenesis of hippocampal neuronal damage by promoting oxidative stress and inflammation. Furthermore, a study revealed that disruption of mitochondrial calcium homeostasis by high glucose promoted mitochondrial dysfunction and subsequently led to diabetic cardiomyopathy (Wu et al, 2019). However, whether and how high-glucose-induced dysregulation of mitochondrial calcium homeostasis contributes to hippocampal neuronal damage and cognitive impairment in diabetes is largely unknown.

The circadian clock is a crucial endogenous regulatory mechanism that exists in nearly all cells throughout the body (Saran et al, 2020). The clock genes have been proven to function as a mediator that regulates both circadian and fundamental metabolic processes such as glucose metabolism. Therefore, aberrant clock gene expressions not only lead to the dysregulation of physiological rhythms but also induce metabolic disorders such as diabetes and its complications (Sato et al, 2018). In addition, previous study by Li et al demonstrated that clock proteins such as Brain and Muscle ARNT-Like 1 (Bmal1) and Circadian Locomoter Output Cycles Kaput (Clock) could be O-GlcNAcylated and this protein modification increased Bmal1 and Clock expression by inhibiting their ubiquitination under high-glucose conditions (Li et al, 2013). These data suggest an interplay between clock proteins and glucose metabolism. More importantly, Clock genes have also been proven to play a major role in maintaining mitochondrial function (Abdel-Rahman et al, 2021; Jordan et al, 2017). However,

¹Department of Endocrinology and Metabolism, The Second Affiliated Hospital of Guilin Medical University, 541199 Guilin, Guangxi, P. R. China. ²Guangxi Clinical Research Center for Diabetes and Metabolic Diseases, The Second Affiliated Hospital of Guilin Medical University, 541199 Guilin, Guangxi, P. R. China. ³Guangxi Key Laboratory of Diabetic Systems Medicine, Guilin Medical University, 541199 Guilin, Guangxi, P. R. China. ⁴Guangxi Key Laboratory of Brain and Cognitive Neuroscience, Guilin Medical University, 541199 Guilin, Guangxi, P. R. China. ⁵Guangxi Key Laboratory of Metabolic Reprogramming and Intelligent Medical Engineering for Chronic Diseases, The Second Affiliated Hospital of Guilin Medical University, 541199 Guilin, Guangxi, P. R. China. ⁶Guangxi Health Commission Key Laboratory of Glucose and Lipid Metabolism Disorders, The Second Affiliated Hospital of Guilin Medical University, 541199 Guilin, Guangxi, P. R. China. ⁷These authors contributed equally: Ya Hui, Yuanmei Zhong, Liuyu Kuang, Jingxi Xu. ✉E-mail: ZhengTP@glmc.edu.cn

it remains unknown whether high glucose could disrupt mitochondrial calcium homeostasis by affecting clock gene expression in hippocampal neurons and subsequently lead to neuronal damage and cognitive impairment in diabetes.

Herein, guided by single-cell RNA sequencing (scRNA-seq), we found that high glucose increased Bmal1 O-GlcNAcylation at S424 in hippocampal neurons. This glycosylation of Bmal1 further promoted the binding of Clock to Bmal1, which upregulated Class E basic helix-loop-helix protein 41 (Bhlhe41) and DnaJ homolog subfamily B member 4 (Dnajb4) expression and led to ubiquitination and degradation of mitochondrial $Na^+/Ca^{2+}/Li^+$ exchanger (NCLX), thereby inducing mitochondrial calcium overload and subsequently leading to neuronal damage and cognitive impairment in diabetes. Notably, we designed a short peptide to target O-GlcNAcylation of Bmal1 S424. This peptide presents evident inhibition of Bmal1 S424 O-GlcNAcylation and protects against high-glucose-induced neuronal apoptosis and cognitive dysfunction by regulating Bhlhe41/Dnajb4/NCLX signaling. These results provide a mechanistic view of how high glucose leads to hippocampal neuronal damage to initiate cognitive dysfunction in diabetes.

# Results

## High glucose impairs cognitive function through upregulating Bhlhe41 expression in hippocampal neurons

Leptin receptor-deficient db/db mice are a widely used animal model to study diabetes and its complications. To investigate the underlying mechanism of high-glucose-induced cognitive impairment, we first analyzed previously published single-cell RNA sequencing (scRNA-seq) data from the hippocampus of db/m and db/db mice (Ma et al, 2022). Appendix Fig. S1A–C showed the scRNA-seq workflow and PCA and UMAP analysis after quality control (Appendix Fig. S1D). All cells were classified into 24 cell clusters (Appendix Fig. S1C) and divided into 9 distinct cell types by using SingleR package (Appendix Fig. S1E). The fraction of db/m vs db/db cells per cluster was presented in Appendix Fig. S1F.

Since hippocampal neurons have long been known to play an important role in a variety of cognitive functions (Mikulovic et al, 2018; Miller et al, 2018), we further performed single-cell regulatory network inference and clustering (pySCENIC) analysis of hippocampal neurons to investigate the transcription factors (TFs) that may play a regulatory role in the pathogenesis of cognitive impairment in diabetes. Figure 1A listed the five most upregulated and five most downregulated regulon activity of TFs in db/db group. As shown by pySCENIC regulon areas under the curve per cell scores, circadian rhythm and mitochondrial damage-related TFs such as Bhlhe41, Trp53, Stat2, and Smad1 exhibited significantly higher transcriptional activity in db/db group than in db/m group. Notably, among the top 5 TFs with increased transcriptional activity, we found that Bhlhe41 gene expression was also significantly increased in the hippocampal neurons of db/db mice (Fig. 1A). To verify this finding from scRNA-seq analysis, we examined gene and protein expression of Bhlhe41 in mouse primary hippocampal neurons that received normal glucose (5.5 mmol/L D-glucose), high glucose (25 mmol/L D-glucose) or osmotic control (5.5 mmol/L D-glucose plus 19.5 mmol/L D-mannitol) treatment. As shown in Fig. 1B and Appendix Fig. S1G, the gene and protein expression of Bhlhe41 were significantly increased in high-glucose-treated neurons. Similarly, db/db mice exhibited higher levels of Bhlhe41 protein expression in hippocampal CA1 neurons than db/m controls (Fig. 1C). To demonstrate that the differences in Bhlhe41 abundance in vivo are not due to the differences in the timing of tissue collection, hippocampal CA1 neurons from all groups of mice were collected from 8:00 AM to 9:30 AM, additionally, we determined CA1 Bhlhe41 expression in db/m and db/db littermates at 8 AM, 9 AM, and 10 AM, respectively, and did not find any within-group differences (Appendix Fig. S1H). The purity of isolated hippocampal CA1 neurons was assessed using flow cytometry (Appendix Fig. S1I). Since the CA1 region of the hippocampus is involved in the acquisition and maintaining of learning and memory function (Sun et al, 2019), we restricted our subsequent in vivo experiments to CA1 subfield.

To mechanistically link the high-glucose-mediated Bhlhe41 upregulation with cognitive impairment in type 2 diabetes, we specifically eliminated Bhlhe41 expression in hippocampal CA1 neurons by crossing db/m mice with Bhlhe41$^{fl/+}$ mice (Appendix Fig. S1J) to generate db/db Bhlhe41$^{fl/fl}$ mice (Appendix Fig. S1K), and bilaterally infusing adeno-associated virus (AAV) containing either a control or CAMKII-Cre vector construct into the hippocampal CA1 region of db/db Bhlhe41$^{fl/fl}$ mice (Appendix Fig. S1L). After two weeks, Western blot detecting Bhlhe41 confirmed that its expression was significantly reduced in the AAV-CAMKII-Cre-injected hippocampal CA1 neurons as compared to controls (Fig. 1D). Remarkably, db/db Bhlhe41$^{fl/fl}$ mice treated with AAV-CAMKII-Cre construct showed significant improvement in cognitive function versus db/db Bhlhe41$^{fl/fl}$ mice infused with the control vector, suggesting that depletion of Bhlhe41 reversed the cognitive deficits in db/db mice (Fig. 1E; Appendix Fig. S1M,N). To further establish that these observed results were not due to leptin receptor deficiency in db/db mice, we assessed the effects of hippocampal Bhlhe41 deletion on cognitive performance in C57BL/6 Bhlhe41$^{fl/fl}$ mice treated with vehicle or streptozotocin (STZ). STZ has been shown to possess diabetogenic properties leading to the destruction of islet β-cell (Lenzen, 2008). Consistently, Bhlhe41 deletion in hippocampal CA1 neurons (Appendix Fig. S1O) successfully improved cognitive function in STZ-treated diabetic mice (Fig. 1F; Appendix Fig. S1P,Q). Overall, these findings indicated that high glucose impaired cognitive function by upregulating Bhlhe41 Expression in hippocampal CA1 neurons.

## Bhlhe41 contributes to mitochondrial calcium overload and oxidative damage in hippocampal neurons under diabetic conditions

To further clarify the mechanism underlying Bhlhe41-induced cognitive impairment in diabetes, we selected differentially expressed genes in hippocampal neurons scRNA-seq dataset for KOBAS analysis (Fig. 2A). Up- and downregulated genes were related to calcium signaling pathway (Ddx5, Gap43, Adcy8, Marcksl1), oxidative stress response (Plcb1, S100a6, Dst) and apoptosis (Fos, Ctsb, Map3k5, Mapk10, Diablo, Actg1) (Fig. 2B). Since mitochondrial dysfunction and oxidative damage have been implicated in the pathogenesis of neuronal apoptosis and cognitive

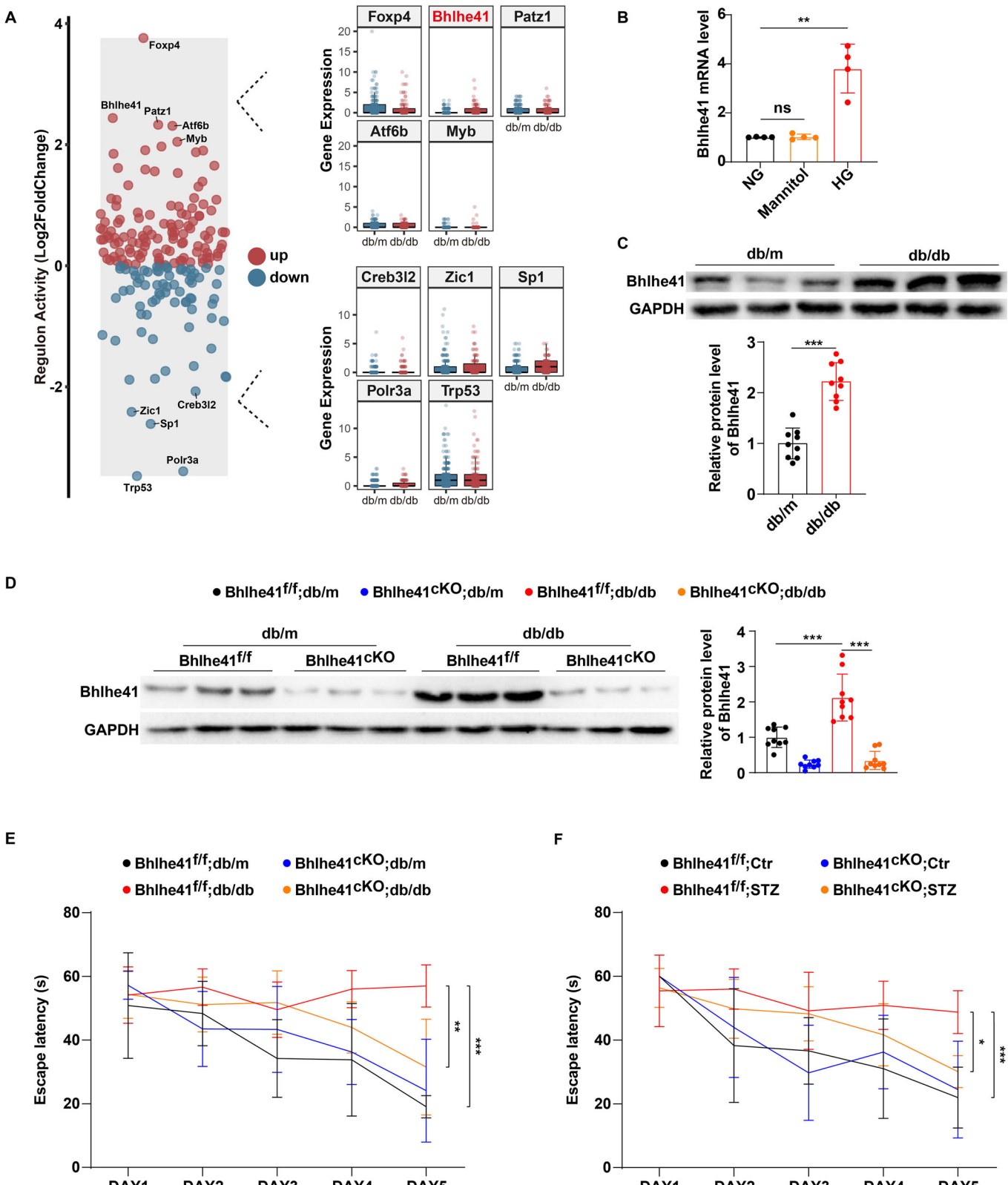

impairment (Li et al, 2023; Sun et al, 2022), hippocampal neurons isolated from CA1 region of db/db and STZ-induced diabetic mice were subjected to mitochondrial function and apoptosis analysis.

As shown in Fig. 2C–E and Appendix Fig. S2A–C, hippocampal CA1 neurons isolated from db/db and STZ-induced diabetic mice exhibited higher apoptosis (Fig. 2D; Appendix Fig. S2B) (as evidenced by decreased expression of Bcl-2 and increased

**Figure 1.  High glucose impairs cognitive function through upregulating Bhlhe41 expression in hippocampal neurons.**

(A) Differential regulon activity analysis showing the five most up- and downregulated regulons in hippocampal neurons of db/db mice (left). Gene expression levels of the five most up- (upper right) and downregulated (lower right) regulons. Boxes represent the first and third quartiles, with the medians indicated by the central lines. Whiskers extend to the lowest and highest values within 1.5 times the interquartile range ($n = 1$ mouse per group). (B) mRNA levels of Bhlhe41 in primary hippocampal neurons treated with normal glucose (NG, 5.5 mmol/L D-glucose), osmotic control (Mannitol, 19.5 mmol/L D-mannitol plus 5.5 mmol/L D-glucose), or high glucose (HG, 25 mmol/L D-glucose) for 48 h ($n = 4$ replicates). P values: 0.8433 (NG vs Mannitol), 0.0014 (NG vs HG). (C) Immunoblots and quantification analysis of protein level of Bhlhe41 in the hippocampal CA1 neurons of db/m and db/db mice ($n = 9$ mice per group). P value: <0.0001. (D) Immunoblots and quantification analysis of protein level of Bhlhe41 in the hippocampal CA1 neurons of db/m Bhlhe41$^{fl/fl}$ or db/db Bhlhe41$^{fl/fl}$ mice injected with AAV-CAMKII or AAV-CAMKII-Cre ($n = 9$ mice per group). P values: <0.0001 (Bhlhe41$^{f/f}$;db/m vs Bhlhe41$^{f/f}$;db/db), <0.0001 (Bhlhe41$^{f/f}$;db/db vs Bhlhe41$^{cKO}$;db/db). (E, F) Escape latency to the platform during the training trials in a Morris water maze of db/m Bhlhe41$^{fl/fl}$, db/db Bhlhe41$^{fl/fl}$, Ctr Bhlhe41$^{fl/fl}$, STZ Bhlhe41$^{fl/fl}$ mice injected with AAV-CAMKII or AAV-CAMKII-Cre ($n = 6$ mice per group). P values: <0.0001 (E, Bhlhe41$^{f/f}$;db/m vs Bhlhe41$^{f/f}$;db/db), 0.0059 (E, Bhlhe41$^{f/f}$;db/db vs Bhlhe41$^{cKO}$;db/db), 0.0008 (F, Bhlhe41$^{f/f}$;Ctr vs Bhlhe41$^{f/f}$;STZ), 0.0188 (Bhlhe41$^{f/f}$;STZ vs Bhlhe41$^{cKO}$;STZ). Data are means ± SEM. ns, not significant, *$P < 0.05$, **$P < 0.01$, ***$P < 0.001$. Two-tailed Student's unpaired $t$ test analysis (C), one-way ANOVA followed by Tukey's test (B), two-way ANOVA followed by Tukey's test (D–F). Source data are available online for this figure.

expression of Bax and caspase3) and mtROS production (Fig. 2C; Appendix Fig. S2A) compared with their respective controls, and these effects were accompanied by a decrease in mitochondrial membrane potential (MMP) (Fig. 2E; Appendix Fig. S2C). Previous studies have shown that mitochondrial calcium homeostasis played a crucial role in maintaining mitochondrial function; therefore, we tested whether mitochondrial calcium homeostasis was disrupted in hippocampal CA1 neurons under high-glucose conditions. As shown in Fig. 2F and Appendix Fig. S2D, hippocampal CA1 neurons from db/db and STZ-induced diabetic mice demonstrated increased mitochondrial calcium levels as compared to their respective controls. Importantly, hippocampal CA1 neurons-specific Bhlhe41 deletion in db/db and STZ-induced diabetic mice resulted in a significant decrease in mitochondrial calcium concentrations within these neurons, and these effects were associated with restoration of mitochondrial function and reduced hippocampal neuron death (Fig. 2C–F; Appendix Fig. S2A–D). Taken together, we concluded that Bhlhe41 deletion effectively prevented excessive mitochondrial calcium accumulation and ameliorated mitochondrial dysfunction-induced oxidative injury in hippocampal neurons under diabetic conditions.

## Increasing O-GlcNAcylation of Bmal1 at S424 contributes to high-glucose-induced upregulation of Bhlhe41 expression in hippocampal neurons

To further clarify how high glucose upregulates Bhlhe41 expression in hippocampal CA1 neurons, we used online tool STRING and ALGGEN to identify potential upstream regulators of Bhlhe41. As shown in Appendix Fig. S3A,B, both STRING and ALGGEN analysis revealed a possible role for Bmal1/Clock heterodimer in regulating Bhlhe41 expression. Previous studies have also shown that this heterodimer functioned as a transcription factor complex to upregulate Bhlhe41 expression (Hamaguchi et al, 2004; Kato et al, 2014). Indeed, we found that Bmal1 overexpression (Fig. 3A) in mouse primary hippocampal neurons markedly increased Bhlhe41 mRNA (Appendix Fig. S3C) and protein levels (Fig. 3A). More importantly, knockdown of Bmal1 (Fig. 3B; Appendix Fig. S3D) decreased both of the mRNA and protein levels of Bhlhe41 in high-glucose-treated primary hippocampal neurons (Appendix Fig. S3E; Fig. 3B), suggesting that high glucose upregulated Bhlhe41 expression in hippocampal neurons through Bmal1.

The coordinated regulation of Cellular O-GlcNAcylation homeostasis is facilitated by the enzymes O-GlcNAcase (OGA) and O-GlcNAc transferase (OGT) (Chatham et al, 2021; Wu et al, 2017). The pathogenic mechanism of diabetic complications has been attributed to abnormal O-GlcNAcylation (Otomo et al, 2020; Peterson and Hart, 2016). To understand whether elevation of O-GlcNAcylation contributes to high-glucose-induced upregulation of Bhlhe41, we conducted an analysis of global O-GlcNAcylation levels in hippocampal neurons treated with high glucose. Our findings, as illustrated in Appendix Fig. S3F, indicated that primary mouse hippocampal neurons exhibited increased levels of global O-GlcNAcylation under high-glucose treatment. Subsequently, the mRNA and protein levels of Bhlhe41 were quantified in primary hippocampal neurons that were subjected to high-glucose treatment, with or without the presence of OGT inhibitor OSMI-1. Our results indicated that OGT inhibitor treatment effectively counteracted the high-glucose-induced upregulation of Bhlhe41 in hippocampal neurons, as demonstrated in Appendix Fig. S3G,H. Collectively, these findings provided evidence that the increase in O-GlcNAcylation levels was a contributing factor to the high-glucose-induced upregulation of Bhlhe41 in hippocampal neurons.

Based on the above findings that either Bmal1 knockdown or O-GlcNAcylation inhibition significantly reversed the high-glucose-induced upregulation of Bhlhe41 expression in hippocampal neurons, it is likely that Bhlhe41 upregulation through Bmal1 O-GlcNAcylation may be responsible for the observed effects. Reciprocal endogenous immunoprecipitation in hippocampal neurons confirmed that OGT bound Bmal1 (Appendix Fig. S3I). To further investigate whether Bmal1 was O-GlcNAcylated by OGT, we knocked down (Appendix Fig. S3J) or overexpressed (Appendix Fig. S3K) OGT, immunoprecipitated Bmal1, blotted with anti-O-GlcNAc antibody. As shown in Fig. 3C,D, OGT knockdown or overexpression significantly decreased (Fig. 3C) or increased (Fig. 3D) Bmal1 O-GlcNAcylation in hippocampal neurons, respectively. Collectively, OGT interacted with and O-GlcNAcylated Bmal1.

Since Bmal1 has been shown to form a heterodimer with Clock and positively regulate the transcription of downstream genes (Chen et al, 2020), we speculate that elevation of O-GlcNAcylation might promote the binding of Bmal1 and Clock. To verify this speculation, we ectopically expressed Bmal1 alone, or Bmal1 with Clock in HEK293T cells treated with or without high glucose, and

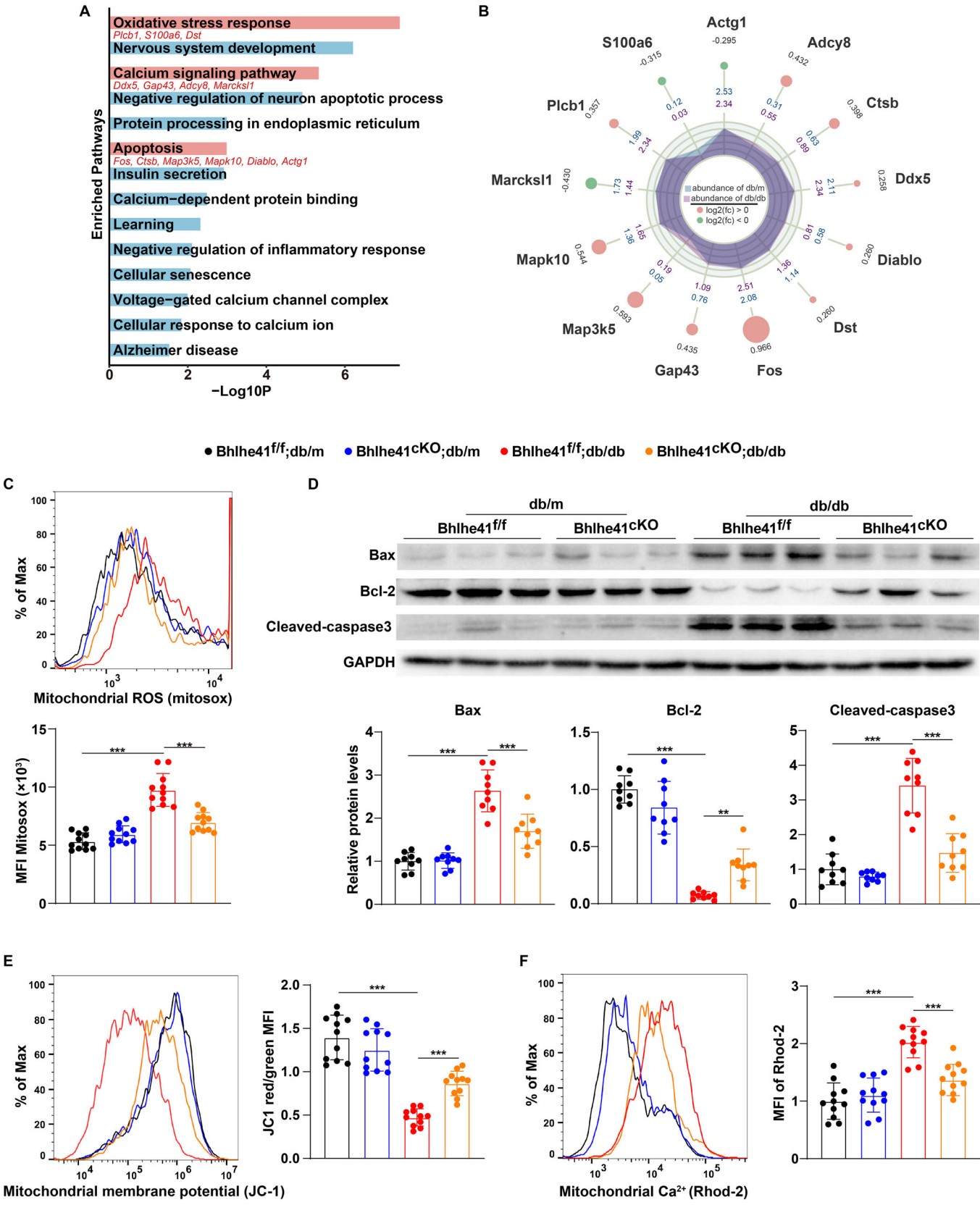

**Figure 2. Bhlhe41 contributes to mitochondrial calcium overload and oxidative damage in hippocampal neurons under diabetic conditions.**

(A) Pathway enrichment analysis for the differentially expressed genes (>1.2-fold) was performed with KOBAS. (B) Radar diagram showing differentially expressed genes related to calcium signaling pathway, oxidative stress response, and apoptosis. (C) Flow cytometry analysis of mtROS levels ($n = 11$ mice per group) in the hippocampal CA1 neurons of db/m Bhlhe41$^{fl/fl}$ or db/db Bhlhe41$^{fl/fl}$ mice injected with AAV-CAMKII or AAV-CAMKII-Cre. $P$ values: <0.0001 (Bhlhe41$^{f/f}$;db/m vs Bhlhe41$^{f/f}$;db/db), <0.0001 (Bhlhe41$^{f/f}$;db/db vs Bhlhe41$^{cKO}$;db/db). (D) Immunoblots and quantification analysis of protein levels of Bax, Bcl-2 and Cleaved-caspase3 ($n = 9$ mice per group) in the hippocampal CA1 neurons of db/m Bhlhe41$^{fl/fl}$ or db/db Bhlhe41$^{fl/fl}$ mice injected with AAV-CAMKII or AAV-CAMKII-Cre. $P$ values: <0.0001 (Bax, Bhlhe41$^{f/f}$;db/m vs Bhlhe41$^{f/f}$;db/db), <0.0001 (Bax, Bhlhe41$^{f/f}$;db/db vs Bhlhe41$^{cKO}$;db/db), <0.0001 (Bcl-2, Bhlhe41$^{f/f}$;db/m vs Bhlhe41$^{f/f}$;db/db), 0.003 (Bcl-2, Bhlhe41$^{f/f}$;db/db vs Bhlhe41$^{cKO}$;db/db), <0.0001 (Cleaved-caspase3, Bhlhe41$^{f/f}$;db/m vs Bhlhe41$^{f/f}$;db/db), <0.0001 (Cleaved-caspase3, Bhlhe41$^{f/f}$;db/db vs Bhlhe41$^{cKO}$;db/db). (E, F) Flow cytometry analysis of MMP (E, $n = 11$ mice per group) and mitochondrial calcium levels (F, $n = 11$ mice per group) in the hippocampal CA1 neurons of Bhlhe41$^{fl/fl}$ db/m or Bhlhe41$^{fl/fl}$ db/db mice injected with AAV-CAMKII or AAV-CAMKII-Cre. $P$ values: <0.0001 (E, Bhlhe41$^{f/f}$;db/m vs Bhlhe41$^{f/f}$;db/db), 0.0002 (E, Bhlhe41$^{f/f}$;db/db vs Bhlhe41$^{cKO}$;db/db), <0.0001 (F, Bhlhe41$^{f/f}$;db/m vs Bhlhe41$^{f/f}$;db/db), <0.0001 (F, Bhlhe41$^{f/f}$;db/db vs Bhlhe41$^{cKO}$;db/db). Data are means ± SEM. **$P < 0.01$, ***$P < 0.001$. Hypergeometric test (A), two-way ANOVA followed by Tukey's test (C–F). Source data are available online for this figure.

then performed immunoprecipitation. As shown in Appendix Fig. S3L, high-glucose treatment enhanced Bmal1 O-GlcNAcylation and increased the binding of Clock to Bmal1. However, OGT inhibitors successfully reversed the above effects induced by high glucose. Next, we examined whether endogenous Bmal1 O-GlcNAcylation could affect the interaction with Clock in mouse primary hippocampal neurons. As shown in Fig. 3E, the endogenous Bmal1 was highly glycosylated in high-glucose-treated hippocampal neurons and formed an increased Bmal1-Clock complex. Similarly, these effects were significantly reversed by OGT inhibitors, implying that high-glucose-induced Bmal1 O-GlcNAcylation promoted the binding of Clock to Bmal1 in hippocampal neurons.

O-GlcNAcylation prediction analyses were performed to identify potential O-GlcNAcylation sites on Bmal1. As shown in Appendix Fig. S3M, we found that S424, S518, and S573 were highly confident O-GlcNAcylation sites on Bmal1. Therefore, we generated S424A, S518A, and S573A mutants for COIP analysis and identified that Bmal1 serine 424 site was necessary for both Bmal1 O-GlcNAcylation and Bmal1-Clock complex formation in high-glucose-treated HEK293T cells (Fig. 3F). Consistently, S424A mutant significantly reversed high-glucose-induced Bmal1 O-GlcNAcylation in mouse primary hippocampal neurons with endogenous Bmal1 knockdown, accompanied by a decrease in binding with Clock and a downregulation of the protein and mRNA levels of Bhlhe41 (Fig. 3G; Appendix Fig. S3N,O). In addition, Chromatin immunoprecipitation analysis showed enhanced binding of Bmal1 to the Bhlhe41 promoter under high-glucose conditions and S424A mutant treatment weakened this binding (Fig. 3H). Furthermore, this treatment significantly reduced mitochondrial calcium concentrations (Fig. 3I) and oxidative damage (Fig. 3J), followed by a decrease in neuronal apoptosis (Fig. 3K). Bmal1 serine 424 is conserved in human, mice, rat, and other species (Appendix Fig. S3P). Collectively, these data suggested that high-glucose-induced Bhlhe41 upregulation was mediated by Bmal1 O-GlcNAcylation at its S424 site, which enhanced the binding of Clock to Bmal1 and activated Bhlhe41 transcription.

Since Bmal1/Clock heterodimer plays a major role in regulating the expression of a number of clock-controlled genes, such as Bhlhe40, period (per1, per2, per3) and cryptochrome (cry1, cry2), we therefore examined the expression of these genes under high-glucose conditions. scRNA-seq analysis revealed that all these gene expressions were significantly increased in the hippocampal neurons of db/db mice (Appendix Fig. S4A). In addition, we found

a significant increase in Bhlhe40, cry1, and per2 mRNA expression in mouse primary hippocampal neurons treated with high glucose (Appendix Fig. S4B). Interestingly, Chromatin immunoprecipitation (ChIP) assays further confirmed that Bhlhe41 significantly bound to the promoter of Bhlhe40, per1, per2, per3, cry1, and cry2 (Appendix Fig. S4C), and Bhlhe41 knockout further upregulated Bhlhe40 mRNA expression in the hippocampal CA1 neurons of diabetic mouse, whereas cry1 and per2 expression were not significantly affected (Appendix Fig. S4D).

## High glucose increases Dnajb4 expression by upregulating Bhlhe41 in hippocampal neurons

To investigate the downstream molecular mechanism of Bhlhe41 in promoting diabetes-associated cognitive dysfunction, we performed differential gene expression (DEGs) analysis of hippocampal neurons in scRNA-seq data and modeled regulatory network to scan the DEGs for downstream target genes of Bhlhe41. As shown in Fig. 4A, a total of 446 upregulated and 205 downregulated genes were identified in diabetes. Additionally, we identified 186 potential target genes of Bhlhe41, which were obtained by pySCENIC (Fig. 4B). Next, we compared 651 DEGs in hippocampal neurons between db/m and db/db mice to those 186 predicted genes and found that 24 genes overlapped between DEGs and predicted Bhlhe41 target genes (Fig. 4B–D). In particular, the difference in Dnajb4, Usp2, and Fosb expression between db/m and db/db remained statistically significant after adjustment (Fig. 4A). These three genes have been reported to be involved in chaperone binding, negative regulation of calcium ion transport and proapoptotic signaling (Gewies and Grimm, 2003; Weihl et al, 2023; Zhang et al, 2023a).

Next, we examined the effects of Bhlhe41 overexpression on Dnajb4, Usp2 and Fosb expression in mouse primary hippocampal neurons. As indicated in Fig. 4E and Appendix Fig. S5A, Bhlhe41 overexpression significantly upregulated Dnajb4 gene expression and its protein levels. ChIP assays further confirmed that Bhlhe41 significantly bound to the promoter of Dnajb4 (Fig. 4F). In addition, higher expression of Dnajb4 were also detected in hippocampal CA1 region of db/db and STZ-induced diabetic mice compared with their respective controls, and Bhlhe41 deletion significantly downregulated hippocampal Dnajb4 expression in db/db and STZ-induced diabetic mice (Fig. 4G; Appendix Fig. S5B). Similar results were also obtained using high glucose to treat mice primary hippocampal neurons in vitro, Bhlhe41 knockdown (Appendix Fig. S5C) significantly reversed high-glucose-induced

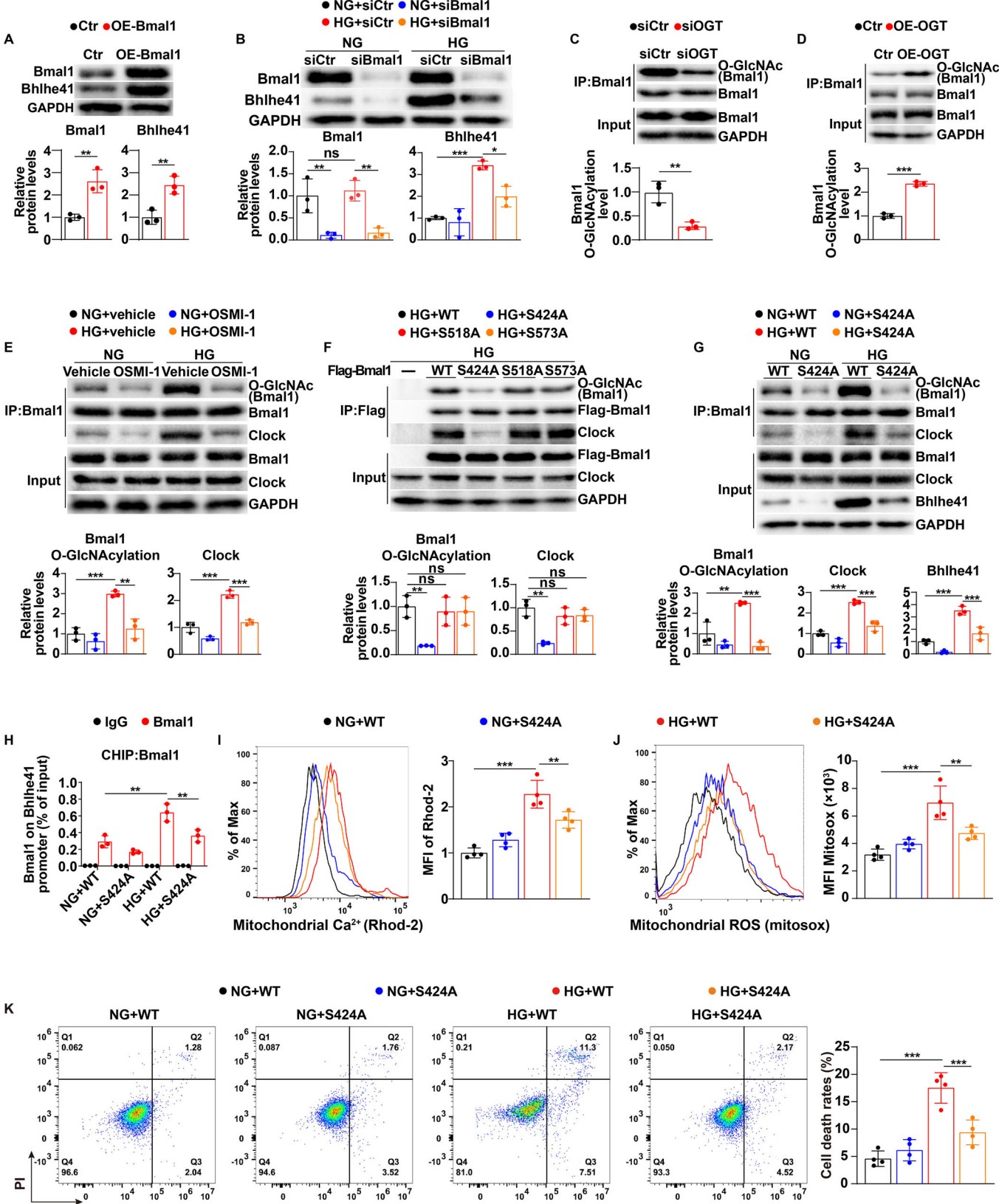

Figure 3.   Increasing O-GlcNAcylation of Bmal1 at S424 contributes to high-glucose-induced upregulation of Bhlhe41 expression in hippocampal neurons.

(A) Immunoblots and quantification analysis of protein levels of Bmal1 and Bhlhe41 in primary hippocampal neurons transfected with Bmal1 overexpression plasmid or control plasmid ($n = 3$ replicates). $P$ values: 0.0065 (Bmal1), 0.008 (Bhlhe41). (B) Immunoblots and quantification analysis of protein levels of Bmal1 and Bhlhe41 in primary hippocampal neurons treated with Bmal1 siRNA or control siRNA in the presence of normal glucose (NG, 5.5 mmol/L D-glucose) or high glucose (HG, 25 mmol/L D-glucose) ($n = 3$ replicates). $P$ values: 0.9244 (Bmal1, NG+siCtr vs HG+siCtr), 0.007 (Bmal1, NG+siCtr vs NG+siBmal1), 0.0046 (Bmal1, HG+siCtr vs HG+siBmal1); 0.0004 (Bhlhe41, NG+siCtr vs HG+siCtr), 0.0109 (Bmal1, HG+siCtr vs HG+siBmal1). (C) Primary hippocampal neurons were transfected with OGT siRNA or control siRNA, and Bmal1 O-GlcNAcylation was determined by immunoprecipitation and western blot using the indicated antibodies ($n = 3$ replicates). $P$ value: 0.0069. (D) Primary hippocampal neurons were transfected with OGT overexpression plasmid or control plasmid, and Bmal1 O-GlcNAcylation was determined by immunoprecipitation and western blot using the indicated antibodies ($n = 3$ replicates). $P$ value: <0.0001. (E) Primary hippocampal neurons were treated with normal glucose (NG, 5.5 mmol/L D-glucose) or high glucose (HG, 25 mmol/L D-glucose) in the presence of vehicle or OSMI-1 (20 μM), and Bmal1 O-GlcNAcylation was determined by immunoprecipitation and western blot using the indicated antibodies ($n = 3$ replicates). $P$ values: 0.0006 (Bmal1 O-GlcNAcylation, NG+Vehicle vs HG+Vehicle), 0.0015 (Bmal1 O-GlcNAcylation, HG+Vehicle vs HG + OSMI-1), <0.0001 (Clock, NG+Vehicle vs HG+Vehicle), <0.0001 (Clock, HG+Vehicle vs HG + OSMI-1). (F) HEK293T cells were transfected with the indicated plasmids, and Bmal1 O-GlcNAcylation was analyzed by immunoprecipitation with anti-FLAG antibody and western blot with the indicated antibodies. Three independent experiments were performed. $P$ values: 0.0086 (HG + WT vs HG + S424A), >0.9999 (HG + WT vs HG + S518A), >0.9999 (HG + WT vs HG + S573A). (G) Primary hippocampal neurons, which were stably silenced endogenous Bmal1 and expressed WT Bmal1 or S424A Bmal1, were treated with normal glucose (NG, 5.5 mmol/L D-glucose) or high glucose (HG, 25 mmol/L D-glucose). Cell lysates were immunoprecipitated with Bmal1 antibodies and western blotted with the indicated antibodies ($n = 3$ replicates). $P$ values: 0.0017 (Bmal1 O-GlcNAcylation, NG + WT vs HG + WT), 0.0002 (Bmal1 O-GlcNAcylation, HG + WT vs HG + S424A), <0.0001 (Clock, NG + WT vs HG + WT), 0.0002 (Clock, HG + WT vs HG + S424A), <0.0001 (Bhlhe41, NG + WT vs HG + WT), 0.0006 (Bhlhe41, HG + WT vs HG + S424A). (H) Binding of Bmal1 to the Bhlhe41 promoter in primary hippocampal neurons with endogenous Bmal1 stably silenced and expressing either wild-type (WT) Bmal1 or S424A Bmal1, treated with normal glucose (NG, 5.5 mmol/L D-glucose) or high glucose (HG, 25 mmol/L D-glucose). IgG was used as negative CHIP control ($n = 3$ replicates). $P$ values: <0.0001 (NG + WT vs HG + WT), 0.0061 (HG + WT vs HG + S424A). (I–K) Primary hippocampal neurons, which were stably silenced endogenous Bmal1 and expressed WT Bmal1 or S424A Bmal1, were treated with normal glucose (NG, 5.5 mmol/L D-glucose) or high glucose (HG, 25 mmol/L D-glucose), and mitochondrial calcium levels (I, $n = 4$ replicates), mtROS levels (J, $n = 4$ replicates) and apoptosis levels (K, $n = 4$ replicates) were analyzed by flow cytometry. $P$ values: <0.0001 (I, NG + WT vs HG + WT), 0.0083 (I, HG + WT vs HG + S424A), <0.0001 (J, NG + WT vs HG + WT), 0.0036 (J, HG + WT vs HG + S424A), <0.0001 (K, NG + WT vs HG + WT), 0.0009 (K, HG + WT vs HG + S424A). Data are means ± SEM. ns, not significant, *$P < 0.05$, **$P < 0.01$, ***$P < 0.001$. Two-tailed Student's unpaired $t$ test analysis (A, C, D), one-way ANOVA followed by Tukey's test (F), two-way ANOVA followed by Tukey's test (B, E, G, H–K). Source data are available online for this figure.

upregulation of Dnajb4 in hippocampal neurons (Fig. 4H,I), further confirming the important role of Bhlhe41 in upregulating Dnajb4 expression. Taken together, we concluded that Bhlhe41 upregulated Dnajb4 expression in hippocampal neurons under high-glucose conditions.

## Dnajb4-mediated NCLX ubiquitination and degradation induce mitochondrial calcium overload in hippocampal neurons and subsequently lead to mitochondrial dysfunction and neuronal damage

Mitochondrial calcium homeostasis is maintained in a coordinated fashion by inositol 1,4,5-trisphosphate receptors (IP3R) and NCLX. IP3R is responsible for triggering calcium influx from the endoplasmic reticulum to mitochondria, whereas NCLX is responsible for mediating sodium-dependent calcium efflux from mitochondria (Bartok et al, 2019; Garbincius and Elrod, 2022). A previous study by Hui et al has already proved that IP3R2 expression was upregulated in the hippocampus of db/db mouse and contributed to mitochondrial calcium overload, oxidative damage and cognitive impairment in diabetes. Interestingly, we also found a significant decrease in NCLX expression in hippocampal CA1 neurons of db/db and STZ-induced diabetic mice as compared to matched controls (Appendix Fig. S6A,B). To mechanistically link hippocampal NCLX downregulation with mitochondrial calcium overload and oxidative damage under high-glucose conditions, we specifically generated an AAV9 construct where NCLX was expressed using a CamKII promoter (AAV9-CamkII-NCLX). We next injected AAV9-CamkII-NCLX or an AAV9-CamkII control bilaterally into the hippocampal CA1 region of db/db and STZ-induced diabetic mice to specifically restore NCLX expression in hippocampal CA1 neurons in vivo (Fig. 5A,B; Appendix

Fig. S6C,D). As expected, NCLX overexpression significantly alleviated mitochondrial calcium overload in hippocampal CA1 neurons of db/db and STZ-induced diabetic mice (Fig. 5C; Appendix Fig. S6E), followed by restoration of mitochondrial function and reduced hippocampal neuron death (Fig. 5D,E; Appendix Fig. S6F,G). These effects were also associated with an improvement in cognitive function (Fig. 5F; Appendix Fig. S6H–J). Collectively, these findings suggested that NCLX overexpression significantly reversed high-glucose-induced mitochondrial calcium overload and oxidative damage in hippocampal CA1 neurons, thereby leading to an improvement in cognitive function in diabetes.

Dnajb4 is a member of the heat shock proteins 40 (HSP40) family that functions as a molecular chaperone to regulate protein synthesis, folding, transportation, and degradation (Howarth et al, 2007; Zhang et al, 2023b). To further determine whether Dnajb4 signaling regulates NCLX/IP3R2 expression, mitochondrial calcium homeostasis, and neuronal survival, mouse primary hippocampal neurons were infected with plasmid overexpressing Dnajb4 or containing control plasmid (Fig. 5G). We did not detect any significant change in IP3R2 protein levels upon overexpression of Dnajb4 (Fig. 5G). However, Dnajb4 overexpression significantly downregulated the protein levels of NCLX in hippocampal neurons (Fig. 5G), and this effect was accompanied by an increase in mitochondrial $Ca^{2+}$ concentrations (Fig. 5H), mtROS generation (Fig. 5I), apoptosis (Fig. 5J) and a decrease in MMP levels (Appendix Fig. S6K). HSP40 family has been shown to interact with target proteins and promote their proteasomal degradation through ubiquitination (Howarth et al, 2007). Consistently, we performed the reciprocal Co-IP assays and verified that Dnajb4 bound to NCLX in hippocampal neurons (Appendix Fig. S6L). In addition, Dnajb4 overexpression significantly increased NCLX ubiquitination

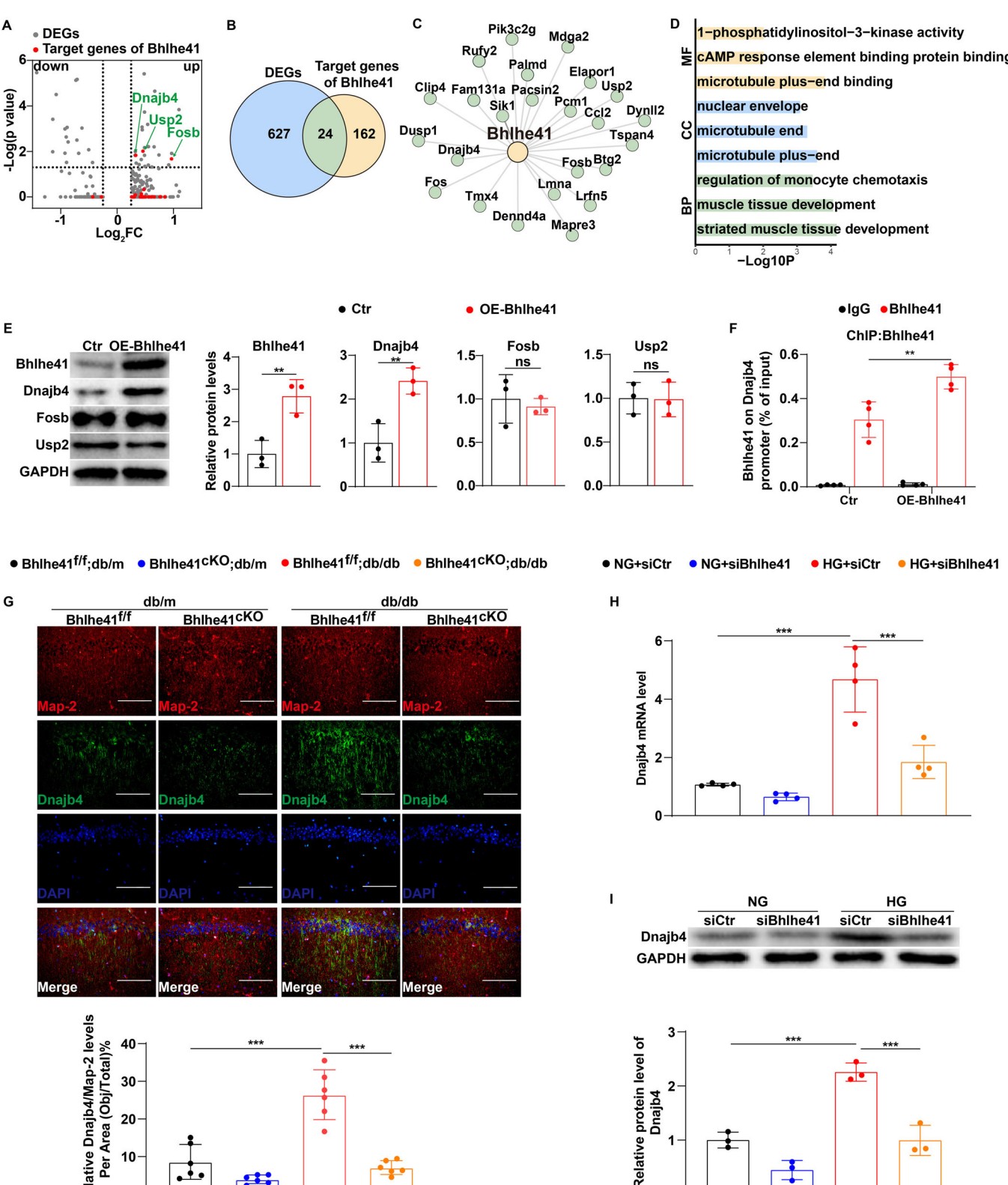

and decreased its protein level in primary hippocampal neurons (Fig. 5K). More importantly, proteasome inhibitor MG132 treatment significantly reversed Dnajb4 overexpression-induced NCLX downregulation (Fig. 5L). Consistently, Bhlhe41 deletion in hippocampal CA1 neurons successfully upregulated the protein levels of NCLX in db/db and STZ-induced diabetic mice (Fig. 5M; Appendix Fig. S6M). The above findings suggested that Bhlhe41/Dnajb4 signaling induced mitochondrial calcium overload and oxidative damage in hippocampal neurons by promoting NCLX ubiquitination and its degradation.

**Figure 4.   High glucose increases Dnajb4 expression by upregulating Bhlhe41 in hippocampal neurons.**

(A) The volcano plot indicated differentially expressed genes. The red dots represented target genes of Bhlhe41, and the gray dots represented differential genes. (B) Venn diagram of the overlap between predicted Bhlhe41 target genes obtained by pySCENIC and DEGs in hippocampal neurons identified by scRNA-seq. (C) Bhlhe41 gene regulatory network was inferred using pySCENIC. (D) GO analysis of DEGs modulated by Bhlhe41. MF molecular function, CC cellular component, BP biological process. (E) Immunoblots and quantification analysis of protein levels of Bhlhe41, Dnajb4, Usp2, Fosb (F, $n = 3$ blots in total) in primary hippocampal neurons transfected with control plasmid or a plasmid overexpressing Bhlhe41 for 48 h. P values: 0.0097 (Bhlhe41), 0.0099 (Dnajb4), 0.6292 (Fosb), 0.9357 (Usp2). (F) Binding of Bhlhe41 to the Dnajb4 promoter in primary hippocampal neurons transfected with control plasmid or a plasmid overexpressing Bhlhe41 for 48 h. IgG was used as negative CHIP control. ($n = 4$ replicates). P value: 0.0075. (G) Representative images and quantifications showing IF staining of Dnajb4 (green) and MAP-2 (red) in the hippocampal CA1 neurons of db/m Bhlhe41^fl/fl or db/db Bhlhe41^fl/fl mice injected with AAV-CAMKII or AAV-CAMKII-Cre ($n = 6$ mice per group), Scale. bar: 100 μm. P values: <0.0001 (Bhlhe41^f/f;db/m vs Bhlhe41^f/f;db/db), <0.0001 (Bhlhe41^f/f;db/db vs Bhlhe41^cKO;db/db). (H, I) mRNA levels (H, $n = 4$ replicates) and immunoblots analysis of protein levels (I, $n = 3$ replicates) of Dnajb4 in primary hippocampal neurons treated with control siRNA or Bhlhe41 siRNA in the presence of normal glucose (NG, 5.5 mmol/L D-glucose) or high glucose (HG, 25 mmol/L D-glucose). P values: <0.0001 (H, NG+siCtr vs HG+siCtr), <0.0001 (H, HG+siCtr vs HG+siBhlhe41), 0.0003 (I, NG+siCtr vs HG+siCtr), 0.0002 (I, HG+siCtr vs HG+siBhlhe41). Data are means ± SEM. ns, not significant, **P < 0.01, ***P < 0.001. Hypergeometric test (D), two-tailed Student's unpaired t test analysis (A, E), two-way ANOVA followed by Tukey's test (F–I). Source data are available online for this figure.

## Short peptide S424-pe protects against high-glucose-induced neuronal apoptosis and cognitive dysfunction by regulating Bhlhe41/Dnajb4/NCLX signaling

To provide direct evidence that O-GlcNAcylation of Bmal1 at S424 impairs cognitive function through the Bhlhe41/Dnajb4/NCLX signaling pathway, we developed a short peptide designed to specifically target the O-GlcNAcylation of Bmal1 at S424. After analyzing the sequence surrounding the Bmal1 S424 site, we synthesized five short peptides fused with a cell-penetrating peptide, rendering them capable of permeating cell membranes and crossing the blood-brain barrier (Qin et al, 2019). This approach has emerged as one of the most popular and effective methods for intracellular delivery of biomolecules. (Koren et al, 2011; Li et al, 2021) (Fig. 6A). Among these peptides, S424-peptide-2# (referred to as S424-pe) exhibited the strongest inhibitory effect on Bmal1 O-GlcNAcylation. (Fig. 6B). Thus, based on the S424-pe peptide, we generated a scrambled peptide denoted as S424A-pe, in which the S424 amino acid was replaced with an A residue. As shown in Fig. 6C,D, S424-pe, but not S424A-pe, significantly reversed high-glucose-induced Bmal1 O-GlcNAcylation in hippocampal CA1 neurons isolated from db/db (Fig. 6C) and STZ-induced diabetic mice (Fig. 6D), accompanied by a decrease in binding of Clock to Bmal1 and a downregulation of Bhlhe41 and Dnajb4 expression (Fig. 6E,F). Furthermore, S424-pe significantly upregulated NCLX expression (Fig. 6E,F) and reduced mitochondrial calcium concentrations (Fig. 6I,K) and oxidative damage (Fig. 6J,L), followed by a decrease in neuronal apoptosis (Fig. 6G,H) and an improvement in cognitive function (Fig. 7A–D). Similar results were also obtained in mouse primary hippocampal neurons treated with high glucose (Appendix Fig. S7A–E). Collectively, our results demonstrated that targeting Bmal1 S424 O-GlcNAcylation with S424-pe may alleviate high-glucose-induced mitochondrial calcium overload and oxidative damage in hippocampal CA1 neurons and thereby improve cognitive function in diabetes.

## Discussion

In the current study, we revealed a mechanism underlying high-glucose-induced hippocampal neuronal damage and cognitive dysfunction in diabetes. The key findings of this work were summarized as follows: (1) high-glucose-induced hippocampal neuron damage by downregulating NCLX expression and subsequently promoting mitochondrial calcium overload; (2) high-glucose-mediated Bmal1 O-GlcNAcylation upregulated Bhlhe41 and Dnajb4 expression and these effects promoted NCLX ubiquitination and its degradation in hippocampal neurons; (3) short peptide S424-pe protected against high-glucose-induced neuronal apoptosis and cognitive dysfunction through inhibiting Bmal1 S424 O-GlcNAcylation (Fig. 8).

In this study, pySCENIC analysis of hippocampal neurons in scRNA-seq data revealed that Bhlhe41 upregulation was involved in the pathogenesis of diabetes-associated cognitive. Bhlhe41 is a basic helix–loop–helix transcription factor that regulates circadian rhythms, including daily sleep/wake rhythms, feeding behavior, and hormone secretions (Hirano et al, 2018). Previous researchers reported that Bhlhe41 expression was increased by glucose supplementation but decreased by glucose depletion (Sato et al, 2018). Consistently, we also found an increase in Bhlhe41 expression in hippocampal neurons under high-glucose conditions, and this increase led to cognitive impairment in both type 1 and type 2 diabetic mice. Based upon previous and present findings, we suggested that aberrant Bhlhe41 expression not only disrupted circadian rhythms but also induced cognitive impairment in diabetes.

We additionally investigated the up- and downstream regulatory mechanisms associated with the upregulation of Bhlhe41 expression in diabetes. Our data showed a significant role of Bmal1-Clock complex in upregulating Bhlhe41 expression in hippocampal neurons, which was consistent with previous reports (Kato et al, 2014). In addition, Li et al reported that O-GlcNAcylation stabilized Bmal1/Clock by inhibiting their ubiquitination and subsequent degradation and thereby increased Bmal1/Clock-mediated transcription of target genes (Li et al, 2013). Our results extended those findings and further demonstrated that O-GlcNAcylation of Bmal1 at S424 increased the binding of Clock to Bmal1 under high-glucose conditions, resulting in increased Bhlhe41 expression in hippocampal neurons. O-GlcNAcylation is a posttranslational, bidirectional, dynamic modification of serine and threonine residues linked to glucose metabolism and centrally involved in regulating cellular homeostasis (Chatham et al, 2021). Abnormal O-GlcNAcylation has been recognized as a general mechanism underlying diabetic complications (Chatham et al, 2021). This modulation of Bmal1 function through O-GlcNAcylation of Bmal1 at S424 links glucose and Bhlhe41

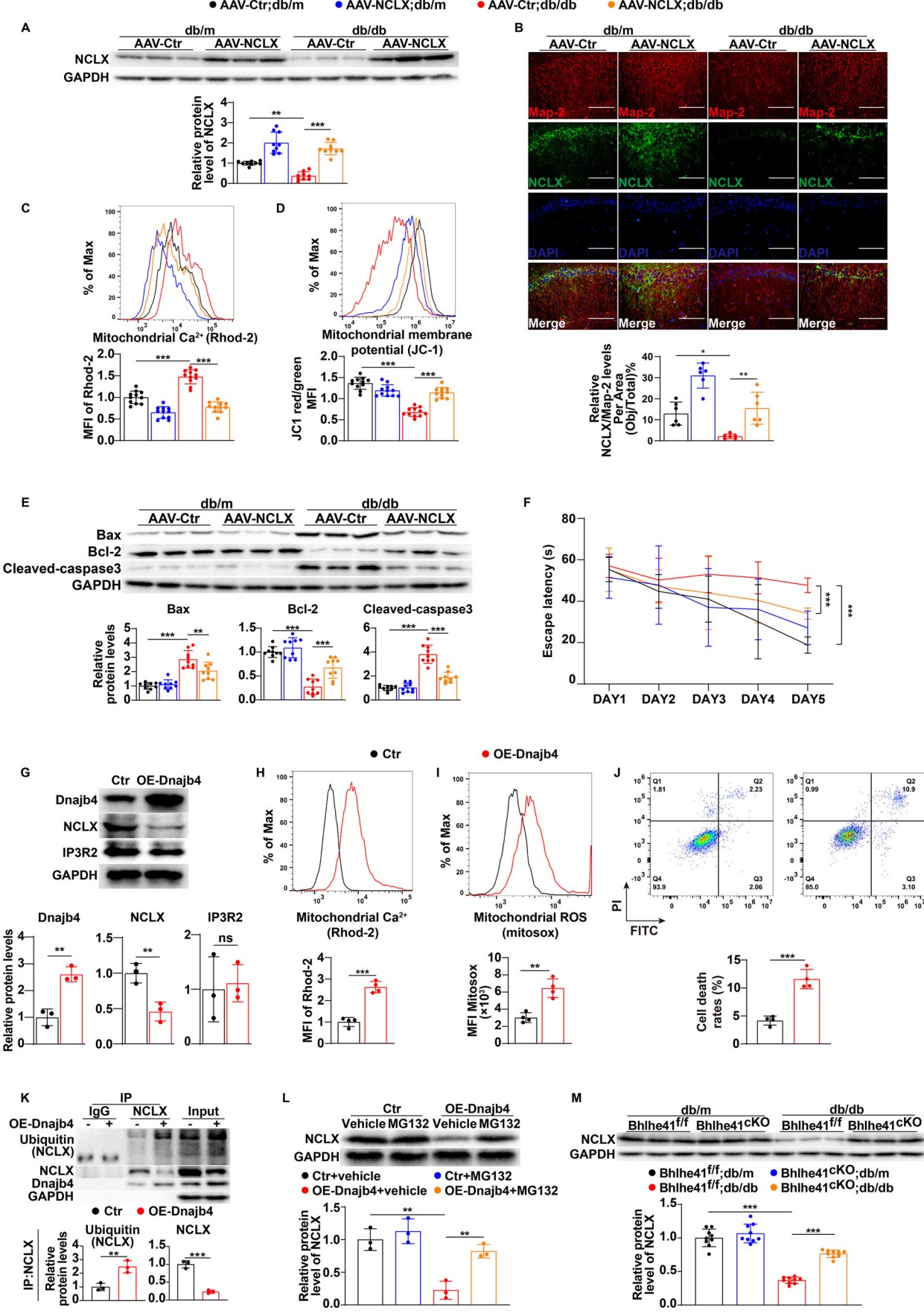

**Figure 5.   Dnajb4-mediated NCLX ubiquitination and degradation induce mitochondrial calcium overload in hippocampal neurons and subsequently lead to mitochondrial dysfunction and neuronal damage.**

(A) Immunoblots and quantification analysis of protein level of NCLX in the hippocampal CA1 neurons of db/m or db/db mice injected with AAV-CAMKII or AAV-CAMKII-NCLX ($n = 9$ mice per group). P values: 0.0013 (AAV-Ctr;db/m vs AAV-Ctr;db/db), <0.0001 (AAV-Ctr;db/db vs AAV-NCLX;db/db). (B) Representative images and quantifications showing IF staining of NCLX (green) and MAP-2 (red) in the hippocampal CA1 neurons of db/m or db/db mice injected with AAV-CAMKII or AAV-CAMKII-NCLX ($n = 6$ mice per group). Scale bar: 100 μm. P values: 0.016 (AAV-Ctr;db/m vs AAV-Ctr;db/db), 0.0025 (AAV-Ctr;db/db vs AAV-NCLX;db/db). (C, D) Flow cytometry analysis of mitochondrial calcium levels (C, $n = 11$ mice per group), MMP (D, $n = 11$ mice per group) in the hippocampal CA1 neurons of db/m or db/db mice injected with AAV-CAMKII or AAV-CAMKII-NCLX. P values: <0.0001 (C, AAV-Ctr;db/m vs AAV-Ctr;db/db), <0.0001 (C, AAV-Ctr;db/db vs AAV-NCLX;db/db), <0.0001 (D, AAV-Ctr;db/m vs AAV-Ctr;db/db), <0.0001 (D, AAV-Ctr;db/db vs AAV-NCLX;db/db). (E) Immunoblots and quantification analysis of protein levels of Bax, Bcl-2, and Cleaved-caspase3 in the hippocampal CA1 neurons of db/m or db/db mice injected with AAV-CAMKII or AAV-CAMKII-NCLX ($n = 6$ mice per group). P values: <0.0001 (Bax, AAV-Ctr;db/m vs AAV-Ctr;db/db), 0.0056 (Bax, AAV-Ctr;db/db vs AAV-NCLX;db/db), <0.0001 (Bcl-2, AAV-Ctr;db/m vs AAV-Ctr;db/db), 0.0003 (Bcl-2, AAV-Ctr;db/db vs AAV-NCLX;db/db), <0.0001 (Cleaved-caspase3, AAV-Ctr;db/m vs AAV-Ctr;db/db), <0.0001 (Cleaved-caspase3, AAV-Ctr;db/db vs AAV-NCLX;db/db). (F) Escape latency to the platform during the training trial in a Morris water maze of db/m or db/db mice injected with AAV-CAMKII or AAV-CAMKII-NCLX ($n = 6$ mice per group). P values: <0.0001 (AAV-Ctr;db/m vs AAV-Ctr;db/db), 0.0007 (AAV-Ctr;db/db vs AAV-NCLX;db/db). (G) Immunoblots and quantification analysis of protein levels of Dnajb4, NCLX, IP3R2 in primary hippocampal neurons transfected with Dnajb4 overexpression plasmid or control plasmid ($n = 3$ replicates). P values: 0.0026 (Dnajb4), 0.0082 (NCLX), 0.7908 (IP3R2). (H–J) Flow cytometry analysis of mitochondrial calcium levels (H), mtROS levels (I), and apoptosis levels (J) in primary hippocampal neurons transfected with Dnajb4 overexpression plasmid or control plasmid ($n = 4$ replicates). P values: <0.0001 (H), 0.0014 (I), 0.0002 (J). (K) Results of immunoprecipitation experiments in primary hippocampal neurons treated with Dnajb4 overexpression plasmid or control plasmid. Anti-NCLX immunoprecipitation was followed by western blotting for ubiquitin to determine levels of ubiquitin NCLX. The protein levels of NCLX and Dnajb4 in the precipitates were determined with protein-specific primary antibodies ($n = 3$ replicates). P values: 0.0079 (Ubiquitin NCLX), 0.0003 (NCLX). (L) Immunoblots and quantification analysis of protein level of NCLX in the primary hippocampal neurons treated with Dnajb4 overexpression plasmid or control plasmid in the presence of vehicle or MG132 ($n = 3$ replicates). P values: 0.0012 (Ctr+vehicle vs OE-Dnajb4+vehicle), 0.006 (OE-Dnajb4+vehicle vs OE-Dnajb4 + MG132). (M) Immunoblots and quantification analysis of protein level of NCLX in the hippocampal CA1 neurons of db/m Bhlhe41$^{fl/fl}$ or db/db Bhlhe41$^{fl/fl}$ mice injected with AAV-CAMKII or AAV-CAMKII-Cre ($n = 9$ mice per group). P values: <0.0001 (Bhlhe41$^{f/f}$;db/m vs Bhlhe41$^{f/f}$;db/db), <0.0001 (Bhlhe41$^{f/f}$;db/db vs Bhlhe41$^{cKO}$;db/db). Data are means ± SEM. ns, not significant, *$P < 0.05$, **$P < 0.01$, ***$P < 0.001$. Two-tailed Student's unpaired $t$ test analysis (G, K), two-way ANOVA followed by Tukey's test (A–F, L, M). Source data are available online for this figure.

upregulation in hippocampal neurons, implicating Bmal1 as a potential therapeutic target for the treatment of cognitive impairment in diabetes.

Bmal1/Clock complex has been reported to regulate the expression of a number of clock-controlled genes (Fagiani et al, 2022). In addition to Bhlhe41, our in vitro results showed that Bhlhe40, cry1, and per2 mRNA expression were also upregulated in hippocampal neurons under high-glucose conditions, whereas the expression of cry2, per1, and per3 were not significantly affected. This result might be partly attributed to different degrees of repression on Bmal1/Clock complex-mediated transactivation exerted by clock-controlled genes themselves. Previous studies have found that Bhlhe, Per, and Cry genes are activated by Bmal1/Clock complex and repressed by their own products through CACGTG E-boxes in their promoter regions (Honma et al, 2002; Kawamoto et al, 2004). All these gene products have been proven to function as transcriptional repressors by binding to the Bmal1/Clock complex. However, only Bhlhe40 and Bhlhe41 have the DNA-binding capacity and can directly compete with Bmal1/Clock complex for CACGTG E-box binding in promoter regions (Hamaguchi et al, 2004). The differential binding ability of Bhlhe40 and Bhlhe41 to the promoter regions of different clock-controlled genes and their inhibitory effects on expression remain to be elucidated in future studies. Consistent with previous reports (Kreslavsky et al, 2017), our data revealed significant recruitment of Bhlhe41 to Bhlhe40, per1, per2, per3, cry1, and cry2 promoter. More interestingly, diabetic mice lacking Bhlhe41 expression in hippocampal CA1 neurons exhibited upregulation of Bhlhe40 and showed no significant changes in the expression patterns of cry1 and per2, similar results were also obtained in a study examining the effects of Bhlhe40 deletion on Bhlhe41 expression (Grechez-Cassiau et al, 2004), indicating that Bhlhe41 can compensate for the function of Bhlhe40 and vice versa. Notably, previous studies have already demonstrated that both type 1 and type 2 diabetic patients

were highly susceptible to disrupted circadian rhythms, such as flattened circadian cortisol profile, earlier dim-light melatonin onset and sleep disorders (Peng et al, 2022; Reutrakul et al, 2016). Our findings also provided a potential mechanism for this phenomenon.

Dnajb4, known as an enhancer of apoptosis (Lei et al, 2011), was found to be upregulated by Bhlhe41 in our study. Additionally, Dnajb4 upregulation led to NCLX ubiquitination and degradation in hippocampal neurons, which induced mitochondrial calcium overload and impaired mitochondrial function, thereby promoting neuronal apoptosis and cognitive impairment in diabetes. Mitochondrial calcium homeostasis has been shown to play a major role in maintaining mitochondrial function (Garbincius and Elrod, 2022; Pathak and Trebak, 2018). Disruptions of mitochondrial calcium homeostasis induced by IP3R upregulation or NCLX downregulation have been proven to promote mitochondrial dysfunction and oxidative damage in multiple cell types (Pathak and Trebak, 2018). Furthermore, we demonstrated that NCLX overexpression significantly reversed high-glucose-induced mitochondrial calcium overload and oxidative damage in hippocampal neurons, accompanied by an improvement in cognitive function. These findings suggested that targeting Dnajb4-mediated NCLX ubiquitination and degradation was a promising therapeutic strategy for cognitive impairment in diabetes.

Small-molecule inhibitors targeting OGT have been created to explore the pathogenic role of O-GlcNAcylation modifications in a variety of diseases (Rohlff et al, 1999; Sharma et al, 2019). However, as OGT catalyzes O-GlcNAcylation on a wide range of protein substrates and plays a crucial role in maintaining normal physiological functions, the inhibition of OGT is likely to result in adverse side effects both in vivo and in vitro (Zhu et al, 2023). To overcome this limitation, we designed an alternative strategy that employs a short peptide containing Bmal1 O-GlcNAcylation site to competitively inhibit the O-GlcNAcylation of Bmal1 S424 with

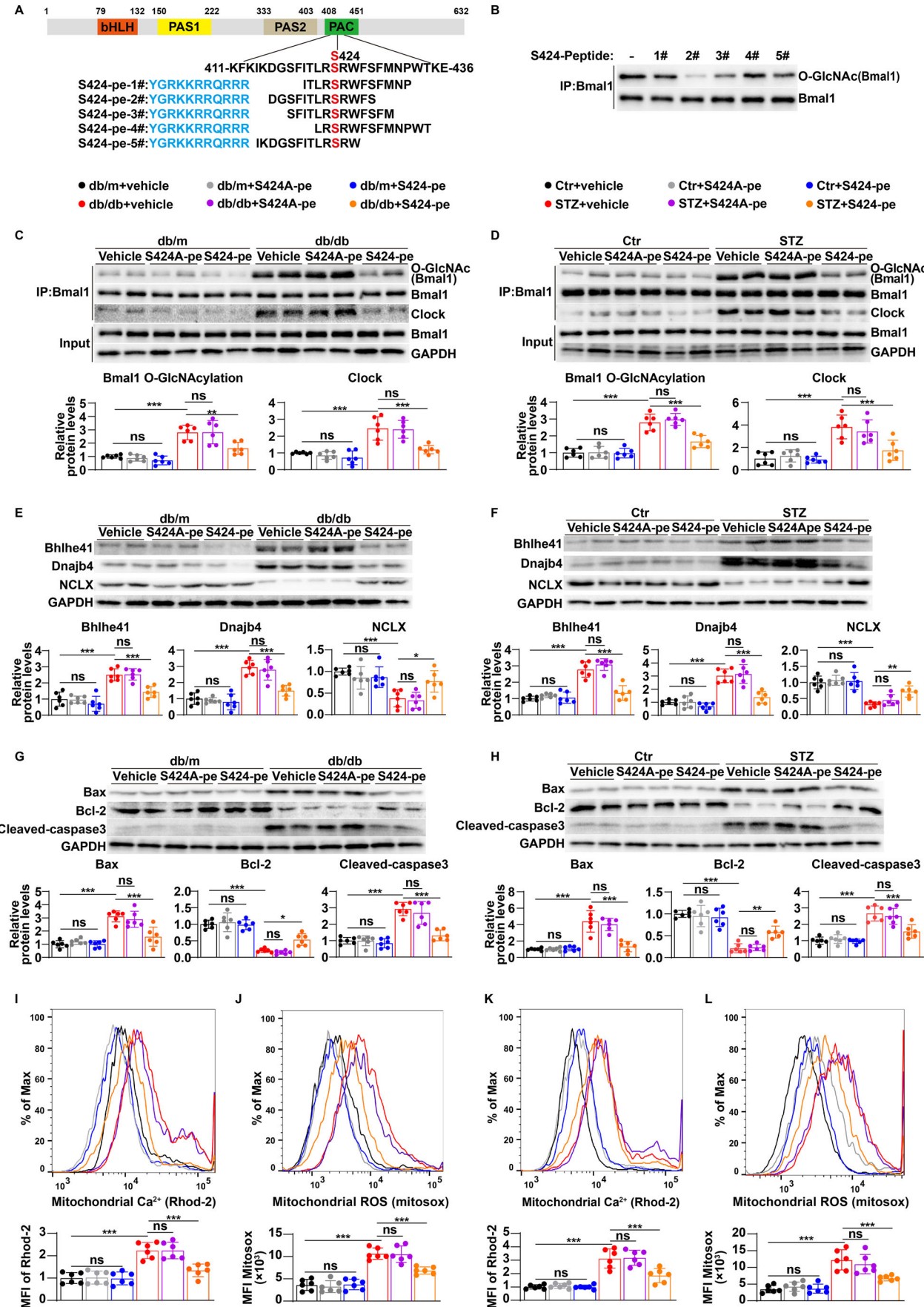

**Figure 6. Short peptide S424-pe protects against high-glucose-induced neuronal apoptosis by regulating Bhlhe41/Dnajb4/NCLX signaling.**

(A) Schematic illustration of designed peptides. Blue: cell-penetrating peptide (CPP). Red: site for Bmal1 O-GlcNAcylation. (B) Primary hippocampal neurons were treated with five short peptides (20 µM) in the presence of high glucose (HG, 25 mmol/L D-glucose), and Bmal1 O-GlcNAcylation was determined by immunoprecipitation and western blot using the indicated antibodies. Three independent experiments were performed. (C, D) Hippocampal CA1 neurons were isolated from db/m, db/db (C), Ctr or STZ mice (D) treated with vehicle, S424A-pe (5 mg/kg) or S424-pe (5 mg/kg), and cell lysates were immunoprecipitated with Bmal1 antibodies and western blotted with the indicated antibodies (n = 6 mice per group). P values: 0.9986 (C, Bmal1 O-GlcNAcylation, db/m+vehicle vs db/m + S424A-pe), 0.8977 (C, Bmal1 O-GlcNAcylation, db/m+vehicle vs db/m + S424-pe), <0.0001 (C, Bmal1 O-GlcNAcylation, db/m+vehicle vs db/db+vehicle), >0.9999 (C, Bmal1 O-GlcNAcylation, db/db +vehicle vs db/db+S424A-pe), 0.0028 (C, Bmal1 O-GlcNAcylation, db/db+vehicle vs db/db+S424-pe); 0.9899 (C, Clock, db/m+vehicle vs db/m + S424A-pe), 0.8427 (C, Clock, db/m+vehicle vs db/m + S424-pe), <0.0001 (C, Clock, db/m+vehicle vs db/db+vehicle), >0.9999 (C, Clock, db/db+vehicle vs db/db+S424A-pe), 0.0002 (C, Clock, db/db+vehicle vs db/db+S424-pe); >0.9999 (D, Bmal1 O-GlcNAcylation, Ctr+vehicle vs Ctr+S424A-pe), >0.9999 (D, Bmal1 O-GlcNAcylation, Ctr+vehicle vs Ctr+S424-pe), <0.0001 (D, Bmal1 O-GlcNAcylation, Ctr+vehicle vs STZ+vehicle), 0.9551 (D, Bmal1 O-GlcNAcylation, STZ+vehicle vs STZ + S424A-pe), <0.0001 (D, Bmal1 O-GlcNAcylation, STZ+vehicle vs STZ + S424-pe); 0.9944 (D, Clock, Ctr+vehicle vs Ctr+S424A-pe), >0.9999 (D, Clock, Ctr+vehicle vs Ctr+S424-pe), <0.0001 (D, Clock, Ctr+vehicle vs STZ+vehicle), 0.9467 (D, Clock, STZ+vehicle vs STZ + S424A-pe), 0.001 (D, Clock, STZ+vehicle vs STZ + S424-pe). (E–H) Immunoblots and quantification analysis of Bhlhe41, Dnajb4, NCLX (E, F), Bax, Bcl-2 and Cleaved-caspase3 (G, H) expression levels in the hippocampal CA1 neurons of db/m, db/db, Ctr or STZ mice treated with vehicle, S424A-pe (5 mg/kg) or S424-pe (5 mg/kg) (n = 6 mice per group). P values: >0.9999 (E, Bhlhe41, db/m+vehicle vs db/m + S424A-pe), 0.7581 (E, Bhlhe41, db/m+vehicle vs db/m + S424-pe), <0.0001 (E, Bhlhe41, db/m+vehicle vs db/db+vehicle), >0.9999 (E, Bhlhe41, db/db+vehicle vs db/db+S424A-pe), 0.0004 (E, Bhlhe41, db/db+vehicle vs db/db+S424-pe); 0.9978 (E, Dnajb4, db/m+vehicle vs db/m + S424A-pe), 0.9604 (E, Dnajb4, db/m+vehicle vs db/m + S424-pe), <0.0001 (E, Dnajb4, db/m+vehicle vs db/db+vehicle), 0.9868 (E, Dnajb4, db/db+vehicle vs db/db+S424A-pe), <0.0001 (E, Dnajb4, db/db+vehicle vs db/db+S424-pe); 0.8301 (E, NCLX, db/m+vehicle vs db/m + S424A-pe), 0.8815 (E, NCLX, db/m+vehicle vs db/m + S424-pe), 0.0002 (E, NCLX, db/m+vehicle vs db/db+vehicle), 0.9986 (E, NCLX, db/db+vehicle vs db/db+S424A-pe), 0.0312 (E, NCLX, db/db+vehicle vs db/db+S424-pe); 0.9421 (F, Bhlhe41, Ctr+vehicle vs Ctr+S424A-pe), 0.9997 (F, Bhlhe41, Ctr+vehicle vs Ctr+S424-pe), <0.0001 (F, Bhlhe41, Ctr+vehicle vs STZ+vehicle), 0.6463 (F, Bhlhe41, STZ+vehicle vs STZ + S424A-pe), <0.0001 (F, Bhlhe41, STZ+vehicle vs STZ + S424-pe); >0.9999 (F, Dnajb4, Ctr+vehicle vs Ctr+S424A-pe), 0.8474 (F, Dnajb4, Ctr+vehicle vs Ctr+S424-pe), <0.0001 (F, Dnajb4, Ctr+vehicle vs STZ+vehicle), 0.996 (F, Dnajb4, STZ+vehicle vs STZ + S424A-pe), <0.0001 (F, Dnajb4, STZ+vehicle vs STZ + S424-pe); 0.9862 (F, NCLX, Ctr+vehicle vs Ctr+S424A-pe), 0.9987 (F, NCLX, Ctr+vehicle vs Ctr+S424-pe), <0.0001 (F, NCLX, Ctr+vehicle vs STZ+vehicle), 0.8104 (F, NCLX, STZ+vehicle vs STZ + S424A-pe), 0.0078 (F, NCLX, STZ+vehicle vs STZ + S424-pe); 0.9997 (G, Bax, db/m+vehicle vs db/m + S424A-pe), >0.9999 (G, Bax, db/m+vehicle vs db/m + S424-pe), <0.0001 (G, Bax, db/m+vehicle vs db/db+vehicle), 0.999 (G, Bax, db/db+vehicle vs db/db+S424A-pe), <0.0001 (G, Bax, db/db+vehicle vs db/db+S424-pe); 0.9843 (G, Bcl-2, db/m+vehicle vs db/m + S424A-pe), >0.9999 (G, Bcl-2, db/m+vehicle vs db/m + S424-pe), <0.0001 (G, Bcl-2, db/m+vehicle vs db/db+vehicle), 0.9852 (G, Bcl-2, db/db+vehicle vs db/db+S424A-pe), 0.0163 (G, Bcl-2, db/db+vehicle vs db/db+S424-pe); >0.9999 (G, Cleaved-caspase3, db/m+vehicle vs db/m + S424A-pe), 0.995 (G, Cleaved-caspase3, db/m+vehicle vs db/m + S424-pe), <0.0001 (G, Cleaved-caspase3, db/m+vehicle vs db/db+vehicle), 0.9106 (G, Cleaved-caspase3, db/db+vehicle vs db/db+S424A-pe), <0.0001 (G, Cleaved-caspase3, db/db+vehicle vs db/db+S424-pe); >0.9999 (H, Bax, Ctr+vehicle vs Ctr+S424A-pe), >0.9999 (H, Bax, Ctr+vehicle vs Ctr+S424-pe), <0.0001 (H, Bax, Ctr+vehicle vs STZ+vehicle), 0.9246 (H, Bax, STZ+vehicle vs STZ + S424A-pe), <0.0001 (H, Bax, STZ+vehicle vs STZ + S424-pe); 0.9925 (H, Bcl-2, Ctr+vehicle vs Ctr+S424A-pe), 0.9483 (H, Bcl-2, Ctr+vehicle vs Ctr+S424-pe), <0.0001 (H, Bcl-2, Ctr+vehicle vs STZ+vehicle), >0.9999 (H, Bcl-2, STZ+vehicle vs STZ + S424A-pe), 0.0032 (H, Bcl-2, STZ+vehicle vs STZ + S424-pe); 0.9965 (H, Cleaved-caspase3, Ctr+vehicle vs Ctr+S424A-pe), >0.9999 (H, Cleaved-caspase3, Ctr+vehicle vs Ctr+S424-pe), <0.0001 (H, Cleaved-caspase3, Ctr+vehicle vs STZ+vehicle), 0.9577 (H, Cleaved-caspase3, STZ+vehicle vs STZ + S424A-pe), 0.0001 (H, Cleaved-caspase3, STZ+vehicle vs STZ + S424-pe). (I–L) Flow cytometry analysis of mitochondrial calcium levels (I, K) and mtROS levels (J, L) in the hippocampal CA1 neurons isolated from different groups of mice (n = 6 mice per group). P values: >0.9999 (I, db/m+vehicle vs db/m + S424A-pe), >0.9999 (I, db/m+vehicle vs db/m + S424-pe), <0.0001 (I, db/m+vehicle vs db/db+vehicle), >0.9999 (I, db/db+vehicle vs db/db+S424A-pe), 0.0003 (I, db/db+vehicle vs db/db+S424-pe); 0.9963 (J, db/m+vehicle vs db/m + S424A-pe), >0.9999 (J, db/m+vehicle vs db/m + S424-pe), <0.0001 (J, db/m+vehicle vs db/db+vehicle), >0.9999 (J, db/db+vehicle vs db/db+S424A-pe), 0.0003 (J, db/db+vehicle vs db/db+S424-pe); 0.9995 (K, Ctr+vehicle vs Ctr+S424A-pe), >0.9999 (K, Ctr+vehicle vs Ctr+S424-pe), <0.0001 (K, Ctr+vehicle vs STZ+vehicle), >0.9999 (K, STZ+vehicle vs STZ + S424A-pe), 0.0003 (K, STZ+vehicle vs STZ + S424-pe); 0.9946 (L, Ctr+vehicle vs Ctr+S424A-pe), >0.9999 (L, Ctr+vehicle vs Ctr+S424-pe), <0.0001 (L, Ctr+vehicle vs STZ+vehicle), 0.8721 (L, STZ+vehicle vs STZ + S424A-pe), 0.0005 (L, STZ+vehicle vs STZ + S424-pe). Data are means ± SEM. ns not significant, *P < 0.05, **P < 0.01, ***P < 0.001. Two-way ANOVA followed by Tukey's test (C–L). Source data are available online for this figure.

minimal interference with the total cellular O-GlcNAcylation level. Such treatment led to a decrease in binding of Clock to Bmal1 and a downregulation of Bhlhe41 and Dnajb4 expression, resulting in NCLX upregulation and an improvement in mitochondrial calcium overload, hippocampal neuron dysfunction, and cognitive impairment in diabetes. Our data suggest that oscillations in mitochondrial calcium accumulation and oxidative damage in hippocampal neurons are indeed affected by O-GlcNAcylation of Bmal1, in addition, it provides solid evidence that Bma1l O-GlcNAcylation directly impairs cognitive function in diabetes. Short peptide has been used in a recent study to elucidate the role of O-GlcNAcylation modification in tumor immune evasion (Zhu et al, 2023). Our study further confirmed that this strategy targeting site-specific O-GlcNAcylation on Bmal1 in hippocampal neurons may serve as a novel therapeutic approach for the treatment of cognitive impairment in diabetes.

In summary, we report a mechanistic link between high-glucose-induced Bmal1 O-GlcNAcylation and the Bhlhe41/Dnajb4/NCLX signaling pathway in the control of mitochondrial calcium concentrations in hippocampal neurons. Our study also highlights the crucial role of O-GlcNAcylation in disrupting clock gene expression under high-glucose conditions and provides insight into the mechanism that underlies the neuroprotective properties of short peptides in diabetes.

## Limitation of the study

The lack of autopsy data is the first limitation of our study as postmortem analysis would be helpful to further elucidate the mechanism underlying high-glucose-induced cognitive impairment. Second, O-GlcNAcylation has been already reported to regulate circadian clock activity (Kaasik et al, 2013; Li et al, 2019). In addition to affecting Bmal1, as also previously reported by Ma and colleagues (Ma et al, 2013), O-GlcNAcylation of additional clock proteins, such as Per2 and Clock, has been reported to have a role in regulating clock gene expression in diverse tissues, further investigation is warranted to determine if these proteins are glycosylated in hippocampal neurons and if they play a pathogenetic role in cognitive impairment in diabetes. Third, Bmal1 has also been shown to be a translation factor (Lipton et al, 2015), such

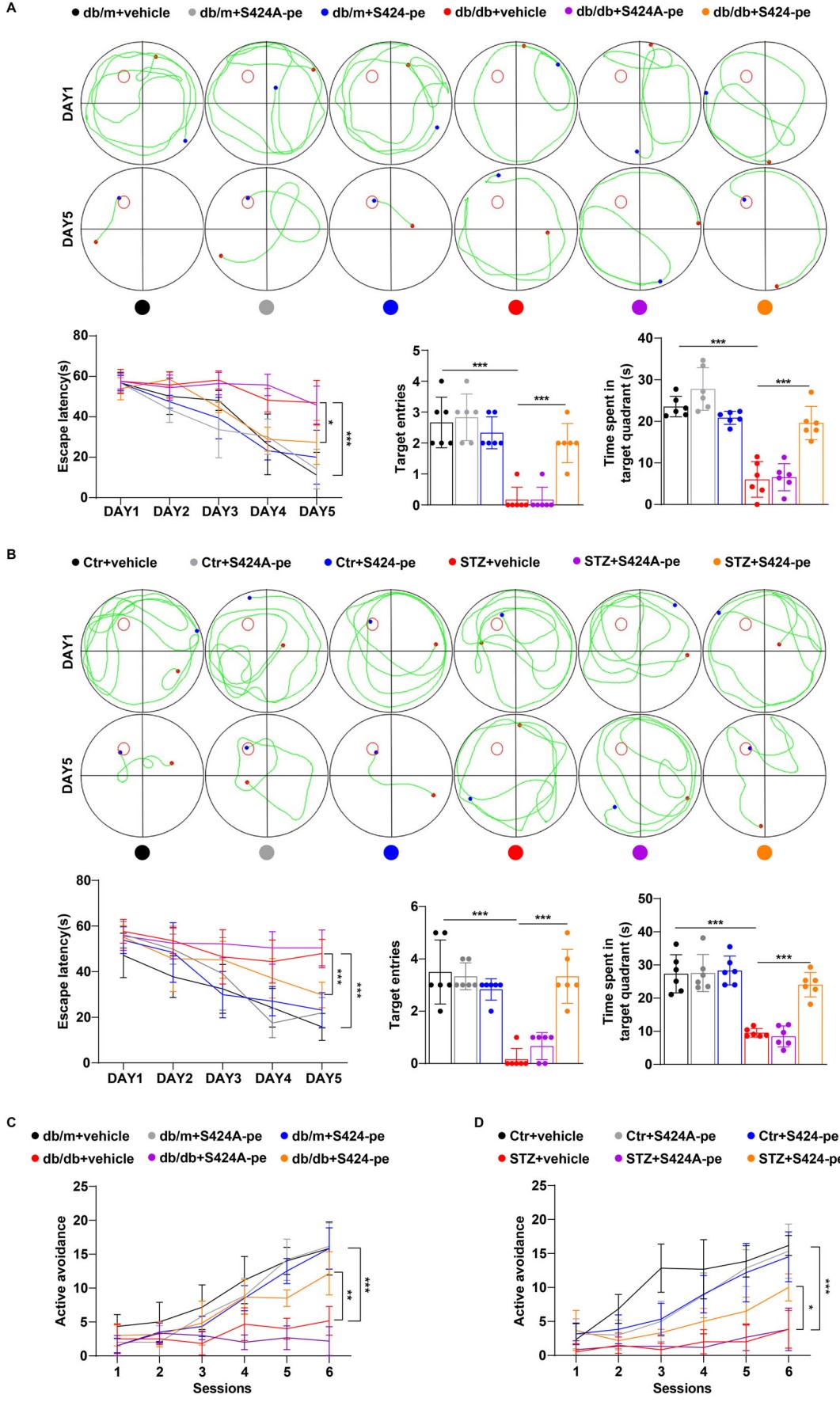

**Figure 7.  Short peptide S424-pe protects against high-glucose-induced cognitive dysfunction by regulating Bhlhe41/Dnajb4/NCLX signaling.**

(A–D) Representative track images and escape latency to the platform during the training trials, target entries and time spent in target quadrant in the probe trial of Morris water maze (**A, B**), and active avoidance performance (**C, D**) of db/m, db/db, Ctr, STZ mice treated with vehicle, S424A-pe (5 mg/kg) or S424-pe (5 mg/kg) ($n = 6$ mice per group). *P* values: <0.0001 (**A**, escape latency, db/m+vehicle vs db/db+vehicle), 0.0448 (**A**, escape latency, db/db+vehicle vs db/db+S424-pe), <0.0001 (**A**, target entries, db/m+vehicle vs db/db+vehicle), 0.0002 (**A**, target entries, db/db+vehicle vs db/db+S424-pe), <0.0001 (**A**, time spent in target quadrant, db/m+vehicle vs db/db+vehicle), <0.0001 (**A**, time spent in target quadrant, db/db+vehicle vs db/db+S424-pe); <0.0001 (**B**, escape latency, Ctr+vehicle vs STZ+vehicle), 0.0008 (**B**, escape latency, STZ+vehicle vs STZ + S424-pe), <0.0001 (**B**, target entries, Ctr+vehicle vs STZ+vehicle), <0.0001 (**B**, target entries, STZ+vehicle vs STZ + S424-pe), <0.0001 (**B**, time spent in target quadrant, Ctr+vehicle vs STZ+vehicle), <0.0001 (**B**, time spent in target quadrant, STZ+vehicle vs STZ + S424-pe); <0.0001 (**C**, db/m+vehicle vs db/db+vehicle), 0.005 (**C**, db/db+vehicle vs db/db+S424-pe); <0.0001 (**D**, Ctr+vehicle vs STZ+vehicle), 0.0134 (**D**, STZ+vehicle vs STZ + S424-pe). Data are means ± SEM. *$P$ < 0.05, **$P$ < 0.01, ***$P$ < 0.001. Two-way ANOVA followed by Tukey's test (**A–D**). Source data are available online for this figure.

that the observed increase in Bhlhe41 protein and mRNA levels could also be a consequence of altered translation mediated by Bmal1 (not just transcription), i.e., by enhancing translation and thereby stabilizing Bhlhe41 mRNA under high glucose. This, however, remains purely speculative and would be an interesting avenue for future research. Fourth, after intervention with MG132 in the control group, there was no significant further increase in the level of NCLX, which may be related to the compensatory mechanism of cellular calcium homeostasis (Rougier et al, 2013). When MG132 is used to inhibit the degradation of NCLX under physiological conditions, neurons may compensatorily reduce the generation of NCLX in order to avoid excessive expression of NCLX leading to increased mitochondrial calcium efflux, thereby maintaining the calcium homeostasis in the mitochondria. In the Dnajb4 overexpression group, intervention with MG132 did not completely restore NCLX expression levels. This may be related to the multifunctionality of Dnajb4. We speculate that while Dnajb4 mediates NCLX ubiquitination and degradation, it may also inhibit its generation. Additionally, differences in the duration and concentration of MG132 intervention may also lead to the above situation. The above speculations require further experimental verification in future studies. Finally, our results indicated that S424-pe treatment reduced Bmal1 O-GlcNAcylation at S424, but we did not demonstrate that binding to other off-targets was not responsible for the observed effects, which should be verified in subsequent studies by performing immunoprecipitation-mass spectrometry (IP-MS).

# Methods

## Ethical approval

All animal procedures were approved by the Animal Care and Experimentation Committee at Guilin Medical University and performed in accordance with approved guidelines.

## Mice

BKS.Cg-Dock7^m +/+ Lepr^db/J(db/m) and Bhlhe41^fl/+ mice were obtained from Jackson Laboratory and Cyagen Biosciences, respectively. C57BL/6 mice were provided by Changsha Hunan Silaike Jingda Laboratory Animal Co., Ltd. For Bhlhe41-floxed mice, two flox sequence (ATAACTTCGTATAGCATACATTA-TACGAAGTTAT) was inserted in the two terminals of Bhlhe41 exon1 and exon5. For genotyping, genomic DNA was extracted from tail tips. db/db mice were generated by self-crossing of db/m

mice. Bhlhe41^fl/+ mice were intercrossed to generate Bhlhe41^fl/fl mice. Bhlhe41^fl/+ mice and db/m mice were crossed to generate Bhlhe41^fl/+ db/m mice, and Bhlhe41^fl/fl db/db mice were produced by further breeding between Bhlhe41^fl/+ db/m heterozygous mice. All mice were housed under standard laboratory conditions (12 h light/dark cycle, 07:00 to 19:00 light on) and temperature (23–24 °C) with ad libitum access to water and standard laboratory chow diet. The sample collection time is between 8:00 AM and 9:30 AM.

Type 1 diabetes was induced by streptozotocin (STZ) injection following the Low-Dose Streptozotocin Induction Protocol (Wu et al, 2019). Briefly, type 1 diabetes was induced in 8-week-old C57BL/6 and Bhlhe41^fl/fl mice via daily STZ intraperitoneal injections (50 mg/kg in citrate buffer, pH 4.5) for 5 consecutive days. Control mice received the same volume of citrate buffer injections (vehicle). Blood glucose was measured two weeks after injection, and the mice with blood glucose levels greater than 16.7 mmol/L were considered diabetic. Mice were used for experiments three months post-induction. To further test whether and how short peptide could improve cognitive function in diabetes, STZ-induced diabetic mice and db/db mice were given short peptide at a dose of 5 mg/kg/day by intraperitoneal injection for 4 weeks, while the age-matched control groups were administered the same volume of vehicle (saline). The total number of mice analyzed for each experiment was reported in the figure legends.

## Cell isolation and culture

Primary cultures of hippocampal neurons were prepared from embryonic (E16-18) mice brain, as previously described (Sun et al, 2022). Briefly, the hippocampus was extracted from the brain and subsequently sectioned into 1-mm³ fragments using a blade. The shredded tissue was then subjected to trypsin digestion at a concentration of 0.125% and incubated at 37 °C for 10 min. Following enzymatic digestion, the reaction was terminated by 10% fetal bovine serum (FBS). The resulting tissue supernatant was collected and centrifuged at $1000 \times g$ for 5 min. The cell pellet underwent a cold PBS wash and was subsequently suspended in Neurobasal A medium supplemented with 2% B27 (Invitrogen, Cat# 17504044) and glutamine (0.5 mM, Invitrogen, Cat# 25030081). Finally, neurons were plated on poly-l-lysine-coated six-well plates (Sigma-Aldrich, Cat# P4707) and cultured at 37 °C with 5% $CO_2$. The extraction site of primary hippocampal neurons is not limited to the CA1 region of the fetal mouse hippocampus.

For in vitro experiments in primary hippocampal neurons, high glucose was engendered by raising the glucose 25 mM over ambient

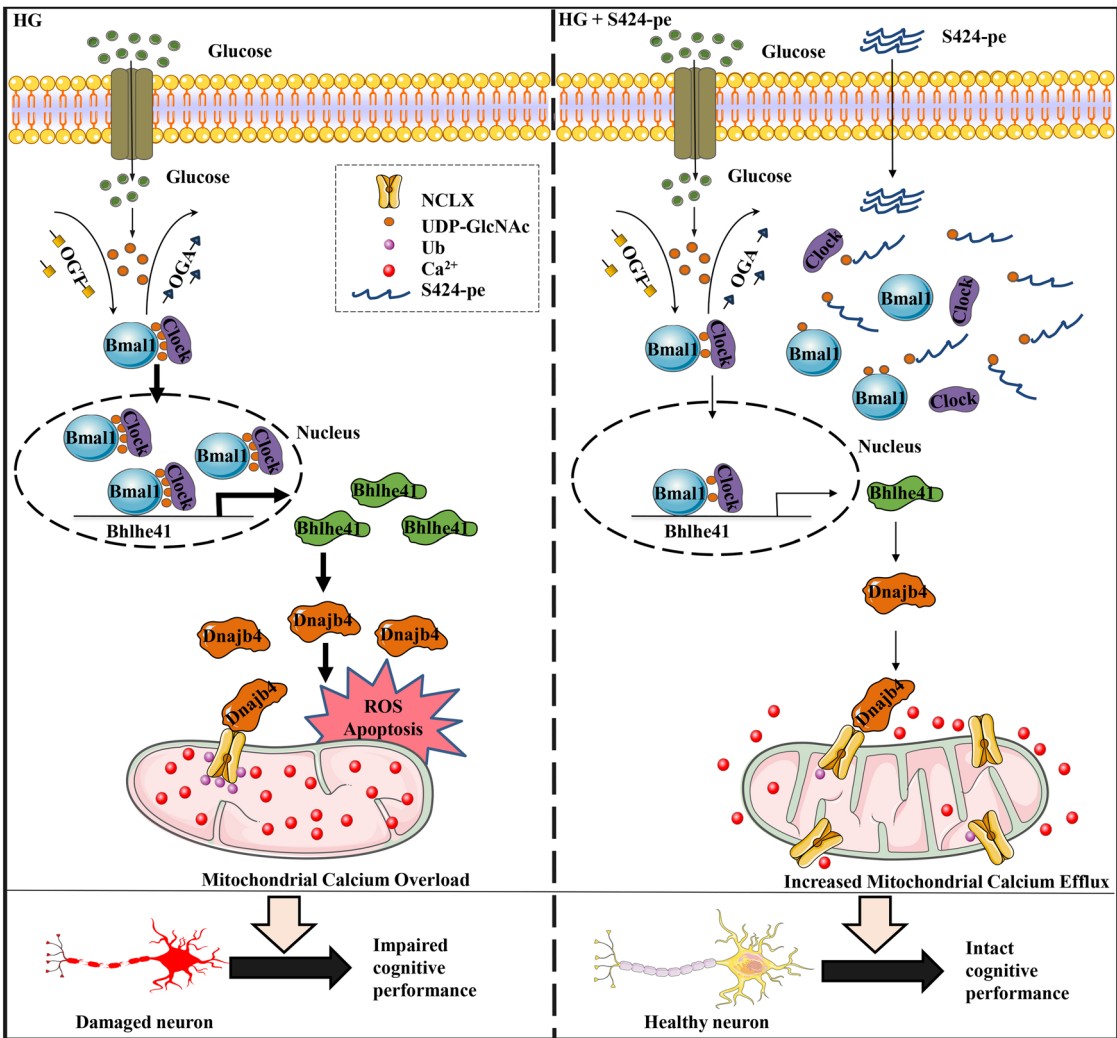

**Figure 8.** Schematic diagram depicting the possible mechanisms through which high glucose induces O-GlcNAcylation of circadian clock protein Bmal1 and cognitive impairment in diabetes.

conditions. The selection of this glucose level was based on the findings of preliminary studies, which revealed a significant dose-response relationship between glucose levels and its effects on hippocampal neurons. Additionally, this level was chosen to emulate moderate-to-severe type 2 diabetes, as it corresponds to a blood glucose range of 450 mg/dl. To exclude the potential osmotic effects of high glucose, equimolar mannitol was used as an osmotic control.

Hippocampal CA1 neurons were isolated as previously described (Guo et al, 2012). Briefly, indicated-age adult mice were subjected to deep anesthesia using isoflurane and subsequently underwent transcardial perfusion with ice-cold saline. The entire brain was promptly extracted and rinsed with prechilled phosphate-buffered saline (PBS). The brain was then sectioned into 400 μm coronal sections using the Tissue Chopper, and the sections containing the hippocampus (4–6 sections) were collected using a wet sterile swab and placed in a petri dish containing 5 ml of 1 × HBSS solution supplemented with 30 mM glucose, 2 mM HEPES, and 26 mM NaHCO₃. The CA1 region of

the mouse hippocampus was then meticulously dissected under a dissecting microscope. After dissection, the hippocampal CA1 region was mechanically and enzymatically lysed using the Adult Brain Dissociation Kit (Miltenyi Biotec, Cat# 130-107-677) to obtain a single-cell suspension. Subsequently, debris and red blood cell were removed, and the total cell pellet was resuspended with PBS containing 0.5% BSA. Neurons were then isolated by negative selection using an adult neuron isolation kit (Miltenyi Biotec, Cat# 130-126-603) following the manufacturer's instructions. Briefly, cells were subjected to staining with Biotin-Antibody Cocktail that targets non-neuronal cells such as microglia, oligodendrocytes, astrocytes, fibroblasts, and endothelial cells. This process was carried out at a temperature of 4 °C for a duration of 5 min, followed by washing with 1×PBS and 0.5% BSA. The cells were then centrifuged, resuspended, and incubated with anti-biotin microbeads for a period of 10 min at 4 °C. Separation of neuron cells was achieved by passing the cells through an LS column under a magnetic field. The purity of hippocampal CA1 neurons was above 95% as determined by flow

cytometry (Appendix Fig. S1I), which is consistent with the manufacturer's prior report.

## Stereotaxic injection

Four-month-old mice from different groups were anesthetized with isoflurane and fixed onto a stereotaxic frame that was equipped with a digital displacement transducer and a syringe pump. Mice were bilaterally injected with virus in the hippocampal CA1 region via a Nanoject III (Drummond) injector, with target coordinates of $x = 1.0$ mm right or left of midline, $y = -1.9$ mm posterior to bregma, and $z = 1.5$ mm below the brain surface. The injections were conducted at a rate of 0.05 μL/min, and the needle was left in place for 5 min post-injection to minimize backflow. Bhlhe41-floxed mice underwent stereotaxic injection of 0.3 μL AAV-CAMKII-Cre or AAV-CAMKII control virus in the dorsal CA1 region of the hippocampus to specifically eliminate Bhlhe41 expression in hippocampal CA1 neurons. In addition, 0.3 μL AAV-CAMKII-NCLX or AAV-CAMKII control virus was administered bilaterally in the hippocampal CA1 region of db/db, STZ-induced diabetic mice and their respective controls to specifically overexpress NCLX in hippocampal CA1 neurons. Behavioral experiments were conducted four weeks following bilateral intracranial injection.

## Gene silencing and overexpression

For siRNA-mediated knockdown of Bmal1, Bhlhe41, OGT, and OGA in vitro, primary hippocampal neurons were transfected with either the targeting or control siRNA (100 nM) using Lipofectamine RNAiMAX (Invitrogen, Cat# 13778030) according to the manufacturer's instruction. The following siRNA sequences were used: siBmal1 5'- CATTGATGAATTGGCTTCTTTGGTA-3'; siBhlhe41 5'-GCTGAAAGATTTACTGCCCGA-3'; siOGT 5'-GAAGGCTGGACAGATCTGCTCATTT-3'; siOGA 5'-AAGAGCTCTTTAGAAGGCTTCAGAA-3. Knockdown efficiency was assessed by immunoblot analysis after 2 days of siRNA transfection.

To overexpress OGT, Bmal1, Clock, Bhlhe41, and Dnajb4, primary hippocampal neurons were transfected with plasmids encoding the indicated proteins or control plasmids using Lipofectamine 3000 (2 μL/μg DNA, Invitrogen, Cat# L3000001), according to the manufacturer's instructions. All overexpression plasmids were constructed using the GV657-CMV-FLAG-GFP-Puromycin vector by GeneChem (Shanghai, China. vector backbone: https://www.genechem.com.cn/index/supports/zaiti_info.html?id=207). Overexpression efficiency was verified by immunoblot analysis after 2 days of plasmid transfection.

Primary hippocampal neurons were stably transfected shRNAs against Bmal1 via lentivirus infection (hU6-MCS-CBh-gcGFP-IRES-puromycin, vector backbone: https://www.genechem.com.cn/index/supports/zaiti_info.html?id=83, GENECHEM) to knock-down the endogenous expression of Bmal1, the knockdown efficiency was confirmed by immunoblot analysis. The shRNA targeted sequence for Bmal1: 5′-CGCGGAGGAAATCATG-GAAAT-3′. The Bmal1 (S424A, S518A and S573A) mutant plasmid was generated using a QuikChange XL site-directed mutagenesis kit (Agilent Technologies, CA, USA, Cat# 200516).

## Immunoblot analysis

Hippocampal neurons were homogenized in RIPA lysis buffer supplemented with 1% proteinase inhibitor and phosphatase inhibitors for 30 min on ice and centrifuged at $12,000 \times g$ for 15 min at 4 °C. The supernatant was collected, and the protein concentration was determined using the BCA Protein assay Kit and normalized to 10 μg/μL. Proteins were separated using 10% SDS-polyacrylamide gel electrophoresis (PAGE) gels and transferred onto polyvinylidene difluoride (PVDF) membranes. Following protein transfer, the membranes were blocked with QuickBlock Blocking Buffer (Beyotime, Cat# P0252) for 15 min at room temperature on a shaker. Subsequently, the membranes were washed thrice with Tris-buffered saline containing 0.1% Tween-20 (TBST) buffer and then incubated with indicated primary antibody diluted in Primary Antibody Dilution Buffer (Beyotime) overnight at 4 °C: anti-Bmal1 antibody (1:5000, Proteintech, Cat# 14268-1-AP, RRID: AB_2878037), anti-Clock antibody (1:3000, Affinity Biosciences, Cat# AF0323, RRID: AB_2833486), anti-Bhlhe41 antibody (1:2000, Affinity Biosciences, Cat# AF0442, RRID: AB_2834318), anti-Dnajb4 antibody (1:2000, Proteintech, Cat# 13064-1-AP, RRID: AB_2094422), anti-OGT antibody (1:1000, Proteintech, Cat# 11576-2-AP, RRID: AB_2156943), anti-OGA antibody (1:3000, Proteintech, Cat# 14711-1-AP, RRID: AB_2143063), anti-O-GlcNAc antibody (1:1000, Cell Signaling Technology, Cat# 9875S, RRID: AB_10950973), anti-ubiquitin antibody (1:1000, Proteintech, Cat# 10201-2-AP, RRID: AB_671515), anti-SLC24A6 antibody (1:1000, Proteintech, Cat# 21430-1-AP, RRID: AB_10858637), anti-IP3R-II antibody (1:100, Santa Cruz Biotechnology, Cat# sc-398434, RRID: AB_2637028), anti-Bax antibody (1:1000, beyotime, Cat# AF0057, RRID: AB_2923045), anti-Bcl-2 antibody (1:1000, beyotime, Cat# AF0060, RRID:AB_2923046), anti-GAPDH antibody (1:5000, Proteintech, Cat# 10494-1-AP, RRID: AB_2263076). After being washed with TBST three times, the membranes were incubated with horseradish peroxidase-conjugated secondary antibody at a dilution of 1:5000 for 1 h. The signal was detected by ECL Chemiluminescent Substrate Reagent Kit with ChemiDoc XRS+ System (BioRad) and quantified by Image Lab.

## Immunoprecipitation (IP) and Co-IP analysis

Immunoprecipitation (IP) and Co-IP of target proteins were conducted using Pierce Magnetic IP/Co-IP kit (Thermo Fisher Scientific, Cat# 88804). After the intervention, HEK293T and hippocampal neuron cells were lysed in IP lysis buffer with proteinase inhibitors for 5 min on ice and centrifuged at $13,000 \times g$ for 10 min at 4 °C. The supernatant was collected and incubated with indicated antibodies overnight at 4 °C on the shaker platform. Normal IgG was used as an immunoprecipitation control. Anti-Bmal1, anti-OGT, anti-NCLX and anti-Dnajb4 antibodies were used at 10 μg per 1000 μg total protein. Next, the antibody mixture was incubated together with pre-washed protein A/G magnetic beads for 1 h at room temperature. After extensive washing, the precipitated proteins were eluted from the magnetic beads and subjected to gel electrophoresis and immunoblotting using antibodies against Bmal1, Clock, Bhlhe41, Dnajb4, NCLX, O-GlcNAcylation, or ubiquitin.

## RNA extraction and RT-qPCR

Total RNA was extracted using cold Trizol followed by chloroform extraction and isopropanol precipitation. Reverse transcription (RT) was conducted using FastKing RT Kit (TIANGEN, Cat#KR116-02). The reaction mixture (20 μl total) was composed of 1 μL diluted cDNA, 0.5 uL of primers (Bhlhe41: F: 5'-AGCGAGACGATACCAAGGATACC-3', R: 5'-CAAGACTACTGCTTTCTCCAAATGC-3'; Dnajb4: F: 5'-AACAC-CAAATAGTATCCCAGCAGAC-3', R: 5'-CCACACAATGCCTCC-CGTAAAC-3'; Bhlhe40: F: 5'-TGGCGAAGCATGAGAACACT-3', R: 5'-CTTTGGGAGCCGAGTCCAAT-3'; Cry1: F: 5'-AGGCTGGCGTG-GAAGTCATC-3', R: 5'-TGGTGTCTGCTGGCATCTCC-3'; Cry2: F: 5'-TGGTTCCGCAAAGGACTACG-3', R: 5'-GGTCGAGGATGTA-GACGCAG-3'; Per1: F: 5'-CTCAGAGTCCCAGACCAGGTG-3', R: 5'-AGTGGAGGACGAAACAGGGA-3'; Per2: F: 5'-GAAGCGCTTATTC-CAGAGCCC-3', R: 5'-GAGCCACTGCTCATGTCCAC-3'; Per3: F: 5'-TCCTGTGTTCAAGCAGGGTC-3', R: 5'-GTGACATGGGGACA-GAGGTG-3'; Usp2: F: 5'-TACAGAATCGTCCCGCTACAC-3', R: 5'-CCCCTGTCACAGTCCAGAAT-3'; Fosb: F:5'- TTTTCCCGGAGAC-TACGACTC-3', R: 5'-GTGATTGCGGTGACCGTTG-3'; GAPDH: F: 5'-GGTTGTCTCCTGCGACTTCA-3', R: 5'-TGGTCCAGGGTTTCT-TACTCC-3'), 10 μL SYBR mix and 8 μL nuclease-free water. Gene expression was then analyzed by RT-qPCR using BioRad CFX96 ™ Real-Time PCR Detection system. The relative expression levels of target mRNA were normalized to glyceraldehyde 3-phosphate dehydrogenase (GAPDH) and calculated using the $2^{-\triangle\triangle CT}$ method (Sun et al, 2022).

## Chromatin immunoprecipitation

The ChIP assay was performed as previously described (Hui et al, 2023). Primary hippocampal neurons were cross-linked in 1% formaldehyde for 10 min at room temperature and then neutralized with glycine. The fixed neuron cells were resuspended in lysis buffer containing protease inhibitors and subsequently subjected to 16 cycles of sonication (30 s on and 30 s off) to produce chromatin fragments ~400 base pairs in length. For immunoprecipitation, the diluted chromatin was incubated with normal IgG (control) or Bhlhe41 antibodies overnight at 4 °C with constant rotation, followed by incubation with Protein A + G Agarose for an additional 2 h. After extensive washing, the precipitated DNA fragments were eluted from agarose, de-cross-linked at 65 °C and purified using the PCR Purification Kit (TIANGEN, Cat# 28104). Dnajb4 promoter enrichment was analyzed using RT-qPCR of ChIP DNA versus input. Primers for promoter amplification were: Dnajb4: F: 5'-AGACTGGCAGACGGAAATCCC-3', R: 5'-AGAGAGACCCTAC-TCTCGTGGGA-3'; Bhlhe40: F: 5'-CCGCCCCACTTCTCATTCAC-3', R: 5'-CTGAGCTTCCTTGCAATGCG-3'; Bhlhe41: F: 5'-GATAGC-GAATGTCCGAGAGTGAA-3', R: 5'-TGTCCCCACAACTCAAGC-TAAAT-3'; Cry1: F: 5'-CTCTATCAGAGGAAAACCCAGCAC-3', R: 5'-TTGCCTCAGCTTCAGGAACAC-3'; Cry2: F: 5'-CAGGAGAA-TATTGCCCTGCACATC-3', R: 5'-CGTGGGTTGAGGCAGTAAT-TATGC-3'; Per1: F: 5'-AAGAGGGCATCGGATCCATTTC-3', R: 5'-ACTGCTCTTCCAAAGGTGCTG-3'; Per2: F: 5'-ATTACCGAGGCT-GGTCACGTC-3', R: 5'-AGTAGGCTCGTCCACTTCCG-3'; Per3: F: 5'-TCACAATCTGGCCCTTCCTTTC-3', R: 5'-AGCAATCACAT-GGTGGCTCAC-3'.

## Immunofluorescence staining

The mice were subjected to anesthesia using isoflurane and subsequently underwent transcardial perfusion with 0.9% saline solution followed by 4% paraformaldehyde prior to the removal of the brain. The brain was then fixed in a solution of 4% paraformaldehyde in PBS at 4 °C for a duration of 12 h. To ensure cryoprotection, the brains were stored in a 30% sucrose solution in PBS at 4 °C for a period of 2 days. Following dehydration, the tissues were embedded in OCT (Tissue-Tek) and cryopreserved at a temperature of −20 °C. Subsequently, the tissues were serially sectioned into 5 μm-thick coronal sections using a freezing microtome (Leica). The antigen retrieval process was performed using the Quick antigen retrieval solution (Beyotime, Cat# P0090). Following antigen retrieval, the sections were subjected to pre-incubation with QuickBlock™ Blocking Buffer (Beyotime, Cat# P0252FT) for 1 h. The respective primary antibodies were then incubated with the sections overnight at 4 °C. Subsequently, the secondary antibodies were incubated at room temperature for 1 h while being protected from light. The sections were then counter-stained with DAPI (5 μg/ml) for a duration of 15 min, and the slices were sealed with anti-fluorescence quenching tablets. For in vitro imaging, primary hippocampal neurons were cultured in a 35-mm dish and subsequently fixed and stained according to the aforementioned protocol. The resulting images were observed utilizing a fluorescence microscope manufactured by Olympus.

## Mitochondrial Ca²⁺ measurement

The Rhod-2 AM probe (Thermo Fisher Scientific, Cat# R-1245MP) was utilized to measure mitochondrial calcium levels in primary hippocampal neurons. Neurons were subjected to incubation with 5 μM Rhod-2 AM and 0.02% Pluronic F-127 for 30 min at 37 °C, followed by washing to eliminate any dye that may have been nonspecifically associated with the cell surface. Subsequently, the neurons were incubated for an additional 30 min to allow for complete de-esterification of intracellular AM esters. Rhod-2 signal was then collected via flow cytometry (BD Accuri C6 Plus) and analyzed using FlowJo (v10) software.

## Mitochondrial membrane potential measurement

JC-1, a fluorescent probe, is commonly employed to detect mitochondrial membrane potential (ΔΨm) through the observation of fluorescent shifts. In instances where the mitochondrial membrane potential is elevated, JC-1 aggregates within the mitochondria matrix, forming a polymer that emits red fluorescence (Ex = 585 nm and Em = 590 nm). Conversely, when the membrane potential is reduced, JC-1 remains in a monomeric state and is unable to accumulate in the matrix, resulting in green fluorescence (Ex = 514 nm and Em = 529 nm). The ratio of red to green fluorescence intensity serves as an indicator of mitochondrial depolarization, with a reduced ratio commonly associated with mitochondrial dysfunction. Neurons were incubated with JC-1 working solution (Beyotime, Cat# C2006) for 20 min at 37 °C in the dark. Then JC-1 dye was removed and washed twice with JC-1 buffer. Following washing, alterations in red and green fluorescence were assessed through flow cytometry analysis.

## Measurement of mitochondrial reactive oxygen species (mtROS)

The measurement of mtROS in primary hippocampal neurons was performed utilizing MitoSOX™ Red (Thermo Fisher Scientific, Cat# M36008). The primary hippocampal neurons were exposed to a working solution of MitoSOX™ reagent at a concentration of 2.5 μM for 10 min at 37 °C, followed by a washing step with HBSS to eliminate any residual MitoSOX. The mean fluorescence intensity was subsequently assessed via flow cytometry, and the resulting data were analyzed using FlowJo (v10) software.

## Apoptosis assays

Apoptosis in primary hippocampal neurons was assayed using Annexin V-FITC Apoptosis Detection Kit (Beyotime, Cat# C1062L). Primary hippocampal neurons were incubated with 5ul Annexin V-FITC and 10ul Propidium iodide (PI) in the dark for 15 min at room temperature. Fluorescence-activated cell sorting (FACS) analysis was performed to analyze the apoptotic neurons, with both Annexin V + /PI− and Annexin V + /PI+ cells being considered apoptotic.

## Morris water maze

The Morris water maze (MWM) test was conducted in a circular tank measuring 120 cm in diameter and 50 cm in height, which was filled with opacified water at a temperature of 20 ± 1 °C. The tank was equipped with a 10 cm diameter platform that was submerged 1 cm below the water surface. The pool was partitioned into four quadrants and the mice were released from four distinct starting positions facing the wall of the pool and allowed to explore the platform for a duration of 60 s. In the event that the mouse failed to locate the platform during the exploration period, it was guided to the platform by the experimenter. For learning, each mouse underwent four trials per day for 5 consecutive days following a single day of adaptation. Upon completion of the 5th day of training, the platform was removed, and a probe trial was executed. The Video Tracker software (Anymaze) was utilized to record behavioral parameters such as escape latency, platform crossings, time in target quadrant, and swimming paths.

## Active avoidance test

Auditory learning and memory of mice were evaluated by active avoidance test using a shuttle BOX (Xinruan, Shanghai, China). The active avoidance test was performed through a computer program that generated auditory, visual, and electrical stimuli. In the training phase, each mouse was placed into the test chamber and permitted unrestricted movement into an alternate chamber. After a 10-s interval, a conditional stimulus (CS) comprising of tone and light was generated. If the mouse failed to escape into the alternate chamber during the CS, an electric shock (0.3 mA, 5 s) was administered. After a 1-day adaptation period, each mouse underwent 3 consecutive days of training consisting of two trials per day, during which their avoidance responses were recorded.

## Peptide synthesis

The peptides utilized in this study were synthesized through Fmoc solid phase synthesis on Rink amide resin. Initially, the resin was swelled and deprotected with dichloromethane (DCM). The condensation reaction was carried out by using Fmoc-AA (amino acid)-OH, hydroxybenzotriazole (HOBT) dissolved in dimethylformamide (DMF), followed by diisopropylcarbodiimide (DIC), and added to the drained resin for 1 h with nitrogen bubble reaction. To remove the Fmoc protecting groups, the resin was washed four times with DMF. After acetic anhydride capping, a cutting solution consisting of trifluoroacetic acid (TFA), $H_2O$, 1,2-ethanedithiol (EDT), and triisopropylsilane (TIS) in a volume ratio of 95:1:2:2 was added to the resin for 2 h to cut the peptide. Synthetic peptides underwent purification via high-pressure liquid chromatography to achieve a purity exceeding 98%, suitable for both in vitro and in vivo applications. Peptides were dissolved in phosphate-buffered saline (PBS) to create a 20 mM stock solution for in vitro experiments. For in vivo use, S424-pe and S424A-pe were dissolved in PBS and stored on ice prior to injection. Prior to injection, the solution was allowed to reach room temperature. Synthetic peptide sequences are as follows:

S424-pe-1#: YGRKKRRQRRRITLRSRWFSFMNP
S424-pe-2#: YGRKKRRQRRRDGSFITLRSRWFS
S424-pe-3#: YGRKKRRQRRRSFITLRSRWFSFM
S424-pe-4#: YGRKKRRQRRRLRSRWFSFMNPWT
S424-pe-5#: YGRKKRRQRRRIKDGSFITLRSRW
S424A-pe: YGRKKRRQRRRDGSFITLRARWFS

## ScRNA-Seq data analysis

The single-cell RNA sequencing data was obtained from the Gene Expression Omnibus (GEO) database (https://www.ncbi.nlm.nih.gov/geo/query/acc.cgi?acc=GSE201644). Seurat package was used to generate the object and filter out cells with poor quality. Subsequently, standard data preprocessing was carried out, which involved the computation of gene numbers, cell counts, and mitochondria sequencing count percentages. Genes with fewer than three detected cells were excluded, and cells with less than 200 detected gene numbers were discarded. Genes that were detected in a minimum of three cells were retained, while cells with a gene count below 200 or above 8000, as well as those with a high mitochondrial content (>20%), were filtered out. After discarding poor-quality cells, a total of 19,358 cells were selected for downstream analysis. To normalize the library size effect in each cell, UMI counts were scaled using scale.factor = 10,000. The data was then log-transformed, and other factors such as "percent.mt", "nCount_RNA", and "nFeature_RNA" were corrected for variation regression using the ScaleData function in Seurat (v4.3.0).

The corrected normalized data metrics were applied to the standard analysis as described in the Seurat R package. Highly variable genes (top 2000) were extracted to perform the principal component analysis (PCA) and top 20 of significant principle components were used for UMAP visualization and cluster analysis. Cell clustering was performed using the FindClusters function (resolution = 0.5) in Seurat. The R package SingleR (v2.0.0) was used to automatically annotate cell types. Briefly, this algorithm computes the spearman correlation between the transcriptome of the test cell and reference data to determine cellular identit (Tang et al, 2022). The reference datasets used in this study was MouseRNAseqData (https://rdrr.io/github/LTLA/celldex/man/MouseRNAseqData.html).

## pySCENIC analysis

The gene regulatory network for hippocampal neurons was inferred using pySCENIC (v0.11.2, a lightning-fast Python implementation

of the SCENIC pipeline) (Liu et al, 2023). First, the GRNboost2 algorithm was used to infer gene regulatory network and generate co-expression modules (pySCENIC grn). Next, the RcisTarget package (pySCENIC ctx) was used to evaluate target binding motif enrichment and create regulons with only genes containing a binding motif, where a regulon is a transcription factor and its target genes. The mouse motif collection v9 (motifs-v9-nr.mgi-m0.001-o0.0.tbl) and the cisTarget databases for mm9 (mm9-tss-centered-10kb-7species.mc9nr.genes_vs_motifs.rankings.feather) were used in the pipeline and downloaded from https://resources.aertslab.org/cistarget/. Finally, the measurement of regulon activity was conducted using AUCell (pySCENIC aucell). Specifically, AUCell used the area under the curve to determine the enrichment of the regulon across the ranking of all genes in a particular cell, thereby generating a matrix that captured the activity of each regulon in each cell. Downstream analyses were performed utilizing the Seurat package (Pham et al, 2022).

### Transcription factors and O-glycosylation sites prediction

Transcription factors that could bind to the promoter region of Bhlhe41 were predicted by PROMO (ALGGEN, http://alggen.lsi.upc.es/cgi-bin/promo_v3/promo/promoinit.cgi?dirDB=TF_8.3) and STRING database (https://string-db.org/). O-glycosylation sites were predicted using the O-GlcNAc Database (v1.3, https://oglcnac.mcw. edu), YinOYang (v1.2, https://services.healthtech.dtu.dk/services/YinOYang-1.2/) and NetO-Glyc (v4.0, https://services.healthtech.dtu.dk/services/NetOGlyc-4.0/).

### Differential expression analysis and functional enrichment analysis

Differential expression analyses (DEGs) were visualized as volcano plots which were generated by using GraphPad. Functional enrichment analyses of DEGs were performed using the KOBAS 3.0 website (http://kobas.cbi.pku.edu.cn/kobas3).

### Statistical analysis

Data were expressed as mean ± standard error of mean (SEM) and analyzed using Prism version 9.0 software (GraphPad). Kolmogorov–Smirnov test was used to evaluate the normality of the data distribution. Comparisons between two groups were performed using two-tailed Student's $t$ test for normally distributed data and Mann–Whitney rank-sum test for non-normally distributed data. One-way ANOVA followed by post hoc Tukey's test was used to compare differences between multiple groups with one variable, whereas two-way ANOVA followed by post hoc Tukey's test was utilized to compare multiple groups with more than one variable. For non-normally distributed data, we performed a nonparametric statistical analysis using Kruskal–Wallis test followed by Dunn's post hoc test for multiple-group comparisons (Fu et al, 2020). Details regarding the statistical test, biological sample size ($n$), and $P$ value were reported in the corresponding figure legends. In figures, asterisks are used to indicate statistical significance ($*P < 0.05$, $**P < 0.01$, $***P < 0.001$). The mice were randomly allocated into different groups, and the investigators were blinded to the group allocation during surgeries and outcome evaluations. The exclusion criteria were established based on the animals' well-being at the study's outset, and

no animals were excluded from the study. A power analysis was not conducted to determine the sample size, and instead, the sample size for each study was determined based on prior experience with diabetic mice in our laboratory.

## Data availability

This study did not generate any unique datasets or codes. All other data can be made available from the authors on reasonable request.

The source data of this paper are collected in the following database record: biostudies:S-SCDT-10_1038-S44318-024-00263-6.

## Peer review information

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

## Acknowledgements

This study was supported by grants from the National Natural Science Foundation of China (Grant No. 82260171, 82470878), the Natural Science Foundation of Guangxi Province for Distinguished Young Scholars (Grant No. 2021GXNSFFA196003), the Scientific Research and Technology Development Projects of Science and Technology Department of Guangxi Province (Grant No. 1598012-13), the Natural Science Foundation of Guangxi Province (Grant No. 2015GXNSFBA139119), the Innovation Project of Guangxi Graduate Education (No. YCSW2023409, YCSW2023424, YCSW2024459) and the Graduate Research Program of Guilin Medical University (No. GYYK2022016).

## Author contributions

**Ya Hui**: Formal analysis; Writing—original draft. **Yuanmei Zhong**: Formal analysis; Writing—original draft. **Liuyu Kuang**: Formal analysis; Writing—original draft. **Jingxi Xu**: Formal analysis; Writing—original draft. **Yuqi Hao**: Investigation. **Jingxue Cao**: Investigation. **Tianpeng Zheng**: Supervision; Funding acquisition; Writing—review and editing.

Source data underlying figure panels in this paper may have individual authorship assigned. Where available, figure panel/source data authorship is listed in the following database record: biostudies:S-SCDT-10_1038-S44318-024-00263-6.

## Disclosure and competing interests statement

The authors declare no competing interests.

