## [Peer Review File · The EMBO Journal]

O-GlcNAcylation of circadian clock protein Bmal1 impairs cognitive function in diabetic mice

Ya Hui, Yuanmei Zhong, Liuyu Kuang, Jingxi Xu, Yuqi Hao, Jingxue Cao, and Tianpeng Zheng

Corresponding author(s): Tianpeng Zheng (ZhengTP@glmc.edu.cn)

Review Timeline:

Submission Date:	2nd Feb 24
Editorial Decision:	26th Mar 24
Appeal Received:	2nd May 24
Editorial Decision:	8th Jun 24
Revision Received:	19th Sep 24
Accepted:	25th Sep 24

Editor: Kelly Anderson

Transaction Report:

Dear Dr. Zheng,

Thank you for submitting your manuscript for consideration by the EMBO Journal. It has now been seen by three referees whose comments are shown below.

Given these opinions and the fact that the EMBO Journal can only afford to accept papers which receive enthusiastic support from a majority of referees, I am afraid we cannot offer to publish it here. Although we can see that the referees are in principle interested in this area of research, we believe the number of experiments required to fully address the referee concerns would take longer than our normal revision time. However, should you be able to address the concerns in full in the future, we would be willing to consider it for publication at that time, but we would once again reassess the current literature for novelty.

Thank you in any case for the opportunity to consider this manuscript. I am sorry we cannot be more positive on this occasion, but we hope nevertheless that you will find our referees' comments helpful.

Yours sincerely,

Kelly M Anderson, PhD
Editor, The EMBO Journal
k.anderson@embojournal.org

Referee #1:

Review of "O-GlcNAcylation of circadian clock protein Bmal1 impair cognitive function in diabetes".

Authors are proposing a mechanism by which O-GlcNAcylation of a key circadian clock protein, BMAL1, is a mechanism by which GLP1 agonists are neuroprotective in mouse models of Type 2 diabetes, through the intermediary transcription factor Bhlhe41 (commonly known as Dec2). The authors are generally provide complementary data from cell-based mechanistic studies with key in vivo manipulations in two different (ob/ob and STZ) mouse models of diabetes. They provide an absolute tremendous amount of data that generally appear very rigorous in approach as they map out a pathway from elevated glucose to dysregulated calcium regulation in the mitochondria leading to neuronal damage. There are important but hopefully minor problems with the data that left unfixed could bely the authors' credibility (noted below). More substantially, there is a general lack of placing studies in the proper context with regard to circadian rhythmicity and Bmal1 function, and other essential and readily doable experiments that provide 'causative' links between pathways components are largely and unexpectedly missing. Overall, while individual data are mostly good, the authors seem to present several 'stories' that all related to O-GlcNAcylation and cognitive decline in diabetes that lack direct connections between them. Finally, the studies suggesting Bmal1 is involved (glycosylation or not) are wholly insufficient to support its role proposed by the authors. These and additional issues are noted below.

1. Bhlhe41 (Dec2) is known to be robustly rhythmically expressed by Clock:Bmal1 and the core clockwork, yet no attention is paid to the potential rhythmic nature of Dec2 expression in the hippocampus. Authors first and foremost need to demonstrate that the differences in Dec2 abundance in vivo are not due to A) - differences in timing of tissue collection as a few hours apart could account for many of their in vivo findings, misrepresenting the actual phenotype; and B) if this was all due to differential regulation of BMAL1 function as the authors propose, there would be wide-spread alterations in circadian function well beyond elevated expression of Bhlhe41 (Dec2). These were not seen in scRNAseq data (Fig 1), nor demonstrated or alluded to anywhere. This is a substantial issue as the authors propose that elevated glucose increase O-GlcNac of BMAL1, a modification that appears to increase its binding to CLOCK (Fig 3) to drive downstream gene expression - if this is truly the case, expression of many more CLOCK:BMAL1 targets should also be directly altered; data in this regard is the first essential piece needed to provide evidence that BMAL1 is at all involved here.
2. The authors provide no direct evidence that O-GlcNAcylation of BMAL1 directly impairs cognitive function (as indicated in the title). They show that O-GlcNAcylation of BMAL1 can happen in a manner coincident with diabetic cognitive decline, but there is no substantial indication that it actually plays a role - many proteins are regulated by O-GlcNAcylation/high glucose. The title must be changed at a minimum.
3. There is no direct evidence provided that demonstrate O-GlcNacylation of Bmal1 changes its transcriptional activity/function here, short of a potential decrease in binding with CLOCK. They did find the key residue in BMAL1 that is o-glcnaacylated, but did not provide any functional studies indicating that mutating this site alters BMAL1 function. Likewise, there were no experiments demonstrating that glycosylation of Bmal1 is functionally involved in regulation of Bhlhe41 expression, in any form. This is a

critical omission, and very surprising given the other detailed studies provided throughout.

4. According to the mechanism shown in Fig 8, and the GLP-1 agonist studies, if these were interacting through correcting or altering BMAL1 activity, then obese/diabetic patients taking these would have substantial alterations in circadian function and sleep regulation. The point is that if BMAL1 was a key mediator here as the authors propose, broad circadian dysfunction would be a highly likely outcome of either obese/diabetic or taking GLP-1 agonists; something the millions of patients taking GLP-1 receptor drugs should corroborate, but so far do not.

5. There are a few issues with some of the data that need to be fixed - some of which raise serious and broad questions about their approaches:

a. Fig 1J - images are difficult to see, may not be needed given Fig 1 G-I.

b. Fig 2a - this heatmap is odd - how can a gene in both groups be "high" (for example)? Shouldn't it be db/db relative to db/m?

c. Fig 3a-b - not necessary (perhaps suppl) as this conclusion is pretty well established.

d. Fig 3d - why was Bmal1 blot not included (would provide key antibody validation, something that all antibodies could benefit from - either by data or citation at least).

e. Fig 4 - several graph labels are clearly copied from other graphs (and are incorrect here). It appears that Fig 4G (lower left) is identical to the graph as Fig 1I... ? (Big problem if was published this way).

f. Figs 6-7 - there are several parts where the quantification does not even closely approximate the examples shown, casting doubt on your quantification methods throughout. For example, Fig 6I-K, Fig 7B, E.

g. Overall, the vast amount of data are commendable, but the sheer volume of figure panels make this paper very difficult to read as many pieces tend to distract from appreciation of the key findings/data.

6. Most of the changes to the various components under the various treatment are all demonstrated in such a way that they could all easily be 'in parallel', not in the same A-B-C-D pathway. The authors present only a few studies on Bhlhe41 floxed mice (fig 1) or other components to demonstrate that the changes in A are causative to changes in B (etc) and part of the same mechanistic pathway. This is a high bar, but they have reagents in many cases to do this (yet didn't for some reason). There is no functional link (just correlation) in this MS between Bmal1 glycosylation directly impacting Bhlhe41 expression, limited data on Bhlhe41 on driving Dnajb4, and so on. They were very good/outstanding at demonstrating that each component they studied is/could be changed when isolated, but data supporting a functional linkage between pathway components is largely missing.

Referee #2:

In the manuscript entitled "O-GlcNAcylation of circadian clock protein bmal1 impairs cognitive function in diabetes", Hui and colleagues identified bhlhe41 (also known as dec2) as a crucial transcription factor in high glucose-induced cognitive impairment in diabetes.

The further identified that high glucose mediated induction of bhlhe41 in hippocampal neurons depends on the master clock complex formed by clock and bmal1, which activity is enhanced by O-GlcNAcylation of bmal1.

The identification of a major role of Bhlhe41 in diabetes is a novel finding that I believe it would be of interest for the wider readership of The EMBO Journal

In contrast, the involvement of O-GlcNAcylation in regulating clock transcriptional activity has been reported by several groups and it has been shown to affect not only Bmal1 but also additional clock proteins, such as Per2 and Clock.

Overall, the experiments have been performed carefully, the rationale behind the experimental plans are well described, and the data are presented in a clear manner.

The major concerns stem from the lack of data supporting the conclusions presented in both the abstract and the discussion sections on "the crucial role of O-GlcNAcylation in disrupting clock gene expression under high glucose conditions".

Specifically:

1) The master clock complex bmal1/clock plays a major role in regulating the expression of a number of clock controlled genes (CCGs). Although the expression of some CCGs depends on tissue-specific co-activator, many clock controlled genes are ubiquitously regulated by bmal1/clock activity, such as the period (per1, per2, per2) and cryptochrome (cry1, cry2) genes. Thus, in addition to bhlhe41, alteration in bmal1/clock activity should affect the transcription of other CCGs, which has not been explored or reported in the present study.

2) Bhlhe41, but also bhlhe40, have been reported to regulate the circadian rhythm by repressing the activity of bmal1/clock complex (see as an example Honma et al., Nature, 2002). Accordingly, the levels of bhlhe40 and the expression of bmal1/clock target genes should be investigated in hippocampal CA1 neurons of bhlhe41 KO mice.

In addition, ChIP samples used for detecting Bhlhe41 recruitment on Dnajb4 promoter could be used to evaluate the recruitment of Bhlhe41 on bmal1/clock target genes.

3) O-GlcNAcylation has been already reported to regulate circadian clock activity (see as examples, Kaasik et al., Cell Metabolism, 2013 and Ying et al., Plos genetics, 2019). In addition to affect Bmal1, as also previously reported by Ma and colleagues in BBRC (2013), O-GlcNAcylation of additional clock proteins, such as Per2 and Clock, has been reported to have a role in regulating clock gene expression in diverse tissues. This aspect should be at least discussed by the authors.

Referee #3:

- General summary and opinion about the principal significance of the study, its questions, and findings:

In the manuscript entitled 'O-GlcNAcylation of circadian clock protein bmal1 impairs cognitive function in diabetes' Hui, Zhong and colleagues aimed to understand the molecular mechanism underlying the cognitive impairment associated with exposure to high glucose in diabetes. Their data show that in the hippocampal neurons of diabetic mice, upregulated Bhlhe41 expression was associated with excessive mitochondrial calcium accumulation, oxidative damage, and cognitive impairment. The high glucose was also shown to be associated with increased S424 O-GlcNAcylation of the circadian clock protein Bmal1 in hippocampal neurons, leading to the upregulation of BHLHE41 and DNAJB4 expression and to enhanced interaction with its well-known binding partner, CLOCK. Subsequently, the upregulation of BHLHE41 led to upregulation of DNAJB4, which downregulated NCLX, thereby contributing to mitochondrial calcium accumulation and oxidative damage in hippocampal neurons. Finally, the authors showed that GLP-1 activation protected against mitochondrial dysfunction and cognitive impairment.

The study is supported by an extensive amount of work and evidence and provides novel insight into the mechanisms of how high glucose/ diabetes impairs cognitive function in mice. I would not recommend this manuscript for publication in its current form. However, I can see its potential - there are several concerns I would like the authors to address, as I have outlined below. These improvements need to be brought to the revised version.

- Specific major concerns essential to be addressed to support the conclusions

The readability/ formatting of figures in a paper is essential in order to engage the reader and convey the message more comprehensibly. It is abundantly clear that significant amount of work has gone into the generation of the data presented. However, I believed that the readability/ presentation of the data in main figures has plenty of room to be improved upon, as the main figures look very busy and quite chaotic at the moment. I have made specific comments and suggestions for most figures, but the general things to be addressed are as follows:

- Remove panels from main figures which do not add to the story, and which are not expanded upon in the text, or move them into supplementary figures
- Keep a consistent set of font sizes for blot labels, figure legends, etc.
- Label the figures in a consistent manner.
- All of these suggestions should help give the figures a sharper, well put-together aspect and improve their readability

Figure 1:

- The content of Figure 1: A-D provides an example of the workflow used for scRNA-seq in the hippocampus. While this is good to know, it is largely irrelevant to be in the main figure, as no specific conclusion is drawn from the results of either of those panels. This should go in the supplementary.
- While Figure 1E lists 'the 10 most up-regulated and 10 most down-regulated regulon activity of TFs in db/db group', this grouping is not clear to me from the figure. For instance, the colours of the discs indicating average expression of these genes in db/db are not ordered such that the top genes are the highest expressed ones, and vice versa for the bottom ones. Please clarify the grouping as stated in the text.
- The text states: 'As shown by pySCENIC regulon areas under the curve per cell scores, circadian rhythm and mitochondrial damage-related TFs such as Bhlhe41, Trp53, Stat2 and Smad1 exhibited significantly higher transcriptional activity in db/db group than in db/m group.' - It is very unclear to me where these 'pySCENIC regulon areas under the curve per cell scores' are represented. I can see the gene names in Figure 1E, but that figure is very confusing to me, and I cannot come to the conclusion the authors state, when looking at it. Please define what 'Average expression' means and why it goes from 0 to approx. 0.3? Same goes for 'Percent expressed', percent of what. For example, the dot for Stat2 is both bigger and more than in db/m than in db/db, which to me would suggest that Stat2 exhibited higher transcriptional activity in db/m than in db/db, which is opposite to what the authors conclude.

Figure 2:

- The way the heatmap is currently represented makes it hard for me to see by eye that the genes the authors talk about in Figure 2A are indeed differentially regulated. Sure, it is obvious for Plcb1 and Fos, but the colour difference between genotypes for the other genes is not quite convincing. Perhaps representing these data in a different manner would be more useful? For instance, maybe it would be better to not show all the genes being investigated in the main figure, but select fewer of them which the authors focus on, or move the heatmap into supplementary figure 2 and plot the genes associated with oxidative stress response in db/m vs db/db differently, to make the difference in expression more obvious to the reader.

- The axis labels for Figure 2E and F are inconsistent with each other (i.e., one says % of Max, while the other looks at count) even though I can only assume they show the same thing, please make clearer and consistent Y- and X-axis labels.
- While all the bar charts showing the quantification for the FACS plots and the western blots is useful, it makes the main figure unnecessarily busy. The authors could do with moving those in the supplementary figure instead. These changes would make Figure 2 look a lot neater - it has quite a messy look at the moment.
- Similarly, since the authors keep a consistent colour scheme (black for db/m + aav-CAMKII, etc.) throughout the figure, they could do with removing the text lines associated with the figures, after the first one. Or perhaps find a better way of showing what the colours mean, without having to repeat the same text so many times in the figure; it would help with the readability of the figure.

Figure 3:

- Figure 3A looks rather crowded, BHLHE41, ARNTL and CLOCK, although highlighted in red, are quite difficult to read. Similarly, Figure 3B is hard to read, the text being very small. Seeing as these two figures highlight a simple STRING/ ALGGEN search for interactors of Bhlhe41, rather than results of a proteomics screen through which CLOCK/BMAL1 have been highlighted as interactors with these tools, I would recommend moving these two figures into supplementary, or removing them entirely, as they do not add anything to the main figure, especially since the authors then go on to validate these interactions in Figure 3C and onwards.
- I would recommend removing Figure 3L, it does not add anything to the main figure, the authors could stick it in supplementary or remove it entirely and simply mention and reference in the text the online tools they consulted.
- Overall, as I have commented for the previous figures, Figure 3 has an overall rather disorganised look. For instance, the sizes of various blot panels are different from each other, and the alignment of various components of the figure could be improved, to give it a sharper look overall. An additional example is the inconsistent way bar chart annotations are localised: In Figure 3D, they are located top right of the bar chart, in 3F they are located middle top, in 3K they are located under the x-axis, while in 3M and 3N they are located to the right or on the top respectively. I would recommend the authors to pick a specific location and stick to it throughout ALL the figures in the paper. Keeping the font size consistent across would also be ideal. All these modifications would make the figure more readable.
- In the first paragraph of the text corresponding to figure 3, the authors show that BMAL1 overexpression upregulates protein/mRNA of BHLHE41, and vice versa upon knockdown. and concluded that 'high glucose upregulated BHLHE41 expression in neurons through BMAL1/ CLOCK heterodimer.' The conclusion in this current form is not entirely supported by the data, as the authors, up until this point have not shown the requirement of BMAL1 interacting with CLOCK to upregulate BHLHE41 expression, they only showed the requirement BMAL1. Please modify the conclusion to reflect this.
- Several paragraphs following this, the authors state that 'high glucose-induced Bhlhe41 upregulation was mediated by Bmal1 O-GlcNAcylation at its S424 site, which enhanced the binding of Clock to Bmal1 and activates Bhlhe41 transcription'. While the first half of the sentence: 'high glucose-induced Bhlhe41 upregulation was mediated by Bmal1 O-GlcNAcylation at its S424 site' is supported by the data, the second part: 'which enhanced the binding of Clock to Bmal1 and activates Bhlhe41 transcription' is not. The authors only show that BMAL1 overexpression/ knockdown impacts BHLHE41 mRNA and protein levels and that mutation of S424A prevented the accumulation of BHLHE41 in response to BMAL1 O-GlcNAcylation. They give no direct evidence to show that the 'binding of CLOCK to BMAL1 activates Bhlhe41 transcription'. In order to support this kind of statement, the authors need to perform ChIP to show that the whole CLOCK:BMAL1 complex binds the E-boxes in the Bhlhe41 promoter. Similarly, they would need to perform RT-qPCR to show the effect on Bhlhe41 mRNA. Both of these types of experiments should ideally be performed with the WT and S424 BMAL1 mutant the authors described. Only then can the authors state that high glucose-induced Bhlhe41 upregulation was mediated by Bmal1 O-GlcNAcylation at its S424 site, which enhanced the binding of Clock to Bmal1 and activates Bhlhe41 transcription'.

Figure 4:

- Could the authors please clarify whether the neurons they looked at were all hippocampal neurons or hippocampal CA1 neurons specifically, which the investigation originally started with. The same point applies for Figure 3, where the authors mention using 'primary hippocampal neurons'.
- The blots showing that overexpression of Bhlhe41 did not upregulate USP2 and FOSB protein levels are not shown, as the authors claim in the text. Show them please.

Figure 5:

- Figure 5C is not mentioned in the text.
- 'Collectively, these findings suggested that NCLX downregulation induced mitochondrial calcium overload and oxidative damage in hippocampal CA1 neurons, thereby leading to neuronal apoptosis and cognitive dysfunction in diabetes.' It would be more appropriate for the authors to state the effects of NCLX overexpression, as this is what they tested in section of the paper, not the NCLX downregulation.
- Figure 5L is too small to interpret, please make it bigger. A size/scale marker for these images would also be appropriate.
- In figure 5O, how do the authors rationalise the similarity in protein levels between NCLX levels under control and MG-132 levels? One would perhaps a more pronounced stabilisation of NCLX upon proteasomal inhibition. Additionally, how do the authors rationalise the apparently incomplete NCLX protein level rescue upon treatment with MG-132 in the DNAJB overexpression background?

- Minor concerns that should be addressed:

Figure 1:

- Figure S1C is missing the label on the y-axis.
- Figure 1F is missing the label on the y-axis.
- Please briefly explain what the drug streptozotocin is meant to do in the experimental set up used, to help people who are not familiar with it.

Figure 2:

- Please make the x-axis label of Figure 2C easier to understand. 'mitosox' doesn't mean anything to me. 'mitochondrial ROS levels' or similar would be more appropriate.
- It would be useful to mention in the text how Bcl-2, Bax and cleaved Caspase-3 relate to apoptosis, i.e., mention that high Caspase-3 and Bax are associated with increased apoptosis, etc.

Figure 4:

- I would recommend performing some GO analysis on the enriched genes, if possible.

Figure 7:

- Figure The authors might wish to consider highlighting S424 as the major site of O-GlcNAcylation, rather than UDP-GlcNAc dotted on BMAL1 randomly.

- Additional non-essential suggestions for improving the study:

- BMAL1 has also been shown to be a translation factor, such that the observed increase in BHLHE41 protein and mRNA levels could in fact be a consequence of altered translation mediated by BMAL1 (not just transcription), i.e., by enhancing translation and thereby stabilising Bhlhe41 mRNA under high glucose. This would be a highly interesting avenue for the authors to explore.
- Seeing as the authors focus so much on the interaction between CLOCK and BMAL1, which are core circadian clock components, have the authors considered assessing for example testing whether oscillations in mitochondrial calcium accumulation and oxidative damage in hippocampal neurons (if any are present) are affected by high glucose/ O-GlcNAcylation of BMAL? I mention this because otherwise, the purpose of characterising the interaction between CLOCK and BMAL1 in this paper is largely irrelevant (The authors show that BMAL1 overexpression/ knockdown alone impacts BHLHE41 mRNA and protein levels already; the direct impact of the interaction of CLOCK and BMAL1 on the transcription of Bhlhe41 is yet to be

proven, as I mentioned earlier).

** As a service to authors, EMBO Press provides authors with the possibility to transfer a manuscript that one journal cannot offer to publish to another EMBO publication or the open access journal Life Science Alliance launched in partnership between EMBO Press, Rockefeller University Press and Cold Spring Harbor Laboratory Press. The full manuscript and if applicable, reviewers' reports, are automatically sent to the receiving journal to allow for fast handling and a prompt decision on your manuscript. For more details of this service, and to transfer your manuscript please click on Link Not Available. **

Referee 1:

Thank you so much for your efficient and professional review of our research, as well as for providing us with very insightful feedback, especially regarding the lack of causality between the mechanistic pathways you pointed out.

Actually, prior to first submission, we were aware of this research limitation and had already begun addressing the issue. Given that conventional methods for mutating the Bmal1 S424 site in hippocampal neurons of diabetic mice would be time-consuming and labor-intensive, we intervened hippocampal neurons under high glucose conditions both *in vivo* and *in vitro* using artificially synthesized short peptides containing the Bmal1 S424 site and short peptides containing the mutated Bmal1 S424 site. We then assessed downstream effects on the Bhlhe41/Dnajb4/NCLX signaling pathway and related outcome measures, providing robust evidence for our scientific hypothesis. This approach has since been adopted by other researchers for similar *in vivo* studies (Chen, Yuping et al. Cell, PMID: 38128537; Zhu, Qiang et al. Proc Natl Acad Sci U S A, PMID: 36943877), yielding promising experimental results. By employing this method, we not only addressed your primary concern but also significantly shortened our revision cycle.

During this revision, we have addressed all the issues through point-by-point modifications and sincerely hope that our revised manuscript can once again receive your professional guidance.

1. Bhlhe41 (Dec2) is known to be robustly rhythmically expressed by Clock:Bmal1 and the core clockwork, yet no attention is paid to the potential rhythmic nature of Dec2 expression in the hippocampus. Authors first and foremost need to demonstrate that the differences in Dec2 abundance *in vivo* are not due to A) - differences in

timing of tissue collection as a few hours apart could account for many of their in vivo findings, misrepresenting the actual phenotype;

Answer:

This is an excellent suggestion that needs to be addressed.

Because we also considered the rhythmic nature of Bhlhe41 in the early stages of our experiment, we scheduled our sampling time between 8:00 a.m. and 10:00 a.m. In this revised version, we further examined the changes in Bhlhe41 expression from 8:00 a.m. to 10:00 a.m., ensuring the reliability of our experimental conclusions.

We have supplemented the corresponding experiments in the results section and added corresponding descriptions.

Figure S1H Immunoblots and quantification analysis of protein level of Bhlhe41 in hippocampal CA1 neurons of db/m and db/db littermates harvested at 8, 9, and 10 am (n = 3 mice per group).

Manuscript page 5, lines 14-18: “To demonstrate that the differences in Bhlhe41 abundance in vivo are not due to the differences in timing of tissue collection, hippocampal CA1 neurons from all groups of mice were collected from 8am to 10am, additionally, we determined CA1 Bhlhe41 expression in db/m and db/db littermates at 8am, 9am and 10am, respectively, and did not find any within-group differences (Figure S1H).”

2. If this was all due to differential regulation of BMAL1 function as the authors propose, there would be wide-spread alterations in circadian function well beyond elevated expression of Bhlhe41 (Dec2). These were not seen in scRNAseq data (Fig 1), nor demonstrated or alluded to anywhere. This is a substantial issue as the authors propose that elevated glucose increase O-GlcNac of BMAL1, a modification that appears to increase its binding to CLOCK (Fig 3) to drive downstream gene expression - if this is truly the case, expression of many more CLOCK:BMAL1 targets should also be directly altered; data in this regard is the first essential piece needed to provide evidence that BMAL1 is at all involved here.

Answer:

Thank you very much for raising this important issue. In this revised manuscript, we addressed it by respectively examining the expression changes of the Bhlhe40, per1, per2, per3, cry1, and cry2 genes under high-glucose conditions using single-cell sequencing data and in vivo and in vitro studies. We also provided a detailed discussion of the findings from these analyses.

Figure S4 High Glucose Increased the Expression of Clock Controlled Genes Regulated by Bmal1/Clock Heterodimer in Hippocampal Neurons

(A): Box plots showing gene expression of Bhlhe40, Cryptochrome (Cry1, Cry2) and Period (Per1, Per2, Per3) in hippocampal neurons of db/m and db/db mice identified by scRNA-seq.

(B): mRNA levels of Bhlhe40, Cryptochrome (Cry1, Cry2) and Period (Per1, Per2, Per3) in primary hippocampal neurons treated with normal glucose (NG, 5.5 mmol/L D-glucose) or high glucose (HG, 25 mmol/L D-glucose) for 48 h (n = 3 replicates).

(C): Binding of Bhlhe41 to the Bhlhe40, Cryptochrome (Cry1, Cry2) and Period (Per1, Per2, Per3) promoter in primary hippocampal neurons transfected with control plasmid or a plasmid overexpressing Bhlhe41 for 48h. IgG was used as negative CHIP control (n = 3 replicates).

(D): mRNA levels of Bhlhe40, Cry1 and Per2 in hippocampal CA1 neurons of Bhlhe41^{fl/fl};db/m, Bhlhe41^{fl/fl};db/db and Bhlhe41^{CKO};db/db mice (n = 6 mice per group). (n = 6 mice per group).

Manuscript page 9, lines 20-26—Manuscript page 10, lines 1-3: “Since Bmal1/Clock heterodimer plays a major role in regulating the expression of a number of clock controlled genes, such as Bhlhe40, period (per1, per2, per3) and cryptochrome (cry1, cry2), we therefore examined the expression of these genes under high glucose conditions. scRNA-seq analysis revealed that all these genes expression were significantly increased in the hippocampal neurons of db/db mice (Figure S4A). In addition, we found a significant increase in Bhlhe40, cry1 and per2 mRNA expression in mouse primary hippocampal neurons treated with high glucose (Figure S4B). Interestingly, Chromatin immunoprecipitation (ChIP) assays further confirmed that Bhlhe41 significantly bound to the promoter of Bhlhe40, per1, per2, per3, cry1 and cry2 (Figure S4C), and Bhlhe41 knockout further upregulated Bhlhe40 mRNA expression in the hippocampal CA1 neurons of diabetic mouse, whereas cry1 and per2 expression were not significantly affected (Figure S4D).”

Manuscript page 14, lines 17-26—Manuscript page 15, lines 1-17: “Bmal1/Clock complex have been reported to regulate the expression of a number of clock controlled genes (Fagiani et al, 2022). In addition to Bhlhe41, our in vitro results showed that Bhlhe40, cry1 and per2 mRNA expression were also upregulated in

hippocampal neurons under high glucose conditions, whereas the expression of *cry2*, *per1* and *per3* were not significantly affected. This result might be partly attributed to different degrees of repression on Bmal1/Clock complex-mediated transactivation exerted by clock controlled genes themselves. Previous studies have found that *Bhlhe*, *Per* and *Cry* genes are activated by Bmal1/Clock complex and repressed by their own products through CACGTG E-boxes in their promoter regions (Honma et al, 2002; Kawamoto et al, 2004). All these gene products have been proved to functions as transcriptional repressors by binding to the Bmal1/Clock complex. However, only *Bhlhe40* and *Bhlhe41* have the DNA-binding capacity and can directly compete with Bmal1/Clock complex for CACGTG E-box binding in promoter regions (Hamaguchi et al., 2004). We hypothesize that O-GlcNAcylation modification of Bmal1 might reduce the negative feedback inhibition caused by the binding of clock-controlled gene products to the Bmal1/Clock complex, thereby upregulating the expression of *Bhlhe40*, *Bhlhe41*, *cry1*, and *per2*. The upregulated *Bhlhe40* and *Bhlhe41* may further bind directly to the promoter regions of *cry2*, *per1*, and *per3* with stronger affinity, bypassing the inhibitory effect mediated by O-GlcNAcylation modification of Bmal1 on the binding of clock-controlled gene products and the Bmal1/Clock complex, thereby inhibiting the upregulation of the expression of these three genes under high-glucose conditions. However, *Bhlhe40* and *Bhlhe41* might have relatively weak binding ability to the promoter regions of *Bhlhe40*, *Bhlhe41*, *cry1*, and *per2*, thus they may not effectively inhibit the upregulation of these four genes under high-glucose conditions. The differential binding ability of *Bhlhe40* and *Bhlhe41* to the promoter regions of different clock-controlled genes and their inhibitory effects on expression remain to be elucidated in future studies. Consistent with previous reports (Kreslavsky et al, 2017), our data revealed significant recruitment of *Bhlhe41* to *Bhlhe40*, *per1*,

per2, per3, cry1 and cry2 promoter. More interestingly, diabetic mice lacking Bhlhe41 expression in hippocampal CA1 neurons exhibited up-regulation of Bhlhe40 and showed no significant changes in the expression patterns of cry1 and per2, similar results were also obtained in study examining the effects of Bhlhe40 deletion on Bhlhe41 expression (Grechez-Cassiau et al, 2004), indicating that Bhlhe41 can compensate for the function of Bhlhe40 and vice versa.”

3. The authors provide no direct evidence that O-GlcNAcylation of BMAL1 directly impairs cognitive function (as indicated in the title). They show that O-GlcNAcylation of BMAL1 can happen in a manner coincident with diabetic cognitive decline, but there is no substantial indication that it actually plays a role - many proteins are regulated by O-GlcNAcylation/high glucose. The title must be changed at a minimum.

Answer:

We greatly appreciate the valuable suggestion provided by the reviewers. In this revised manuscript, we employed a short peptide containing the Bmal1 S424 site and a control peptide containing the mutated Bmal1 S424 site as intervention tools. Through comprehensive in vivo and in vitro validation, we confirmed that the high-glucose-induced glycosylation modification of the Bmal1 S424 site regulates the Bhlhe41/Dnajb4/NCLX signaling pathway by forming a dimer with Clock, ultimately leading to mitochondrial calcium overload, hippocampal neuron apoptosis, and cognitive dysfunction in diabetes.

Figure 6. Short Peptide S424-pe Protects Against High Glucose-Induced Neuronal Apoptosis and Cognitive Dysfunction by Regulating Bhlhe41/Dnajb4/NCLX Signaling

(A): Schematic illustration of designed peptides. Blue: cell-penetrating peptide (CPP).

Red: site for Bmal1 O-GlcNAcylation.

(B): Primary hippocampal neurons were treated with five short peptides (20 μ M) in the presence of high glucose (HG, 25 mmol/L D-glucose), and Bmal1 O-GlcNAcylation was determined by immunoprecipitation and western blot using the indicated antibodies. Three independent experiments were performed.

(C-D): Hippocampal CA1 neurons were isolated from db/m, db/db (C), Ctr or STZ mice (D) treated with vehicle, S424A-pe (5 mg/kg) or S424-pe (5 mg/kg), and cell lysates were immunoprecipitated with Bmal1 antibodies and western blotted with the indicated antibodies (n = 6 mice per group).

(E-H): Immunoblots and quantification analysis of Bhlhe41, Dnajb4, NCLX (E-F), Bax, Bcl-2 and Cleaved-caspase3 (G-H) expression levels in the hippocampal CA1 neurons of db/m, db/db, Ctr or STZ mice treated with vehicle, S424A-pe (5 mg/kg) or S424-pe (5 mg/kg) (n = 6 mice per group).

(I-L): Flow cytometry analysis of mitochondrial calcium levels (I and K) and mtROS levels (J and L) in the hippocampal CA1 neurons isolated from different groups of mice (n = 6 mice per group).

Data are means \pm SEM. *P < 0.05, **P < 0.01, ***P < 0.001. Two-way ANOVA followed by Tukey's test (C-L).

Figure 7. Short Peptide S424-pe Protects Against High Glucose-Induced Neuronal Apoptosis and Cognitive Dysfunction by Regulating Bhlhe41/Dnajb4/NCLX Signaling

(A-D): Representative track images and escape latency to the platform during the

training trials, target entries and time spent in target quadrant in the probe trial of Morris water maze (A-B), and active avoidance performance (C-D) of db/m, db/db, Ctr, STZ mice treated with vehicle, S424A-pe (5 mg/kg) or S424-pe (5 mg/kg) (n = 6 mice per group).

Data are means \pm SEM. *P < 0.05, **P < 0.01, ***P < 0.001. Two-way ANOVA followed by Tukey's test (A-D).

Figure S6. Short Peptide S424-pe Protects Against High Glucose-Induced Neuronal Apoptosis and Cognitive Dysfunction by Regulating Bhlhe41/Dnajb4/NCLX Signaling

(A): Primary hippocampal neurons were treated with normal glucose (NG, 5.5 mmol/L D-glucose) or high glucose (HG, 25 mmol/L D-glucose) in the presence of

vehicle, S424A-pe (20 μ M) or S424-pe (20 μ M), and cell lysates were immunoprecipitated with Bmal1 antibodies and western blotted with the indicated antibodies (n = 3 replicates).

(B): Immunoblots and quantification analysis of Bhlhe41, Dnajb4 and NCLX expression levels in primary hippocampal neurons treated with normal glucose (NG, 5.5 mmol/L D-glucose) or high glucose (HG, 25 mmol/L D-glucose) in the presence of vehicle, S424A-pe (20 μ M) or S424-pe (20 μ M) (n = 3 replicates).

(C-E): Flow cytometry analysis of mitochondrial calcium levels (C), mtROS levels (D) and apoptosis levels (E) in primary hippocampal neurons with different treatments as indicated (n = 3 replicates).

Data are means \pm SEM. *P < 0.05, **P < 0.01, ***P < 0.001. Two-way ANOVA followed by Tukey's test (A-E).

Manuscript page 12, lines 17-26—Manuscript page 13, lines 1-10: “To provide direct evidence that O-GlcNAcylation of Bmal1 at S424 impairs cognitive function through the Bhlhe41/Dnajb4/NCLX signaling pathway, we developed a short peptide designed to specifically target the O-GlcNAcylation of Bmal1 at S424. After analyzing the sequence surrounding the Bmal1 S424 site, we synthesized five short peptides fused with a cell-penetrating peptide, rendering them capable of permeating cell membranes and crossing the blood-brain barrier (Qin et al, 2019). This approach has emerged as one of the most popular and effective methods for intracellular delivery of biomolecules. (Koren et al, 2011; Li et al, 2021) (Figure 6A). Among these peptides, S424-peptide-2# (referred to as S424-pe) exhibited the strongest inhibitory effect on Bmal1 O-GlcNAcylation. (Figure 6B). Thus, based on the S424-pe peptide, we generated a scrambled peptide denoted as S424A-pe, in which the S424 amino acid was replaced with an A residue. As shown in Figure 6C and D, S424-pe, but not S424A-pe, significantly reversed high glucose-induced Bmal1

O-GlcNAcylation in hippocampal CA1 neurons isolated from db/db (Figure 6C) and STZ-induced diabetic mice (Figure 6D), accompanied by a decrease in binding of Clock to Bmal1 and a downregulation of Bhlhe41 and Dnajb4 expression (Figure 6E-F). Furthermore, S424-pe significantly upregulated NCLX expression (Figure 6E-F) and reduced mitochondrial calcium concentrations (Figure 6I and K) and oxidative damage (Figure 6J and L), followed by a decrease in neuronal apoptosis (Figure 6G-H) and an improvement in cognitive function (Figure 7A-D). Similar results were also obtained in mouse primary hippocampal neurons treated with high glucose (Figure S6A-E). Collectively, our results demonstrated that targeting Bmal1 S424 O-GlcNAcylation with S424-pe may alleviate high glucose-induced mitochondrial calcium overload and oxidative damage in hippocampal CA1 neurons and thereby improve cognitive function in diabetes.”

Manuscript page 16, lines 8-23: “Small-molecule inhibitors targeting OGT have been created to explore the pathogenic role of O-GlcNAcylation modifications in a variety of diseases (Rohlf et al, 1999; Sharma et al, 2019). However, as OGT catalyzes O-GlcNAcylation on a wide range of protein substrates and plays a crucial role in maintaining normal physiological functions, the inhibition of OGT is likely to result in adverse side effects both in vivo and in vitro (Zhu et al, 2023). To overcome this limitation, we designed an alternative strategy that employs a short peptide containing Bmal1 O-GlcNAcylation site to competitively inhibit the O-GlcNAcylation of Bmal1 S424 with minimal interference with the total cellular O-GlcNAcylation level. Such treatment led to a decrease in binding of Clock to Bmal1 and a downregulation of Bhlhe41 and Dnajb4 expression, resulting in NCLX upregulation and an improvement in mitochondrial calcium overload, hippocampal neuron dysfunction, and cognitive impairment in diabetes. Our data suggest that

oscillations in mitochondrial calcium accumulation and oxidative damage in hippocampal neurons are indeed affected by O-GlcNAcylation of Bmal1, in addition, it provides solid evidence that Bmal1 O-GlcNAcylation directly impairs cognitive function in diabetes. Short peptide has been used in a recent study to elucidate the role of O-GlcNAcylation modification in tumor immune evasion (Zhu et al., 2023). Our study further confirmed that this strategy targeting site-specific O-GlcNAcylation on Bmal1 in hippocampal neurons may serve as a novel therapeutic approach for the treatment of cognitive impairment in diabetes.”

Manuscript page 27, lines 4-9: “The peptides utilized in this study were synthesized by Guoping Pharmaceutic Inc. Synthetic peptides underwent purification via high-pressure liquid chromatography to achieve a purity exceeding 98%, suitable for both in vitro and in vivo applications. Peptides were dissolved in phosphate-buffered saline (PBS) to create a 20 mM stock solution for in vitro experiments. For in vivo use, S424-pe and S424A-pe were dissolved in PBS and stored on ice prior to injection. Prior to injection, the solution was allowed to reach room temperature.”

4. There is no direct evidence provided that demonstrate O-GlcNAcylation of Bmal1 changes its transcriptional activity/function here, short of a potential decrease in binding with CLOCK. They did find the key residue in BMAL1 that is o-glcnaacylated, but did not provide any functional studies indicating that mutating this site alters BMAL1 function. Likewise, there were no experiments demonstrating that glycosylation of Bmal1 is functionally involved in regulation of Bhlhe41 expression, in any form. This is a critical omission, and very surprising given the other detailed studies provided throughout.

Answer:

We employed two intervention methods, the S424A mutant and a short peptide, and combined them with three techniques: COIP, CHIP, and qPCR. Through these approaches, we confirmed that the glycosylation modification of the Bmal1 S424 site induced by high glucose promotes its binding to Clock, consequently enhancing its binding in the Bhlhe41 promoter region and ultimately upregulating the expression of Bhlhe41.

Figure 3K Primary hippocampal neurons, which were stably silenced endogenous Bmal1 and expressed WT Bmal1 or S424A Bmal1, were treated with normal glucose (NG, 5.5 mmol/L D-glucose) or high glucose (HG, 25 mmol/L D-glucose). Cell lysates were immunoprecipitated with Bmal1 antibodies and western blotted with the indicated antibodies.

Figure S3K mRNA levels of Bhlhe41 in primary hippocampal neurons with endogenous Bmal1 stably silenced and expressing either wild-type (WT) Bmal1 or S424A Bmal1, treated with normal glucose (NG, 5.5 mmol/L D-glucose) or high

glucose (HG, 25 mmol/L D-glucose) (n = 3 replicates).

Figure 3L Binding of Bmal1 to the Bhlhe41 promoter in primary hippocampal neurons with endogenous Bmal1 stably silenced and expressing either wild-type (WT) Bmal1 or S424A Bmal1, treated with normal glucose (NG, 5.5 mmol/L D-glucose) or high glucose (HG, 25 mmol/L D-glucose). IgG was used as negative CHIP control (n = 3 replicates).

Figure 6C-D Hippocampal CA1 neurons were isolated from db/m, db/db (C), Ctr or STZ mice (D) treated with vehicle, S424A-pe (5 mg/kg) or S424-pe (5 mg/kg), and cell lysates were immunoprecipitated with Bmal1 antibodies and western blotted with the indicated antibodies (n = 6 mice per group).

Figure 6E-F Immunoblots and quantification analysis of Bhlhe41, Dnajb4 and NCLX expression levels in the hippocampal CA1 neurons of db/m, db/db (E), Ctr or STZ mice (F) treated with vehicle, S424A-pe (5 mg/kg) or S424-pe (5 mg/kg) (n = 6 mice per group).

Figure S6A-B

(A): Primary hippocampal neurons were treated with normal glucose (NG, 5.5 mmol/L D-glucose) or high glucose (HG, 25 mmol/L D-glucose) in the presence of vehicle, S424A-pe (20 μ M) or S424-pe (20 μ M), and cell lysates were immunoprecipitated with Bmal1 antibodies and western blotted with the indicated antibodies (n = 3 replicates).

(B): Immunoblots and quantification analysis of Bhlhe41, Dnajb4 and NCLX expression levels in primary hippocampal neurons treated with normal glucose (NG,

5.5 mmol/L D-glucose) or high glucose (HG, 25 mmol/L D-glucose) in the presence of vehicle, S424A-pe (20 μ M) or S424-pe (20 μ M) (n = 3 replicates).

Manuscript page 9, lines 9-14: “Consistently, S424A mutant significantly reversed high glucose-induced Bmal1 O-GlcNAcylation in mouse primary hippocampal neurons with endogenous Bmal1 knockdown, accompanied by a decrease in binding with Clock and a downregulation of the protein and mRNA levels of Bhlhe41 (Figure 3K and S3J-K). Additionally, Chromatin immunoprecipitation analysis showed enhanced binding of Bmal1 to the Bhlhe41 promoter under high glucose conditions and S424A mutant treatment weakened this binding (Figure 3L).”

Manuscript page 12, lines 24-26—Manuscript page 13, lines 1-3: “Thus, based on the S424-pe peptide, we generated a scrambled peptide denoted as S424A-pe, in which the S424 amino acid was replaced with an A residue. As shown in Figure 6C and D, S424-pe, but not S424A-pe, significantly reversed high glucose-induced Bmal1 O-GlcNAcylation in hippocampal CA1 neurons isolated from db/db (Figure 6C) and STZ-induced diabetic mice (Figure 6D), accompanied by a decrease in binding of Clock to Bmal1 and a downregulation of Bhlhe41 and Dnajb4 expression (Figure 6E-F).”

Manuscript page 13, lines 6-7: “Similar results were also obtained in mouse primary hippocampal neurons treated with high glucose (Figure S6A-E).”

5. According to the mechanism shown in Fig 8, and the GLP-1 agonist studies, if these were interacting through correcting or altering BMAL1 activity, then obese/diabetic patients taking these would have substantial alterations in circadian function and sleep regulation. The point is that if BMAL1 was a key mediator here as

the authors propose, broad circadian dysfunction would be a highly likely outcome of either obese/diabetic or taking GLP-1 agonists; something the millions of patients taking GLP-1 receptor drugs should corroborate, but so far do not.

Answer:

Thank you very much for pointing out this issue.

According to previous literature reports, high glucose environments indeed tend to induce circadian rhythm disturbances in diabetic patients. GLP-1 receptor agonists have been proven to ameliorate circadian rhythm disturbances in diabetic patients. Therefore, we believe that high glucose can induce circadian rhythm disturbances in organisms through glycosylation modification of Bmal1. GLP-1 receptor agonists may potentially improve the disrupted circadian rhythms in diabetic patients by reducing the glycosylation levels of Bmal1.

In order to better highlight the focus of our research and the causal relationships between mechanistic pathways, this revision has removed all content related to the treatment of diabetes-associated cognitive dysfunction with GLP-1 receptor agonists. Instead, it has been entirely replaced by the study of the effects and molecular mechanisms of short peptide treatment on diabetes-associated cognitive dysfunction. However, we have still added in the discussion section the phenomenon of high glucose-induced circadian rhythm disturbances in diabetic patients and its potential mechanisms.

Manuscript page 15, lines 17-21: “Notably, previous studies have already demonstrated that both type 1 and type 2 diabetic patients were highly susceptible to

disrupted circadian rhythms, such as flattened circadian cortisol profile, earlier dim-light melatonin onset and sleep disorders (Peng et al, 2022; Reutrakul et al, 2016). Our findings also provided a potential mechanism for this phenomenon.”

6. There are a few issues with some of the data that need to be fixed - some of which raise serious and broad questions about their approaches:

a. Fig 1J - images are difficult to see, may not be needed given Fig 1 G-I.

Answer:

It has been deleted as requested.

b. Fig 2a - this heatmap is odd - how can a gene in both groups be "high" (for example)? Shouldn't it be db/db relative to db/m?

Answer:

Thank you very much for pointing out this issue. In this revised manuscript, we have employed another bioinformatics analysis method to better present the relevant data.

Figure 2B Radar diagram showing differentially expressed genes related to calcium signaling pathway, oxidative stress response, and apoptosis.

c. Fig 3a-b - not necessary (perhaps suppl) as this conclusion is pretty well established.

Answer:

We have moved these two figures into the supplementary material.

d. Fig 3d - why was Bmal1 blot not included (would provide key antibody validation, something that all antibodies could benefit from - either by data or citation at least).

Answer:

Bmal1 blot have been added in **Fig 3d** as requested.

Figure 3B Immunoblots and quantification analysis of protein levels of Bmal1 and Bhlhe41 in primary hippocampal neurons treated with Bmal1 siRNA or control siRNA in the presence of normal glucose (NG, 5.5 mmol/L D-glucose) or high glucose (HG, 25 mmol/L D-glucose) (n = 3 replicates).

e. Fig 4 - several graph labels are clearly copied from other graphs (and are incorrect here). It appears that Fig 4G (lower left) is identical to the graph as Fig 1I... ? (Big problem if was published this way).

Answer:

After careful examination and review, we believe that it may have been our inappropriate way of describing that led to this issue. We aim to demonstrate through Fig 1I that we have successfully achieved specific knockout of Bhlhe41 expression in hippocampal neurons of diabetic mice. Additionally, we aim to prove through Fig 4G that the specific knockout of Bhlhe41 expression in hippocampal neurons of diabetic mice can downregulate the expression of its downstream protein Dnajb4, thereby

confirming causal relationships between different parts of the mechanistic pathway. Therefore, in this revision, we did not modify the graph labels in Fig 4G. If we have misunderstood the reviewers' editing suggestions, we kindly request further professional guidance from our reviewer.

f. Figs 6-7 - there are several parts where the quantification does not even closely approximate the examples shown, casting doubt on your quantification methods throughout. For example, Fig 6I-K, Fig 7B, E.

Answer:

Although we have removed all the research content from Figures 6-7 in this revision, we have made the aforementioned corrections in similar experimental results based on your valuable feedback.

g. Overall, the vast amount of data are commendable, but the sheer volume of figure panels make this paper very difficult to read as many pieces tend to distract from appreciation of the key findings/data.

Answer:

Thank you very much for the professional advice on improving the aesthetics and conciseness of our figures. Based on your suggestions, we have thoroughly revised all the main figures and supplementary figures. We have moved some results to the supplementary figures, simplified and standardized the annotations of many result figures, and optimized the size and placement of some subfigures.

If you feel that there are any areas in our main figures and supplementary figures that still need further modification, we will thoroughly revise them according to your

requirements in subsequent revisions.

7. Most of the changes to the various components under the various treatment are all demonstrated in such a way that they could all easily be 'in parallel', not in the same A-B-C-D pathway.

The authors present only a few studies on Bhlhe41 floxed mice (fig 1) or other components to demonstrate that the changes in A are causative to changes in B (etc) and part of the same mechanistic pathway.

This is a high bar, but they have reagents in many cases to do this (yet didn't for some reason). There is no functional link (just correlation) in this MS between Bmal1 glycosylation directly impacting Bhlhe41 expression, limited data on Bhlhe41 on driving Dnajb4, and so on. They were very good/outstanding at demonstrating that each component they studied is/could be changed when isolated, but data supporting a functional linkage between pathway components is largely missing.

Answer:

Thank you very much for the suggestion regarding the importance of establishing causal relationships in mechanistic studies.

In this revision, we addressed this crucial issue by using a short peptide as described above. All relevant research results are presented in Figure 6-7 and Supplementary Figure 6.

This suggestion not only significantly enhances the quality and readability of our study but also helps us avoid similar mistakes in future research.

Once again, we sincerely appreciate your valuable time in providing such professional feedback. We have learned a lot from this revision and eagerly look forward to your review of our revised manuscript and your valuable insights.

Referee 2:

Thank you very much for your recognition of our research direction and novelty.

We sincerely appreciate the time you devoted to reviewing our manuscript and providing valuable feedback. Your review not only greatly improves the quality of this study but also further guides us to consider other factors closely related to pathogenicity and their potential mechanisms in future research.

In response to your review, we have supplemented all the necessary experimental content and provided detailed descriptions and discussions. We eagerly look forward to receiving your professional review and guidance again.

1. The master clock complex *bmal1/clock* plays a major role in regulating the expression of a number of clock controlled genes (CCGs). Although the expression of some CCGs depends on tissue-specific co-activator, many clock controlled genes are ubiquitously regulated by *bmal1/clock* activity, such as the period (*per1*, *per2*, *per3*) and cryptochrome (*cry1*, *cry2*) genes. Thus, in addition to *bhlhe41*, alteration in *bmal1/clock* activity should affect the transcription of other CCGs, which has not

been explored or reported in the present study.

Answer:

Thank you very much for raising this issue. We further investigated the changes in the genes *Bhlhe40*, *per1*, *per2*, *per3*, *cry1*, and *cry2* under high-glucose conditions through single-cell sequencing data and in vivo and in vitro experiments. We also provided detailed descriptions and discussions on these findings.

Figure S4A-B

(A): Box plots showing gene expression of *Bhlhe40*, Cryptochrome (*Cry1*, *Cry2*) and Period (*Per1*, *Per2*, *Per3*) in hippocampal neurons of *db/m* and *db/db* mice identified by scRNA-seq.

(B): mRNA levels of *Bhlhe40*, Cryptochrome (*Cry1*, *Cry2*) and Period (*Per1*, *Per2*, *Per3*) in primary hippocampal neurons treated with normal glucose (NG, 5.5 mmol/L D-glucose) or high glucose (HG, 25 mmol/L D-glucose) for 48 h ($n = 3$ replicates).

Manuscript page 9, lines 20-25: “Since Bmal1/Clock heterodimer plays a major role in regulating the expression of a number of clock controlled genes, such as *Bhlhe40*, period (*per1*, *per2*, *per3*) and cryptochrome (*cry1*, *cry2*), we therefore examined the

expression of these genes under high glucose conditions. scRNA-seq analysis revealed that all these genes expression were significantly increased in the hippocampal neurons of db/db mice (Figure S4A). In addition, we found a significant increase in *Bhlhe40*, *cry1* and *per2* mRNA expression in mouse primary hippocampal neurons treated with high glucose (Figure S4B).”

Manuscript page 14, lines 17-26—Manuscript page 15, lines 1-11: “*Bmal1*/Clock complex have been reported to regulate the expression of a number of clock controlled genes (Fagiani et al, 2022). In addition to *Bhlhe41*, our in vitro results showed that *Bhlhe40*, *cry1* and *per2* mRNA expression were also upregulated in hippocampal neurons under high glucose conditions, whereas the expression of *cry2*, *per1* and *per3* were not significantly affected. This result might be partly attributed to different degrees of repression on *Bmal1*/Clock complex-mediated transactivation exerted by clock controlled genes themselves. Previous studies have found that *Bhlhe*, *Per* and *Cry* genes are activated by *Bmal1*/Clock complex and repressed by their own products through CACGTG E-boxes in their promoter regions (Honma et al, 2002; Kawamoto et al, 2004). All these gene products have been proved to functions as transcriptional repressors by binding to the *Bmal1*/Clock complex. However, only *Bhlhe40* and *Bhlhe41* have the DNA-binding capacity and can directly compete with *Bmal1*/Clock complex for CACGTG E-box binding in promoter regions (Hamaguchi et al., 2004). We hypothesize that O-GlcNAcylation modification of *Bmal1* might reduce the negative feedback inhibition caused by the binding of clock-controlled gene products to the *Bmal1*/Clock complex, thereby upregulating the expression of *Bhlhe40*, *Bhlhe41*, *cry1*, and *per2*. The upregulated *Bhlhe40* and *Bhlhe41* may further bind directly to the promoter regions of *cry2*, *per1*, and *per3* with stronger affinity, bypassing the inhibitory effect mediated by O-GlcNAcylation modification of *Bmal1*

on the binding of clock-controlled gene products and the Bmal1/Clock complex, thereby inhibiting the upregulation of the expression of these three genes under high-glucose conditions. However, Bhlhe40 and Bhlhe41 might have relatively weak binding ability to the promoter regions of Bhlhe40, Bhlhe41, cry1, and per2, thus they may not effectively inhibit the upregulation of these four genes under high-glucose conditions. The differential binding ability of Bhlhe40 and Bhlhe41 to the promoter regions of different clock-controlled genes and their inhibitory effects on expression remain to be elucidated in future studies.”

2. Bhlhe41, but also bhlhe40, have been reported to regulate the circadian rhythm by repressing the activity of bmal1/clock complex (see as an example Honma et al., Nature, 2002). Accordingly, the levels of bhlhe40 and the expression of bmal1/clock target genes should be investigated in hippocampal CA1 neurons of bhlhe41 KO mice.

In addition, ChIP samples used for detecting Bhlhe41 recruitment on Dnajb4 promoter could be used to evaluate the recruitment of Bhlhe41 on bmal1/clock target genes.

Answer:

Thank you very much for providing this professional suggestion. We have completed all the experiments that needed to be supplemented according to your requirements.

C

CHIP:Bhlhe41

D

Figure S4C-D

(C): Binding of Bhlhe41 to the Bhlhe40, Cryptochrome (Cry1, Cry2) and Period (Per1, Per2, Per3) promoter in primary hippocampal neurons transfected with control plasmid or a plasmid overexpressing Bhlhe41 for 48h. IgG was used as negative CHIP control (n = 3 replicates).

(D): mRNA levels of Bhlhe40, Cry1 and Per2 in hippocampal CA1 neurons of Bhlhe41^{fl/fl};db/m, Bhlhe41^{fl/fl};db/db and Bhlhe41^{cKO};db/db mice (n = 6 mice per group). (n = 6 mice per group).

Manuscript page 9, lines 25-26—Manuscript page 10, lines 1-3: “Interestingly, Chromatin immunoprecipitation (ChIP) assays further confirmed that Bhlhe41 significantly bound to the promoter of Bhlhe40, per1, per2, per3, cry1 and cry2 (Figure S4C), and Bhlhe41 knockout further upregulated Bhlhe40 mRNA expression in the hippocampal CA1 neurons of diabetic mouse, whereas cry1 and per2

expression were not significantly affected (Figure S4D).”

Manuscript page 15, lines 12-17: “Consistent with previous reports (Kreslavsky et al, 2017), our data revealed significant recruitment of Bhlhe41 to Bhlhe40, per1, per2, per3, cry1 and cry2 promoter. More interestingly, diabetic mice lacking Bhlhe41 expression in hippocampal CA1 neurons exhibited up-regulation of Bhlhe40 and showed no significant changes in the expression patterns of cry1 and per2, similar results were also obtained in study examining the effects of Bhlhe40 deletion on Bhlhe41 expression (Grechez-Cassiau et al, 2004), indicating that Bhlhe41 can compensate for the function of Bhlhe40 and vice versa.”

3. O-GlcNAcylation has been already reported to regulate circadian clock activity (see as examples, Kaasik et al., Cell Metabolsim, 2013 and Ying et al., Plos genetics, 2019). In addition to affect Bmal1, as also previously reported by Ma and colleagues in BBRC (2013), O-GlcNAcylation of additional clock proteins, such as Per2 and Clock, has been reported to have a role in regulating clock gene expression in diverse tissues. This aspect should be at least discussed by the authors.

Answer:

Thank you very much for your valuable feedback. We have incorporated the relevant content and corresponding references into the discussion section.

Manuscript page 17, lines 5-11: “Second, O-GlcNAcylation has been already reported to regulate circadian clock activity (Kaasik et al, 2013; Li et al, 2019). In addition to affecting Bmal1, as also previously reported by Ma and colleagues (Ma et al, 2013), O-GlcNAcylation of additional clock proteins, such as Per2 and Clock, has been reported to have a role in regulating clock gene expression in diverse tissues, further investigation is warranted to determine if these proteins are glycosylated in

hippocampal neurons and if they play a pathogenetic role in cognitive impairment in diabetes.”

Referee 3:

Heartfelt thanks for your recognition of the novelty of our research.

We sincerely appreciate your valuable time spent reviewing our paper and providing highly professional feedback. Your insightful comments not only significantly enhance the quality of our study but also encourage us to prioritize the rationality of experimental design and the aesthetics and conciseness of figure presentations in future research.

In this revision, we have made corresponding modifications to the size, labeling, font, layout, and placement of all figures based on your suggestions. Additionally, we have completed all necessary supplementary experiments and provided detailed descriptions and discussions of the experimental results. We eagerly look forward to your review of our revised manuscript and your professional guidance once again.

1. The content of Figure 1: A-D provides an example of the workflow used for scRNA-seq in the hippocampus. While this is good to know, it is largely irrelevant to be in the main figure, as no specific conclusion is drawn from the results of either of those panels. This should go in the supplementary.

Answer:

Thank you so much for your thoughtful suggestions. Figure 1: A-D have been moved into supplementary figure1.

While Figure 1E lists 'the 10 most up-regulated and 10 most down-regulated regulon activity of TFs in db/db group', this grouping is not clear to me from the figure. For instance, the colours of the discs indicating average expression of these genes in db/db are not ordered such that the top genes are the highest expressed ones, and vice versa for the bottom ones. Please clarify the grouping as stated in the text.

Answer:

In this revision, we have employed another bioinformatics analysis method to present our results more clearly and logically.

Figure 1A Differential regulon activity analysis showing the 5 most up- and down-regulated regulons in hippocampal neurons of db/db mice (left). Gene expression levels of the 5 most up- (upper right) and down-regulated (lower right) regulons.

The text states: 'As shown by pySCENIC regulon areas under the curve per cell scores, circadian rhythm and mitochondrial damage-related TFs such as Bhlhe41, Trp53, Stat2 and Smad1 exhibited significantly higher transcriptional activity in db/db group than in db/m group.' - It is very unclear to me where these 'pySCENIC regulon areas under the curve per cell scores' are represented. I can see the gene names in Figure 1E, but that figure is very confusing to me, and I cannot come to the conclusion the authors state, when looking at it. Please define what 'Average expression' means and why it goes from 0 to approx. 0.3? Same goes for 'Percent expressed', percent of what. For example, the dot for Stat2 is both bigger and more that in db/m than in db/db, which to me would suggest that Stat2 exhibited higher transcriptional activity in db/m than in db/db, which is opposite to what the authors conclude.

Answer:

Thank you for the careful reminder. This time, we have employed a different bioinformatics analysis method to demonstrate the changes in transcription factor transcriptional activity under high-glucose conditions (Figure 1A).

2. The way the heatmap is currently represented makes it hard for me to see by eye that the genes the authors talk about in Figure 2A are indeed differentially regulated. Sure, it is obvious for Plcb1 and Fos, but the colour difference between genotypes for the other genes is not quite convincing. Perhaps representing these data in a different manner would be more useful? For instance, maybe it would be better to not show all the genes being investigated in the main figure, but select fewer of them which the authors focus on, or move the heatmap into supplementary figure 2 and plot the genes associated with oxidative stress response in db/m vs db/db differently, to make the difference in expression more obvious to the reader.

Answer:

Thank you very much for the professional guidance. We have revised the presentation of the heatmap content according to your request.

Figure 2B Radar diagram showing differentially expressed genes related to calcium signaling pathway, oxidative stress response, and apoptosis.

The axis labels for Figure 2E and F are inconsistent with each other (i.e., on says % of Max, while the other looks at count) even though I can only assume they show the same thing, please make clearer and consistent Y- and X-axis labels.

Answer:

Thank you very much for pointing out this error. We have made the necessary modifications as per your request.

Figure 2E-F Flow cytometry analysis of MMP (E, n = 11 mice per group) and mitochondrial calcium levels (F, n = 11 mice per group) in the hippocampal CA1 neurons of *Bhlhe41^{fl/fl} db/m* or *Bhlhe41^{fl/fl} db/db* mice injected with AAV-CAMKII or AAV-CAMKII-Cre.

While all the bar charts showing the quantification for the FACS plots and the western blots is useful, it makes the main figure unnecessarily busy. The authors could do with moving those in the supplementary figure instead. These changes would make Figure 2 look a lot neater - it has quite a messy look at the moment.

Answer:

We have optimized the overall layout of Figure 2 to make it appear neater. In this revision, due to the streamlining of the heatmap presentation, Figure 2 now has more space available to properly place the bar charts for flow cytometry and Western blotting. To allow readers to directly view both a representative image and statistical bar chart of a result simultaneously, we have temporarily decided not to move the bar charts to the supplementary figures.

Similarly, since the authors keep a consistent colour scheme (black for db/m + aav-CAMKII, etc.) throughout the figure, they could do with removing the text lines associated with the figures, after the first one. Or perhaps find a better way of showing what the colours mean, without having to repeat the same text so many times in the figure; it would help with the readability of the figure.

Answer:

It has been modified in all figures as requested.

3. Figure 3A looks rather crowded, BHLHE41, ARNTL and CLOCK, although highlighted in red, are quite difficult to read. Similarly, Figure 3B is hard to read, the text being very small. Seeing as these two figures highlight a simple STRING/ALGGEN search for interactors of Bhlhe41, rather than results of a proteomics screen through which CLOCK/BMAL have been highlighted as interactors with these tools, I would recommend moving these two figures into supplementary, or removing them entirely, as they do not add anything to the main figure, especially since the authors then go on to validate these interactions in Figure 3C and onwards.

Answer:

It has been moved into the supplementary figure as requested.

I would recommend removing Figure 3L, it does not add anything to the main figure, the authors could stick it in supplementary or remove it entirely and simply mention and reference in the text the online tools they consulted.

Answer:

It has been moved into the supplementary figure as requested.

Overall, as I have commented for the previous figures, Figure 3 has an overall rather disorganised look. For instance, the sizes of various blot panels are different from each other, and the alignment of various components of the figure could be improved, to give it a sharper look overall.

Answer:

It has been modified as requested.

An additional example is the inconsistent way bar chart annotations are localised: In Figure 3D, they are located top right of the bar chart, in 3F they are located middle top, in 3K they are located under the x-axis, while in 3M and 3N they are located to the right or on the top respectively. I would recommend the authors to pick a specific location and stick to it throughout ALL the figures in the paper. Keeping the font size consistent across would also be ideal. All these modifications would make the figure more readable.

Answer:

This problem has been fixed in all figures as requested.

In the first paragraph of the text corresponding to figure 3, the authors show that BMAL1 overexpression upregulates protein/mRNA of BHLHE41, and vice versa upon knockdown. and concluded that 'high glucose upregulated BHLHE41 expression in neurons through BMAL1/ CLOCK heterodimer.' The conclusion in this current form is not entirely supported by the data, as the authors, up until this point have not shown the requirement of BMAL1 interacting with CLOCK to upregulate BHLHE41 expression, they only showed the requirement BMAL1. Please modify the conclusion to reflect this.

Answer:

It has been modified as requested.

Manuscript page 7, lines 20-23: More importantly, knockdown of Bmal1 (Figure 3B and S3D) decreased both of the mRNA and protein levels of Bhlhe41 in high glucose-treated primary hippocampal neurons (Figure S3E and 3B), suggesting that

high glucose upregulated Bhlhe41 expression in hippocampal neurons through Bmal1.

Several paragraphs following this, the authors state that 'high glucose-induced Bhlhe41 upregulation was mediated by Bmal1 O-GlcNAcylation at its S424 site, which enhanced the binding of Clock to Bmal1 and activates Bhlhe41 transcription'. While the first half of the sentence: 'high glucose-induced Bhlhe41 upregulation was mediated by Bmal1 O-GlcNAcylation at its S424 site' is supported by the data, the second part: 'which enhanced the binding of Clock to Bmal1 and activates Bhlhe41 transcription' is not. The authors only show that BMAL1 overexpression/ knockdown impacts BHLHE41 mRNA and protein levels and that mutation of S424A prevented the accumulation of BHLHE41 in response to BMAL1 O-GlcNAcylation. They give no direct evidence to show that the 'binding of CLOCK to BMAL1 activates Bhlhe41 transcription'.

In order to support this kind of statement, the authors need to perform CHIP to show that the whole CLOCK:BMAL1 complex binds the E-boxes in the Bhlhe41 promoter. Similarly, they would need to perform RT-qPCR to show the effect on Bhlhe41 mRNA. Both of these types of experiments should ideally be performed with the WT and S424 BMAL1 mutant the authors described. Only then can the authors state that high glucose-induced Bhlhe41 upregulation was mediated by Bmal1 O-GlcNAcylation at its S424 site, which enhanced the binding of Clock to Bmal1 and activates Bhlhe41 transcription'.

Answer:

This is a very good suggestion. We employed two intervention methods, the S424A mutant and a short peptide (A short amino acid sequence containing the Bmal1 S424

site, which can block the endogenous glycosylation of Bmal1 at the S424 site both in vivo and in vitro), and combined them with three techniques: COIP, CHIP, and qPCR. Through these approaches, we confirmed that the glycosylation modification of the Bmal1 S424 site induced by high glucose promotes its binding to Clock, consequently enhancing its binding in the Bhlhe41 promoter region and ultimately upregulating the expression of Bhlhe41.

Figure 3K Primary hippocampal neurons, which were stably silenced endogenous Bmal1 and expressed WT Bmal1 or S424A Bmal1, were treated with normal glucose (NG, 5.5 mmol/L D-glucose) or high glucose (HG, 25 mmol/L D-glucose). Cell lysates were immunoprecipitated with Bmal1 antibodies and western blotted with the indicated antibodies.

Figure S3K mRNA levels of Bhlhe41 in primary hippocampal neurons with endogenous Bmal1 stably silenced and expressing either wild-type (WT) Bmal1 or S424A Bmal1, treated with normal glucose (NG, 5.5 mmol/L D-glucose) or high

glucose (HG, 25 mmol/L D-glucose) (n = 3 replicates).

Figure 3L Binding of Bmal1 to the Bhlhe41 promoter in primary hippocampal neurons with endogenous Bmal1 stably silenced and expressing either wild-type (WT) Bmal1 or S424A Bmal1, treated with normal glucose (NG, 5.5 mmol/L D-glucose) or high glucose (HG, 25 mmol/L D-glucose). IgG was used as negative CHIP control (n = 3 replicates).

Figure 6C-D Hippocampal CA1 neurons were isolated from db/m, db/db (C), Ctr or STZ mice (D) treated with vehicle, S424A-pe (5 mg/kg) or S424-pe (5 mg/kg), and cell lysates were immunoprecipitated with Bmal1 antibodies and western blotted with the indicated antibodies (n = 6 mice per group).

Figure 6E-F Immunoblots and quantification analysis of Bhlhe41, Dnajb4 and NCLX expression levels in the hippocampal CA1 neurons of db/m, db/db (E), Ctr or STZ mice (F) treated with vehicle, S424A-pe (5 mg/kg) or S424-pe (5 mg/kg) (n = 6 mice per group).

Figure S6A-B

(A): Primary hippocampal neurons were treated with normal glucose (NG, 5.5 mmol/L D-glucose) or high glucose (HG, 25 mmol/L D-glucose) in the presence of vehicle, S424A-pe (20 μ M) or S424-pe (20 μ M), and cell lysates were immunoprecipitated with Bmal1 antibodies and western blotted with the indicated antibodies (n = 3 replicates).

(B): Immunoblots and quantification analysis of Bhlhe41, Dnajb4 and NCLX expression levels in primary hippocampal neurons treated with normal glucose (NG, 5.5 mmol/L D-glucose) or high glucose (HG, 25 mmol/L D-glucose) in the presence of vehicle, S424A-pe (20 μ M) or S424-pe (20 μ M) (n = 3 replicates).

Manuscript page 9, lines 9-14: “Consistently, S424A mutant significantly reversed high glucose-induced Bmal1 O-GlcNAcylation in mouse primary hippocampal neurons with endogenous Bmal1 knockdown, accompanied by a decrease in binding with Clock and a downregulation of the protein and mRNA levels of Bhlhe41 (Figure 3K and S3J-K). Additionally, Chromatin immunoprecipitation analysis showed enhanced binding of Bmal1 to the Bhlhe41 promoter under high glucose conditions and S424A mutant treatment weakened this binding (Figure 3L).”

Manuscript page 12, lines 24-26—Manuscript page 13, lines 1-3: “Thus, based on the S424-pe peptide, we generated a scrambled peptide denoted as S424A-pe, in which the S424 amino acid was replaced with an A residue. As shown in Figure 6C and D, S424-pe, but not S424A-pe, significantly reversed high glucose-induced Bmal1 O-GlcNAcylation in hippocampal CA1 neurons isolated from db/db (Figure 6C) and STZ-induced diabetic mice (Figure 6D), accompanied by a decrease in binding of Clock to Bmal1 and a downregulation of Bhlhe41 and Dnajb4 expression (Figure 6E-F).”

Manuscript page 13, lines 6-7: “Similar results were also obtained in mouse primary hippocampal neurons treated with high glucose (Figure S6A-E).”

4. Could the authors please clarify whether the neurons they looked at were all hippocampal neurons or hippocampal CA1 neurons specifically, which the investigation originally started with. The same point applies for Figure 3, where the authors mention using 'primary hippocampal neurons'.

Answer:

Thank you very much for pointing out the error in our description. In the in vivo studies on diabetic mice, we exclusively examined neurons from the CA1 region of

the hippocampus. However, in the in vitro cell studies, we used primary hippocampal neurons extracted from the hippocampus of fetal mice. Due to the small size of the fetal mouse hippocampus, these extracted primary hippocampal neurons do not exclusively originate from the CA1 region. We have provided further clarification regarding the extraction site of primary hippocampal neurons in the Methods section.

Manuscript page 19, lines 10-11: “The extraction site of primary hippocampal neurons is not limited to the CA1 region of the fetal mouse hippocampus.”

The blots showing that overexpression of Bhlhe41 did not upregulated USP2 and FOSB protein levels are not shown, as the authors claim in the text. Show them please.

Answer:

It has been modified as requested.

Figure 4F Immunoblots analysis of protein levels of Bhlhe41, Dnajb4, Usp2 and Fosb (n = 3 blots in total) in primary hippocampal neurons transfected with control plasmid or a plasmid overexpressing Bhlhe41 for 48h.

5. Figure 5C is not mentioned in the text.

Answer:

It has been modified in the text.

Manuscript page 11, lines 13-15: We next injected AAV9-CamkII-NCLX or an AAV9-CamkII control bilaterally into hippocampal CA1 region of db/db and STZ-induced diabetic mice to specifically restore NCLX expression in hippocampal CA1 neurons in vivo (Figure 5A-B, S5D-E).

Collectively, these findings suggested that NCLX downregulation induced mitochondrial calcium overload and oxidative damage in hippocampal CA1 neurons, thereby leading to neuronal apoptosis and cognitive dysfunction in diabetes.' It would be more appropriate for the authors to state the effects of NCLX overexpression, as this is what they tested in section of the paper, not the NCLX downregulation.

Answer:

It has been modified as requested.

Manuscript page 11, lines 19-21: “Collectively, these findings suggested that NCLX overexpression significantly reversed high glucose-induced mitochondrial calcium overload and oxidative damage in hippocampal CA1 neurons, thereby leading to an improvement in cognitive function in diabetes.”

Figure 5L is too small to interpret, please make it bigger. A size/scale marker for these images would also be appropriate.

Answer:

It has been modified as requested.

Figure 5K MMP measurement using JC-1 dye in primary hippocampal neurons transfected with Dnajb4 overexpression plasmid or control plasmid (n = 3 replicates). Scale bar: 200 μ m.

In figure 5O, how do the authors rationalise the similarity in protein levels between NCLX levels under control and MG-132 levels? One would perhaps a more pronounced stabilisation of NCLX upon proteasomal inhibition. Additionally, how do the authors rationalise the apparently incomplete NCLX protein level rescue upon treatment with MG-132 in the DNAJB overexpression background?

Answer:

This is a very good suggestion, we have extensively discussed the possible reasons for this phenomenon in the discussion section.

Manuscript page 17, lines 15-25: “Finally, after intervention with MG132 in the control group, there was no significant further increase in the level of NCLX, which may be related to the compensatory mechanism of cellular calcium homeostasis (Rougier et al, 2013). When MG132 is used to inhibit the degradation of NCLX under physiological conditions, neurons may compensatorily reduce the generation of NCLX in order to avoid excessive expression of NCLX leading to increased mitochondrial calcium efflux, thereby maintaining the calcium homeostasis in the mitochondria. In the Dnajb4 overexpression group, intervention with MG132 did not

completely restore NCLX expression levels. This may be related to the multifunctionality of Dnajb4. We speculate that while Dnajb4 mediates NCLX ubiquitination and degradation, it may also inhibit its generation. Additionally, differences in the duration and concentration of MG132 intervention may also lead to the above situation. The above speculations require further experimental verification in future studies.”

6. Minor concerns that should be addressed:

Figure 1:

- Figure S1C is missing the label on the y-axis.

Answer:

Thank you for your kind reminder. Since Figure S1C does not effectively display the research results, we have used another method for presentation (Figure 1A).

- Figure 1F is missing the label on the y-axis.

Answer:

It has been modified as requested (Figure 1A).

- Please briefly explain what the drug streptozotocin is meant to do in the experimental set up used, to help people who are not familiar with it.

Answer:

It has been modified as requested.

Manuscript page 6, lines 8-9: “STZ has been shown to possess diabetogenic properties leading to the destruction of islet β -cell (Lenzen, 2008).”

Figure 2:

- Please make the x-axis label of Figure 2C easier to understand. 'mitosox' doesn't mean anything to me. 'mitochondrial ROS levels' or similar would be more appropriate.

Answer:

It has been modified in all figures as requested.

- It would be useful to mention in the text how Bcl-2, Bax and cleaved Caspase-3 relate to apoptosis, i.e., mention that high Caspase-3 and Bax are associated with increased apoptosis, etc.

Answer:

It has been modified as requested.

Manuscript page 6, lines 24-25: “As shown in Figure 2C-E and S2A-S2C, hippocampal CA1 neurons isolated from db/db and STZ-induced diabetic mice exhibited higher apoptosis (Figure 2D and S2B) (as evidenced by decreased expression of Bcl2 and increased expression of Bax and caspase3)”

Figure 4:

- I would recommend performing some GO analysis on the enriched genes, if possible.

Answer:

It has been modified as requested.

Figure 4D GO analysis of DEGs modulated by Bhlhe41. MF, molecular function; CC, cellular component; BP, biological process.

Figure 7:

- Figure The authors might wish to consider highlighting S424 as the major site of O-GlcNAcylation, rather than UDP-GlcNAc dotted on BMAL1 randomly.

Answer:

Based on your last suggestion, we have removed all relevant research content on GLP-1 receptor agonists from the article (including Figure 7), and instead opted to use a short peptide to directly intervene in Bmal1 glycosylation levels to validate the direct causal relationship between high glucose-induced Bmal1 glycosylation modification and downstream mitochondrial calcium overload and oxidative stress

damage.

7. Additional non-essential suggestions for improving the study:

- BMAL1 has also been shown to be a translation factor, such that the observed increase in BHLHE41 protein and mRNA levels could in fact be a consequence of altered translation mediated by BMAL1 (not just transcription), i.e., by enhancing translation and thereby stabilising Bhlhe41 mRNA under high glucose. This would be a highly interesting avenue for the authors to explore.

Answer:

This is a very good suggestion, which has provided us with a direction for further in-depth research. Due to time constraints, we are currently unable to complete the experimental content in this aspect in this study. However, we have included this suggestion in the discussion section.

Manuscript page 17, lines 11-15: “Third, Bmal1 has also been shown to be a translation factor (Lipton et al, 2015), such that the observed increase in Bhlhe41 protein and mRNA levels could also be a consequence of altered translation mediated by Bmal1 (not just transcription), i.e., by enhancing translation and thereby stabilizing Bhlhe41 mRNA under high glucose. This, however, remains purely speculative and would be an interesting avenue for future research.”

- Seeing as the authors focus so much on the interaction between CLOCK and

BMAL1, which are core circadian clock components, have the authors considered assessing for example testing whether oscillations in mitochondrial calcium accumulation and oxidative damage in hippocampal neurons (if any are present) are affected by high glucose/ O-GlcNAcylation of BMAL?

Answer:

This is a great suggestion, actually, prior to first submission, we were aware of this research limitation and had already begun addressing the issue. Given that conventional methods for mutating the Bmal1 S424 site in hippocampal neurons of diabetic mice would be time-consuming and labor-intensive, we intervened hippocampal neurons under high glucose conditions both in vivo and in vitro using artificially synthesized short peptides containing the Bmal1 S424 site and short peptides containing the mutated Bmal1 S424 site. These short peptides were fused with a cell-penetrating peptide, rendering them capable of permeating cell membranes and crossing the blood-brain barrier (Qin, Luye et al. *iScience*, PMID: 31247448). We then assessed downstream effects on the Bhlhe41/Dnajb4/NCLX signaling pathway and related outcome measures, providing robust evidence for our scientific hypothesis that oscillations in mitochondrial calcium accumulation and oxidative damage in hippocampal neurons are affected by O-GlcNAcylation of Bmal1. This approach has since been adopted by other researchers for similar studies (Chen, Yuping et al. *Cell*, PMID: 38128537; Zhu, Qiang et al. *Proc Natl Acad Sci U S A*, PMID: 36943877), yielding promising experimental results. By employing this method, we not only addressed this important issue but also significantly shortened our revision cycle.

In order to better highlight the focus of our research and the causal relationships between mechanistic pathways, this revision has removed all content related to the treatment of diabetes-associated cognitive dysfunction with GLP-1 receptor agonists.

Instead, it has been entirely replaced by the study of the effects and molecular mechanisms of short peptide treatment on mitochondrial calcium accumulation and oxidative damage in hippocampal neurons under high glucose conditions.

Figure 6. Short Peptide S424-pe Protects Against High Glucose-Induced Neuronal Apoptosis and Cognitive Dysfunction by Regulating Bhlhe41/Dnajb4/NCLX Signaling

(A): Schematic illustration of designed peptides. Blue: cell-penetrating peptide (CPP). Red: site for Bmal1 O-GlcNAcylation.

(B): Primary hippocampal neurons were treated with five short peptides (20 μ M) in the presence of high glucose (HG, 25 mmol/L D-glucose), and Bmal1 O-GlcNAcylation was determined by immunoprecipitation and western blot using the indicated antibodies. Three independent experiments were performed.

(C-D): Hippocampal CA1 neurons were isolated from db/m, db/db (C), Ctr or STZ mice (D) treated with vehicle, S424A-pe (5 mg/kg) or S424-pe (5 mg/kg), and cell lysates were immunoprecipitated with Bmal1 antibodies and western blotted with the indicated antibodies (n = 6 mice per group).

(E-H): Immunoblots and quantification analysis of Bhlhe41, Dnajb4, NCLX (E-F), Bax, Bcl-2 and Cleaved-caspase3 (G-H) expression levels in the hippocampal CA1 neurons of db/m, db/db, Ctr or STZ mice treated with vehicle, S424A-pe (5 mg/kg) or S424-pe (5 mg/kg) (n = 6 mice per group).

(I-L): Flow cytometry analysis of mitochondrial calcium levels (I and K) and mtROS levels (J and L) in the hippocampal CA1 neurons isolated from different groups of mice (n = 6 mice per group).

Data are means \pm SEM. *P < 0.05, **P < 0.01, ***P < 0.001. Two-way ANOVA followed by Tukey's test (C-L).

Figure 7. Short Peptide S424-pe Protects Against High Glucose-Induced Neuronal Apoptosis and Cognitive Dysfunction by Regulating Bhlhe41/Dnajb4/NCLX Signaling

(A-D): Representative track images and escape latency to the platform during the

training trials, target entries and time spent in target quadrant in the probe trial of Morris water maze (A-B), and active avoidance performance (C-D) of db/m, db/db, Ctr, STZ mice treated with vehicle, S424A-pe (5 mg/kg) or S424-pe (5 mg/kg) (n = 6 mice per group).

Data are means \pm SEM. *P < 0.05, **P < 0.01, ***P < 0.001. Two-way ANOVA followed by Tukey's test (A-D).

Figure S6. Short Peptide S424-pe Protects Against High Glucose-Induced Neuronal Apoptosis and Cognitive Dysfunction by Regulating Bhlhe41/Dnajb4/NCLX Signaling

(A): Primary hippocampal neurons were treated with normal glucose (NG, 5.5 mmol/L D-glucose) or high glucose (HG, 25 mmol/L D-glucose) in the presence of

vehicle, S424A-pe (20 μ M) or S424-pe (20 μ M), and cell lysates were immunoprecipitated with Bmal1 antibodies and western blotted with the indicated antibodies (n = 3 replicates).

(B): Immunoblots and quantification analysis of Bhlhe41, Dnajb4 and NCLX expression levels in primary hippocampal neurons treated with normal glucose (NG, 5.5 mmol/L D-glucose) or high glucose (HG, 25 mmol/L D-glucose) in the presence of vehicle, S424A-pe (20 μ M) or S424-pe (20 μ M) (n = 3 replicates).

(C-E): Flow cytometry analysis of mitochondrial calcium levels (C), mtROS levels (D) and apoptosis levels (E) in primary hippocampal neurons with different treatments as indicated (n = 3 replicates).

Data are means \pm SEM. *P < 0.05, **P < 0.01, ***P < 0.001. Two-way ANOVA followed by Tukey's test (A-E).

Manuscript page 12, lines 17-26—Manuscript page 13, lines 1-10: “To provide direct evidence that O-GlcNAcylation of Bmal1 at S424 impairs cognitive function through the Bhlhe41/Dnajb4/NCLX signaling pathway, we developed a short peptide designed to specifically target the O-GlcNAcylation of Bmal1 at S424. After analyzing the sequence surrounding the Bmal1 S424 site, we synthesized five short peptides fused with a cell-penetrating peptide, rendering them capable of permeating cell membranes and crossing the blood-brain barrier (Qin et al, 2019). This approach has emerged as one of the most popular and effective methods for intracellular delivery of biomolecules. (Koren et al, 2011; Li et al, 2021) (Figure 6A). Among these peptides, S424-peptide-2# (referred to as S424-pe) exhibited the strongest inhibitory effect on Bmal1 O-GlcNAcylation. (Figure 6B). Thus, based on the S424-pe peptide, we generated a scrambled peptide denoted as S424A-pe, in which the S424 amino acid was replaced with an A residue. As shown in Figure 6C and D, S424-pe, but not S424A-pe, significantly reversed high glucose-induced Bmal1

O-GlcNAcylation in hippocampal CA1 neurons isolated from db/db (Figure 6C) and STZ-induced diabetic mice (Figure 6D), accompanied by a decrease in binding of Clock to Bmal1 and a downregulation of Bhlhe41 and Dnajb4 expression (Figure 6E-F). Furthermore, S424-pe significantly upregulated NCLX expression (Figure 6E-F) and reduced mitochondrial calcium concentrations (Figure 6I and K) and oxidative damage (Figure 6J and L), followed by a decrease in neuronal apoptosis (Figure 6G-H) and an improvement in cognitive function (Figure 7A-D). Similar results were also obtained in mouse primary hippocampal neurons treated with high glucose (Figure S6A-E). Collectively, our results demonstrated that targeting Bmal1 S424 O-GlcNAcylation with S424-pe may alleviate high glucose-induced mitochondrial calcium overload and oxidative damage in hippocampal CA1 neurons and thereby improve cognitive function in diabetes.”

Manuscript page 16, lines 8-23: “Small-molecule inhibitors targeting OGT have been created to explore the pathogenic role of O-GlcNAcylation modifications in a variety of diseases (Rohlf et al, 1999; Sharma et al, 2019). However, as OGT catalyzes O-GlcNAcylation on a wide range of protein substrates and plays a crucial role in maintaining normal physiological functions, the inhibition of OGT is likely to result in adverse side effects both in vivo and in vitro (Zhu et al, 2023). To overcome this limitation, we designed an alternative strategy that employs a short peptide containing Bmal1 O-GlcNAcylation site to competitively inhibit the O-GlcNAcylation of Bmal1 S424 with minimal interference with the total cellular O-GlcNAcylation level. Such treatment led to a decrease in binding of Clock to Bmal1 and a downregulation of Bhlhe41 and Dnajb4 expression, resulting in NCLX upregulation and an improvement in mitochondrial calcium overload, hippocampal neuron dysfunction, and cognitive impairment in diabetes. Our data suggest that

oscillations in mitochondrial calcium accumulation and oxidative damage in hippocampal neurons are indeed affected by O-GlcNAcylation of Bmal1, in addition, it provides solid evidence that Bmal1 O-GlcNAcylation directly impairs cognitive function in diabetes. Short peptide has been used in a recent study to elucidate the role of O-GlcNAcylation modification in tumor immune evasion (Zhu et al., 2023). Our study further confirmed that this strategy targeting site-specific O-GlcNAcylation on Bmal1 in hippocampal neurons may serve as a novel therapeutic approach for the treatment of cognitive impairment in diabetes.”

Manuscript page 27, lines 4-9: “The peptides utilized in this study were synthesized by Guoping Pharmaceutic Inc. Synthetic peptides underwent purification via high-pressure liquid chromatography to achieve a purity exceeding 98%, suitable for both in vitro and in vivo applications. Peptides were dissolved in phosphate-buffered saline (PBS) to create a 20 mM stock solution for in vitro experiments. For in vivo use, S424-pe and S424A-pe were dissolved in PBS and stored on ice prior to injection. Prior to injection, the solution was allowed to reach room temperature.”

Dear Prof. Zheng,

Congratulations on a great revision! Overall, the referees have been positive. However, the referees have raised a few concerns that we ask you to non-experimentally address in a revised version. When you submit your revision, please also take care of the following editorial items and add this also to your point-by-point response:

1. Please provide us with your institutional email address.
2. Please reduce the number of keywords to five.
3. Please remove the author contribution section from the main manuscript.
4. Please update the callouts of the Appendix Figures. They should be "Appendix Figure S1" instead of Figure S1.
5. Please provide an author checklist, which you can find online.
6. Please provide the appendix file in PDF and update the nomenclature to "Appendix Figure S1" etc, and add a title page with table of contents and page numbers.
7. We require the publication of source data, particularly for electrophoretic gels and blots and graphs, with the aim of making primary data more accessible and transparent to the reader. It would be great if you could provide me with a PDF file per figure that contains the original, uncropped and unprocessed scans of all or key gels used in the figure or for graphs, an Excel spreadsheet with the original data used to generate the graphs. The PDF files should be labeled with the appropriate figure/panel number, and should have molecular weight marker; further annotation could be useful but is not essential. The PDF files will be published online with the article as supplementary "Source Data" files.
8. Please provide a summary statement with bullet point describing the main findings of your manuscript for use on our website, please see our website for examples.
9. Please update the order of sections to the following: Title page - Abstract & Keywords - Introduction - Results - Discussion - Methods - Data Availability - Acknowledgments - Disclosure Statement & Competing Interests - References - Figure Legends - (Tables with legends) - Expanded View Figure Legends.
10. Please note that the exact p values are not provided in the legends of figures 1b-f; 2c-f; 3a-b, d, f-l; 4e-k; 5a-n; 6c-l; 7a-d, supplementary figures 1g, m-o; 2a-d; 3c-h, j, k-n; 4b-d; 5a-i, k; 6a-e.
11. Please indicate the statistical test used for data analysis in the legends of figures 2a; 4a, d, supplementary figure 4a.
12. Please note that in figures 2c-f; 4e-k, supplementary figures 2a-d; 3c-h, j, k-n; there is a mismatch between the annotated p values in the figure legend and the annotated p values in the figure file that should be corrected.
13. Please note that the box plots need to be defined in terms of minima, maxima, centre, bounds of box and whiskers, and percentile in the legends of figure 1a, supplementary figure 4a.
14. Please note that information related to n is missing in the legends of figures 1a; 3k, supplementary figure 4a.

Thank you for the opportunity to consider your work for publication. I look forward to your revision.

Kind regards,
Kelly

Kelly M Anderson, PhD
Editor, The EMBO Journal
k.anderson@embjournal.org

Use the link below to submit your revision:

Referee #1:

The authors address most of the initially reported concerns. A few things (one major) persist that should be fixed:

1. The last sentence of the abstract (and related Discussion) should be removed - this was not demonstrated sufficiently (Per1-3, Cry1/2 at a single timepoint is not nearly enough to justify this conclusion) - fully day timecourse sampling in the related manipulations would be required to support that conclusion. However, just deleting that conclusions about 'circadian disruption' is all that is required in my view.
2. Fig 6 and 7 have the same title, this is incorrect.
3. Since these factors are rhythmic in most tissues, at a minimum the authors need to report for each animal experiment, what time animals were selected. Any groups where there's a 6 or more hour difference between when control and experimental groups were collected must be revalidated, or data needs to be added to indicate that these events in the hippocampus are NOT rhythmic, meaning that differences in time-of-day of collection are unlikely to contribute. This is still a major weakness of this paper that was not suitably addressed.
4. It is a striking paradox that glucose/db increase both Bhlhe41 binding to promoters of genes that are also apparently upregulated... Bhlhe41 is known to be a repressor... ? The authors add a complex possible explanation, but it is pretty 'hand wavy'. Instead, I would suggest exploring expression on genes where CLOCK/BMAL1 are the primary drivers of expression - Pers/Crys are influenced by a number of factors in addition to CLOCK/BMAL.

Referee #2:

In the revised version of the manuscript, Hui and colleagues added novel experimental data and comments that successfully addressed my previous comments.
In particular, the molecular role of the circadian clock machinery in their model of glucose-mediated cognitive function has been now evaluated and discussed considering the recent literature.
I believe that the present version of the manuscript is suitable for publication in EMBOJ.

Referee #3:

Congratulations to the authors for the extra effort put into improving the manuscript, it clearly shows.
In my opinion, based on feedback from the reviewers in the first round, the authors managed to restructure significant portions of the manuscript making it more easy to follow and with a better logical flow.

The authors have also made a great addition to the study by using a short-peptide strategy designed to specifically target O-glcNAcylation of BMAL1. This is a strong point of the study.

This time around, the vast majority of conclusions of the paper are justified based on the presented data.

Major comments:

- The authors addressed the vast majority of my comments. However, I continue to have a big issue with the presentation of the data.

Despite listening to my advice of moving certain bits into supplementary figures to free up the space in the main figure, the majority of the main figures remain, with the exception of Figures 1 and 2, very cluttered. While as a reviewer, I have the patience to read through everything, your general reader will not be so patient. I strongly urge the authors to reconsider the general organisation of their figures in order to more effectively convey their message to the reader. Less (fewer, but more relevant figures) is more in this case.

- The short BMAL1 peptide strategy is really great. However, I cannot find details of how these are constructed, in terms of sequence in the methods section. Yes, you reference Qin et al 2019, but I would prefer to see the full details, the full sequences of all the S424-based peptides used in your study, in this manuscript. By this I mean not just the BMAL1 sequences that are shown in the figure, but the whole constructs containing the cell-penetrating peptide, the nucleotide and amino acid sequences used.

Additionally, I can see no validation of the specificity of these short peptides. Indeed the blots show that these short peptides are

capable of reducing O-glcNAcylation of BMAL1 at S424, but how does one know for sure that binding to other off-targets is not responsible for the observed effects. The authors could approach this skepticism by performing IP-MS on their top peptide to interrogate its interacting partners.

Minor comments:

- Figure 1: I still think the figure can be simplified more. I am particularly looking at E and F. To someone who is not familiar with these assays, the traces of how the mice move in those circles as well as the 'nr of target entries' and 'time spent in the quadrant' is largely irrelevant. What I want to see is a simple graph telling me whether cognition is improved or not. As such, I would just keep the graphs about escape latency and lengthen the y axes, so the separation between the traces is more apparent, and move the rest in supplementary.

- Figure 3H: move into supplementary, hardly any need to show these HEK cell results in the main figure when you already show it in neurons in 3I/J.

- 'Furthermore, this treatment significantly reduced mitochondrial calcium concentrations (Figure S3L) and oxidative damage (Figure S3M), followed by a decrease in neuronal apoptosis (Figure S3N).' I would much rather think you should show this in the main figure, as this is a tangible functional output that is affected by the S424A mutation.

We sincerely thank the editor and reviewers for their thoughtful and encouraging comments on our manuscript and greatly appreciate their time and effort in reviewing our work. The manuscript has been revised in response to the editor's and reviewers' comments, with all textual changes highlighted in yellow. Below is a detailed, point-by-point response to each comment.

Editorial comments

1. Please provide us with your institutional email address.

Response: Thanks for pointing this out. We have provided the institutional email address in the revised manuscript.

2. Please reduce the number of keywords to five.

Response: We have reduced the number of keywords to five in the revised manuscript.

Manuscript page 2, lines 15-17:

Keywords

Diabetes; Cognitive impairment; O-GlcNAcylation; Bmal1-Clock complex;
Mitochondria calcium overload

3. Please remove the author contribution section from the main manuscript.

Response: We have removed the Author Contribution section.

4. Please update the callouts of the Appendix Figures. They should be "Appendix Figure S1" instead of Figure S1.

Response: We have updated the callouts of the Appendix Figures.

5. Please provide an author checklist, which you can find online.

Response: We have filled out and submitted the author checklist.

6. Please provide the appendix file in PDF and update the nomenclature to "Appendix Figure S1" etc, and add a title page with table of contents and page numbers.

Response: We have compiled the appendix file into a PDF document with reference to the latest article in *The EMBO Journal*, updated the nomenclature and added a title page with table of contents and page numbers.

7. We require the publication of source data, particularly for electrophoretic gels and blots and graphs, with the aim of making primary data more accessible and transparent to the reader. It would be great if you could provide me with a PDF file per figure that contains the original, uncropped and unprocessed scans of all or key gels used in the figure or for graphs, an Excel spreadsheet with the original data used to generate the graphs. The PDF files should be labeled with the appropriate figure/panel number, and should have molecular weight marker; further annotation could be useful but is not essential. The PDF files will be published online with the article as supplementary "Source Data" files.

Response: We have provided all the source data in the main figures with the kind help of the SourceData Science Coordinator.

8. Please provide a summary statement with bullet point describing the main findings of your manuscript for use on our website, please see our website for examples.

Response: We have provided a synopsis of the manuscript by referencing the latest articles on *The EMBO Journal* website, to help readers quickly grasp the bullet points of our research.

9. Please update the order of sections to the following: Title page Abstract & Keywords - Introduction - Results - Discussion - Methods - Data Availability - Acknowledgments - Disclosure Statement & Competing Interests - References - Figure Legends - (Tables with legends) Expanded View Figure Legends.

Response: We have updated the order of sections in the manuscript as requested.

10. Please note that the exact p values are not provided in the legends of figures 1b-f; 2c-f; 3a-b, d, f-l; 4e-k; 5a-n; 6c-l; 7a-d, supplementary figures 1g, m-o; 2a-d; 3c-h, j, k-n; 4b-d; 5a-i, k; 6a-e.

Response: Thanks. We have provided exact p values for the above images in the revised manuscript.

11. Please indicate the statistical test used for data analysis in the legends of figures 2a; 4a, d, supplementary figure 4a.

Response: Thanks for pointing this out. We have indicated the statistical test used for data analysis in the legends of the above figures (Manuscript page 38, line 21; page 41, lines 25-26; Supplementary Materials page 10, line 17).

12. Please note that in figures 2c-f; 4e-k, supplementary figures 2a-d; 3c-h, j, k-n; there is a mismatch between the annotated p values in the figure legend and the annotated p values in the figure file that should be corrected.

Response: We have corrected the figure legends to match the p values annotated in the figure file.

13. Please note that the box plots need to be defined in terms of minima, maxima, centre, bounds of box and whiskers, and percentile in the legends of figure 1a, supplementary figure 4a.

Response: We have defined the box plots (Figure 1A and Appendix Figure 4A) as requested.

14. Please note that information related to n is missing in the legends of figures 1a; 3k, supplementary figure 4a.

Response: We greatly appreciate your patience in handling our manuscript. We have added information related to n in the legends of Figures 1A, 3K (now 3G), and Supplementary Figure 4A (Manuscript page 37, line 8; page 39, line 30; Supplementary Materials page 9, line 7).

Referee #1

The authors address most of the initially reported concerns. A few things (one major) persist that should be fixed:

Response: We are very grateful for the constructive feedback provided by the reviewer, which has greatly contributed to the improvement of our manuscript.

1. The last sentence of the abstract (and related Discussion) should be removed - this was not demonstrated sufficiently (Per1-3, Cry1/2 at a single timepoint is not nearly enough to justify this conclusion) - fully day timecourse sampling in the related manipulations would be required to support that conclusion. However, just deleting that conclusions about 'circadian disruption' is all that is required in my view.

Response: Thank you very much for raising this important issue. We have removed the last sentence of the abstract and related discussion.

2. Fig 6 and 7 have the same title, this is incorrect.

Response: Thanks for pointing this out. We have modified the titles of Figures 6 and 7.
Manuscript page 44, lines 1-2: Figure 6. Short Peptide S424-pe Protects Against High Glucose-Induced Neuronal Apoptosis by Regulating Bhlhe41/Dnajb4/NCLX Signaling
Manuscript page 47, lines 1-2: Figure 7. Short Peptide S424-pe Protects Against High Glucose-Induced Cognitive Dysfunction by Regulating Bhlhe41/Dnajb4/NCLX Signaling

3. Since these factors are rhythmic in most tissues, at a minimum the authors need to report for each animal experiment, what time animals were selected. Any groups where there's a 6 or more hour difference between when control and experimental groups were collected must be revalidated, or data needs to be added to indicate that these events in the hippocampus are NOT rhythmic, meaning that differences in time-of-day of collection are unlikely to contribute. This is still a major weakness of this paper that was not suitably addressed.

Response: This is an excellent suggestion. To avoid the impact of sample collection time differences on gene expression, we have standardized the sampling time to between 8:00 AM and 9:30 AM, and redone all experiments that did not meet this timeframe in the revised manuscript.

Manuscript page 18, lines 22-23:

The sample collection time is between 8:00 AM and 9:30 AM.

4. It is a striking paradox that glucose/db increase both Bhlhe41 binding to promoters of genes that are also apparently upregulated... Bhlhe41 is known to be a repressor... ? The authors add a complex possible explanation, but it is pretty 'hand wavy'. Instead, i would suggest exploring expression on genes where CLOCK/BMAL1 are the primary drivers of expression - Pers/Crys are influenced by a number of factors in addition to CLOCK/BMAL.

Response: Thank you for your insightful comments. We apologize for the previous "hand-wavy" explanation. Our discussion of these points was initially motivated by Reviewer 2's request. In the revised manuscript, we have removed unnecessary explanations while addressing Reviewer 2's comments (Manuscript page 15, lines 5-14).

Referee #2

In the revised version of the manuscript, Hui and colleagues added novel experimental data and comments that successfully addressed my previous comments. In particular, the molecular role of the circadian clock machinery in their model of glucose-mediated cognitive function has been now evaluated and discussed considering the recent literature. I believe that the present version of the manuscript is suitable for publication in EMBOJ.

Response: We are very grateful for the comments from Reviewer 2. The comments from Reviewer 2 have strengthened the foundation of our research conclusions. We deeply appreciate the valuable time and insightful comments provided by the knowledgeable reviewer.

Referee #3

Congratulations to the authors for the extra effort put into improving the manuscript, it clearly shows.

In my opinion, based on feedback from the reviewers in the first round, the authors managed to restructure significant portions of the manuscript making it more easy to follow and with a better logical flow.

The authors have also made a great addition to the study by using a short-peptide strategy designed to specifically target O-glcNAcylation of BMAL1. This is a strong point of the study.

This time around, the vast majority of conclusions of the paper are justified based on the presented data.

Response: We sincerely appreciate the valuable time and unique insights provided by the reviewer. The comments from Reviewer 3 have significantly improved the logical coherence of the research.

Major comments:

1. The authors addressed the vast majority of my comments. However, I continue to have a big issue with the presentation of the data. Despite listening to my advice of moving certain bits into supplementary figures to free up the space in the main figure, the majority of the main figures remain, with the exception of Figures 1 and 2, very cluttered. While as a reviewer, I have the patience to read through everything, your general reader will not be so patient. I strongly urge the authors to reconsider the general organisation of their figures in order to more effectively convey their message to the reader. Less (fewer, but more relevant figures) is more in this case.

Response: We sincerely appreciate your thoughtful suggestions. In the revised manuscript, we have moved more results to supplementary figures based on your feedback and optimized the layout of main figures to make them clearer and more

concise. If further modifications are needed in our main and supplementary figures, we would be willing to address them in subsequent revisions.

2.1 The short BMAL1 peptide strategy is really great. However, I cannot find details of how these are constructed, in terms of sequence in the methods section. Yes, you reference Qin et al 2019, but I would prefer to see the full details, the full sequences of all the S424-based peptides used in your study, in this manuscript. By this I mean not just the BMAL1 sequences that are show in the figure, but the whole constructs containing the cell-penetrating peptide, the nucleotide and aminoacid sequences used.

Response: We deeply appreciate the reviewer's attention to these important details. In the revised manuscript, we have included more methodological details and the full amino acid sequences of S424-pe-1# to 5# and S424A-pe, which contain the cell-penetrating peptide, in the Methods-Peptide synthesis section.

Manuscript page 27, lines 12-26 - Manuscript page 28, lines 1-4:

The peptides utilized in this study were synthesized through Fmoc solid phase synthesis on Rink amide resin. Initially, the resin was swelled and deprotected with dichloromethane (DCM). The condensation reaction was carried out by using Fmoc-AA (amino acid)-OH, hydroxybenzotriazole (HOBT) dissolved in dimethylformamide (DMF), followed by diisopropylcarbodiimide (DIC), and added to the drained resin for 1 hour with nitrogen bubble reaction. To remove the Fmoc protecting groups, the resin was washed four times with DMF. After acetic anhydride capping, a cutting solution consisting of trifluoroacetic acid (TFA), H₂O, 1,2-ethanedithiol (EDT), and triisopropylsilane (TIS) in a volume ratio of 95:1:2:2 was added to the resin for 2 hours to cut the peptide. Synthetic peptides underwent purification via high-pressure liquid chromatography to achieve a purity exceeding 98%, suitable for both in vitro and in vivo applications. Peptides were dissolved in phosphate-buffered saline (PBS) to create a 20 mM stock solution for in vitro experiments. For in vivo use, S424-pe and S424A-pe were dissolved in PBS and stored on ice prior to injection. Prior to injection, the solution was allowed to reach room temperature. Synthetic peptides sequences are as follows:

S424-pe-1#: YGRKKRRQRRRITLRSRWFSFMNP

S424-pe-2#: YGRKKRRQRRRDGSFITLRSRWFS

S424-pe-3#: YGRKKRRQRRRSFITLRSRWFSFM

S424-pe-4#: YGRKKRRQRRRLRSRWFSFMNPWT

S424-pe-5#: YGRKKRRQRRRIKDGSFITLRSRW

S424A-pe: YGRKKRRQRRRDGSFITLRARWFS

The diagram of the peptide synthesis steps is shown below:

1) First, we choose resin as the "foundation" for our peptide synthesis, acting as the solid-phase support. The resin is equipped with "hooks" that can securely hold the first amino acid in place. This allows us to gradually add other amino acids step by step, ultimately constructing the entire peptide chain.

2) Next, we wash off the "protective cover" on the resin's "hooks," allowing them to do their job and securely anchor the first amino acid.

3) We then introduce the first amino acid into the peptide, and the "hooks" on the resin firmly secure it in place. The amino group of the amino acid is covered with an Fmoc group, acting as a "protective cover." This Fmoc group prevents any unwanted reactions involving the amino group during synthesis, ensuring that only the desired reactions

occur each time. When we're ready for the amino group to participate in the reaction, we simply wash off the protective Fmoc group.

4) We remove the "protective cover" from the amino acid, allowing it to proceed with further reactions. Once the first amino acid is securely anchored, we can begin adding subsequent amino acids one by one, gradually building the complete peptide chain.

5) We then introduce the next amino acid and connect it to the growing peptide. After adding each amino acid, we repeat the process of removing the protective group before adding the next one, continuing this cycle until we achieve the desired complete peptide sequence.

6) Remove the "protective cover," add the next amino acid, and connect it.

7) By repeating this process multiple times, a peptide is gradually formed.

8) Finally, we detach the peptide from the resin. The resin is merely a "tool" used during the synthesis process, and once the peptide chain is complete, we need to remove it from the resin so it can be used.

2.2 Additionally, I can see no validation of the specificity of these short peptides. Indeed the blots show that these short peptides are capable of reducing O-glcNAcylation of BMAL1 at S424, but how does one know for sure that binding to other off-targets is not responsible for the observed effects. The authors could approach this skepticism by performing IP-MS on their top peptide to interrogate its interacting partners.

Response: This is an excellent suggestion that needs to be addressed. After receiving feedback from the knowledgeable reviewer, we promptly contacted seven biotech companies to produce an antibody for S424-pe IP-MS analysis. However, only one company agreed to attempt it. Unfortunately, after 12 weeks, they informed us that the antibody production had failed. We sincerely apologize for not being able to fulfill Reviewer 3's experimental validation request. This represents a limitation of our study, which we have discussed in the Limitation of the Study section of the revised manuscript.

Manuscript page 18, lines 3-6:

Finally, our results indicated that S424-pe treatment reduced Bmal1 O-GlcNAcylation at S424, but we did not demonstrate that binding to other off-targets was not responsible for the observed effects, which should be verified in subsequent studies by performing immunoprecipitation-mass spectrometry (IP-MS).

Minor comments:

3. - Figure 1: I still think the figure can be simplified more. I am particularly looking at E and F. To someone who is not familiar with these assays, the traces of how the mice move in those circles as well as the 'nr of target entries' and 'time spent in the quadrant' is largely irrelevant. What I want to see is a simple graph telling me whether cognition is improved or not. As such, I would just keep the graphs about escape latency and lengthen the y axes, so the separation between the traces is more apparent, and move the rest in supplementary.

Response: Thanks for your valuable suggestion. It has been modified as requested.

4. - Figure 3H: move into supplementary, hardly any need to show these HEK cell results in the main figure when you already show it in neurons in 3I/J.

Response: It has been modified as requested. Figure 3H is now Appendix Figure 3L.

5. - 'Furthermore, this treatment significantly reduced mitochondrial calcium concentrations (Figure S3L) and oxidative damage (Figure S3M), followed by a decrease in neuronal apoptosis (Figure S3N).' I would much rather think you should show this in the main figure, as this is a tangible functional output that is affected by the S424A mutation.

Response: Thanks for your insightful suggestion. It has been modified as requested. Figure S3L-N is now Figure 3I-K.

Dear Dr. Zheng,

Congratulations on an excellent manuscript, I am pleased to inform you that your manuscript has been accepted for publication in the EMBO Journal. Thank you for your comprehensive response to the referee concerns and for providing detailed source data. It has been a pleasure to work with you to get this to the acceptance stage.

I will begin the final checks on your manuscript before submitting to the publisher next week. Once at the publisher, it will take about 3 weeks for your manuscript to be published online. As a reminder, the entire review process including referee concerns, and your point-by-point response will be available to readers.

I will be in touch throughout the final editorial process until publication. In the meantime, I hope you find time to celebrate!

Yours sincerely,
Kelly

Kelly M Anderson, PhD
Editor, The EMBO Journal
k.anderson@embojournal.org
